# Optimal Regret of Bandits under Differential Privacy

**Achraf Azize**[*]
FairPlay Joint Team
CREST, ENSAE Paris
achraf.azize@ensae.fr

**Yulian Wu**[†]
King Abdullah University of Science & Technology (KAUST)
Thuwal 23955-6900, Kingdom of Saudi Arabia
yulian.wu@kaust.edu.sa

**Junya Honda**
Kyoto University
RIKEN AIP
honda@i.kyoto-u.ac.jp

**Francesco Orabona**
King Abdullah University for Science & Technology (KAUST)
Thuwal 23955-6900, Kingdom of Saudi Arabia
francesco@orabona.com

**Shinji Ito**
The University of Tokyo
RIKEN AIP
shinji@mist.i.u-tokyo.ac.jp

**Debabrota Basu**
Univ. Lille, Inria, CNRS
Centrale Lille, UMR 9189-CRIStAL
debabrota.basu@inria.fr

## Abstract

As sequential learning algorithms are increasingly applied to real life, ensuring data privacy while maintaining their utilities emerges as a timely question. In this context, regret minimisation in stochastic bandits under $\epsilon$-global Differential Privacy (DP) has been widely studied. The present literature poses a significant gap between the best-known regret lower and upper bound in this setting, though they "match in order". Thus, we revisit the regret lower and upper bounds of $\epsilon$-global DP bandits and improve both. First, we prove a tighter regret lower bound involving a novel information-theoretic quantity characterising the hardness of $\epsilon$-global DP in stochastic bandits. This quantity smoothly interpolates between Kullback–Leibler divergence and Total Variation distance, depending on the privacy budget $\epsilon$. Then, we choose two asymptotically optimal bandit algorithms, *i.e.*, KL-UCB and IMED, and propose their DP versions using a unified blueprint, *i.e.*, (a) running in arm-dependent phases, and (b) adding Laplace noise to achieve privacy. For Bernoulli bandits, we analyse the regrets of these algorithms and show that their regrets asymptotically match our lower bound up to a constant arbitrary close to 1. At the core of our algorithms lies a new concentration inequality for sums of Bernoulli variables under Laplace mechanism, which is a new DP version of the Chernoff bound. Finally, our numerical experiments validate that DP-KLUCB and DP-IMED achieve lower regret than the existing $\epsilon$-global DP bandit algorithms.

## 1 Introduction

Multi-armed bandit is a classical setup of sequential decision-making under partial information, where the agent collects more information about an environment by interacting with it. To understand the setting, let us consider a clinical trial, where a doctor has $K$ candidate medicines to choose from and wants to recommend "effective" medicines to their patients. At each step $t$ of the trial, a new patient $p_t$ arrives, the doctor prescribes $a_t \in [K] \triangleq \{1, \ldots, K\}$ one of the $K$ medicines, and

---

[*]Part of the work is done during A. Azize's visit to Kyoto University and PhD at Scool Team, Inria Lille.
[†]Part of the work is done during Y. Wu's internship at RIKEN AIP and visit to Kyoto University.

39th Conference on Neural Information Processing Systems (NeurIPS 2025).

observes the reaction of the patient to the medicine. The observations are quantified as rewards, such that $r_t = 1$ if the patient $p_t$ is cured and 0 otherwise. To design an algorithm recommending "effective" medicines, the doctor can use a regret-minimising bandit algorithm [Thompson, 1933], *i.e.*, a bandit algorithm that aims to maximise the expected number of cured patients during the trial.

Following the trial, the doctor wants to release the trial results to the public, *i.e.*, the sequence of recommended medicines $(a_1, \ldots, a_T)$, in order to communicate the findings. However, the doctor fears that publishing the results may compromise the privacy of the patients who participated in the trial. Specifically, the rewards $(r_1, \ldots, r_T)$ constitute the private information that needs to be protected, since rewards in clinical trials may reveal sensitive information about the health condition of the patients. In addition to clinical trials, many applications of bandits, such as recommendation systems [Silva et al., 2022], online advertisement [Chen et al., 2014], crowd-sourcing [Zhou et al., 2014], user studies [Losada et al., 2022], hyper-parameter tuning [Li et al., 2017], communication networks [Lindståhl et al., 2022], and pandemic mitigation [Libin et al., 2019]), involve sensitive user data, and thus invokes the data privacy concerns. Motivated by the privacy concerns in bandits, *we study the privacy-utility trade-off in stochastic multi-armed bandits*.

We adhere to Differential Privacy (DP) [Dwork and Roth, 2014] as the privacy framework, and regret minimisation [Auer et al., 2002] in stochastic bandits as the utility measure. DP has been studied for multi-armed bandits under different bandit settings: finite-armed stochastic [Mishra and Thakurta, 2015, Sajed and Sheffet, 2019, Zheng et al., 2020a, Hu et al., 2021, Azize and Basu, 2022, Hu and Hegde, 2022, Azize and Basu, 2024, Wang and Zhu, 2024], adversarial [Thakurta and Smith, 2013, Agarwal and Singh, 2017, Tossou and Dimitrakakis, 2017], linear [Hanna et al., 2022, Li et al., 2022, Azize and Basu, 2024], contextual linear [Shariff and Sheffet, 2018, Neel and Roth, 2018, Zheng et al., 2020b, Azize and Basu, 2024], and kernel bandits [Pavlovic et al., 2025], among others. Most of these works were for regret minimisation, but the problem has also been explored for best-arm identification, with fixed confidence [Azize et al., 2023, 2024] and fixed budget [Chen et al., 2024]. The problem has also been studied under three different DP trust models: (a) global DP where the users trust the centralised decision maker [Mishra and Thakurta, 2015, Shariff and Sheffet, 2018, Sajed and Sheffet, 2019, Azize and Basu, 2022, Hu and Hegde, 2022], (b) local DP where each user deploys a local perturbation mechanism to send a "noisy" version of the rewards to the policy [Basu et al., 2019, Zheng et al., 2020a,b, Han et al., 2021], and (c) shuffle DP where users still feed their data to a local perturbation, but now they trust an intermediary to apply a uniformly random permutation on all users' data before sending to the central servers [Tenenbaum et al., 2021, Garcelon et al., 2022, Chowdhury and Zhou, 2022]. In this paper, *we focus on $\epsilon$-pure DP, under a global trust model, in stochastic finite-armed bandits, with the aim of regret minimisation*.

**Related Works.** This problem setting has been studied by Mishra and Thakurta [2015], Sajed and Sheffet [2019], Hu et al. [2021], Azize and Basu [2022], Hu and Hegde [2022]. All the regret upper and lower bounds in this setting are summarised in Table 1. DP-UCB [Mishra and Thakurta, 2015] was the first DP version of the Upper Confidence Bound (UCB) algorithm [Auer et al., 2002] that achieved logarithmic regret. DP-UCB uses the tree-based mechanism [Dwork et al., 2010, Chan et al., 2011] to compute privately the sum of rewards. For each arm, the tree mechanism maintains a binary tree of depth $\log(T)$ over the $T$ streaming reward observations. As a result, the noise added to the sum of rewards has a scale of $\mathcal{O}\left(\log^{2.5}(T)/\epsilon\right)$ for rewards in $[0, 1]$. DP-UCB builds a high probability upper bound on the means using the noisy sum of rewards to design a private UCB index and yields a regret bound of $\mathcal{O}\left(\sum_a \frac{\log(T)}{\Delta_a} + K \log^{2.5}(T)/\epsilon\right)$, where $\Delta_a$ is the difference between the mean reward of an optimal arm and arm $a$. This upper bound has an additional $\log^{1.5}(T)$ factor compared to the $\Omega(K \log(T)/\epsilon)$ regret lower bound, first proved by Shariff and Sheffet [2018].

DP-SE [Sajed and Sheffet, 2019] was the first DP bandit algorithm to eliminate the additional multiplicative factor $\log^{1.5}(T)$ in the regret. DP-SE is a DP version of the Successive Elimination algorithm [Even-Dar et al., 2002]. DP-SE runs in *independent* episodes. At each episode, the algorithm explores a set of active arms uniformly. At the end of an episode, DP-SE eliminates provably sub-optimal arms, but *only uses the samples collected at the current episode* to decide the arms to eliminate. Due to the addition of the Laplace noise to the sum of rewards, each arm is explored longer, resulting in the additional $\mathcal{O}\left(K \log(T)/\epsilon\right)$ in the regret.

A careful reading of DP-SE suggests that running the algorithm in independent episodes while forgetting the previous samples shreds the extra $\log^{1.5}(T)$ in the regret. These ingredients, *i.e.*, running in

Table 1: A summary of regret upper and lower bounds for $\epsilon$-global DP bandits.

| | Regret Upper Bound | Regret Lower Bound |
|---|---|---|
| Mishra and Thakurta [2015] | $\mathcal{O}\left(\frac{K\log(T)^{2.5}}{\epsilon} + \sum_{a \neq a^*} \frac{\log(T)}{\Delta_a}\right)$ (DP-UCB) | – |
| Sajed and Sheffet [2019] | $\mathcal{O}\left(\frac{K\log(T)}{\epsilon} + \sum_{a \neq a^*} \frac{\log(T)}{\Delta_a}\right)$ (DP-SE) | $\Omega\left(\frac{K\log(T)}{\epsilon}\right)$ |
| Hu and Hegde [2022] | $\mathcal{O}\left(\sum_{a \neq a^*} \frac{\Delta_a \log(T)}{\min(\Delta_a^2, \epsilon\Delta_a)}\right)$ (Lazy-DP-TS) | – |
| Azize and Basu [2022] | $\mathcal{O}\left(\sum_{a \neq a^*} \frac{\Delta_a \log(T)}{\min(\Delta_a^2, \epsilon\Delta_a)}\right)$ (AdaP-UCB) | $\sum_{a \neq a^*} \frac{\Delta_a \log(T)}{\min(\mathrm{kl}(\mu_a, \mu^*), 6\epsilon\Delta_a)}$ |
| **Our results** | $\alpha \sum_{a \neq a^*} \frac{\Delta_a \log(T)}{\mathrm{d}_\epsilon(\mu_a, \mu^*)}$ (Thm. 2, $\forall \alpha > 1$) | $\sum_{a \neq a^*} \frac{\Delta_a \log(T)}{\mathrm{d}_\epsilon(\mu_a, \mu^*)}$ (Thm. 1) |

independent phases with forgetting and adding Laplace noise, have been further adapted to UCB in Hu et al. [2021], Azize and Basu [2022], Wu et al. [2023] and to Thompson Sampling in Hu and Hegde [2022]. The state-of-the art regret upper bound is thus $\mathcal{O}\left(\sum_a \log(T)/\min\{\Delta_a, \epsilon\}\right)$. Similarly, Azize and Basu [2022] use the same three components of doubling, forgetting, and Laplace mechanism to propose AdaP-KLUCB that achieves a regret uppe bound of $\frac{C_1(\tau)\Delta_a}{\min\{\mathrm{kl}(\mu_a, \mu^*), C_2\epsilon\Delta_a\}}\log(T)$ for $\tau > 3$. Though the regret of AdaP-KLUCB is order-optimal, we observe that $C_1(\tau)$ and $C_2$ are not universal constants, *i.e.*, may depend on the environment.

On the other hand, Azize and Basu [2022] improve the problem-dependent regret lower bound of Shariff and Sheffet [2018] to $\sum_a \log(T)\frac{\Delta_a}{\min(d_a, 6\epsilon t_a)}$. Here, $d_a$ is the Kullback-Leibler (KL) indistinguishability gap for arm $a$ characterising the hardness of non-private bandits [Lai and Robbins, 1985], and $t_a$ is a "Total Variation" (TV) version of $d_a$ characterising the hardness of private bandits. For Bernoulli bandits, $t_a = \Delta_a$ and $d_a = \mathrm{kl}(\mu_a, \mu^*)$. Under the approximation $d_a \approx \Delta_a^2$, the lower bound of Azize and Basu [2022] recovers that of Shariff and Sheffet [2018], and the regret upper bounds of Sajed and Sheffet [2019], Azize and Basu [2022], Hu and Hegde [2022] match approximately the lower bound. However, this approximation can be arbitrarily bad, exposing a gap between the state-of-the-art upper and lower bounds in DP bandits. This motivates us to ask:

Q1. *Can we derive matching regret upper and lower bounds* up to the same constant *for $\epsilon$-global DP bandits?*

Additionally, following the triumph of doubling and forgetting as an algorithmic blueprint in DP bandits, Hu et al. [2021] conjectured that forgetting is necessary for designing any $\epsilon$-global DP bandit algorithm with an optimal regret upper bound matching the lower bound. Thus, we wonder:

Q2. *Is it possible to design an optimal $\epsilon$-global DP bandit algorithm without applying* forgetting*?*

**Aim and Contributions.** To address these questions, we revisit regret minimisation for Bernoulli bandits under $\epsilon$-global DP. Our main goal is to provide matching regret upper and lower bounds *up to the same constant*. Answering this question leads to the following contributions:

1. *Tighter regret lower bound*: In Theorem 1, we provide a new asymptotic regret lower bound for any consistent $\epsilon$-global DP policy. This result is a strict improvement over the lower bound of Azize and Basu [2022] for all $\epsilon$. This lower bound depends on a new information-theoretic quantity $\mathrm{d}_\epsilon$ (Eq. (6)) interpolating smoothly between KL and TV depending on $\epsilon$. This quantity also indicates a smooth transition between high and low privacy regimes, where the impact of DP does and does not appear, respectively. In addition to the existing techniques, our proof applies a new "double change" of environment idea to couple the impacts of DP and bandit feedback (Lemma 1).

2. *Tighter concentration inequality*: In Proposition 1, we provide a DP version of Chernoff-style concentration bound for sum of Bernoullis with added Laplace noise. $\mathrm{d}_\epsilon$ naturally appears in this bound. Also, the bound suggests that as long as the number of summed Laplace noise is negligible compared to the number of summed Bernoullis, the effect of the noise is comparable to having one Laplace noise in the dominant term of the bound. This bound is universally interesting for DP literature as the concentrations of random variables and Laplace noises are commonly treated separately unlike the coupled treatment in Proposition 1.

3. *Algorithm design and tighter regret upper bounds*: Based on the concentration bound of Proposition 1, we modify the generic blueprint used by Sajed and Sheffet [2019], Azize and Basu [2022], Hu and Hegde [2022]. We (a) get rid of "reward-forgetting" and thus sum all rewards at each phase, and (b) develop new private indexes using $d_\epsilon$. We also run the algorithms in geometrically increasing arm-dependent batches, with ratio $\alpha > 1$. We instantiate these modifications for two algorithms that achieve constant optimal regrets withour privacy, *i.e.*, KL-UCB and IMED, to propose DP-KLUCB and DP-IMED (Algorithm 1). We analyse the regret of both algorithms (Theorem 2) and show that their regret upper bounds match asymptotically the regret lower bound of Theorem 1 up to the constant $\alpha$, which can be set arbitrarily close to 1, for all bandit instances and values of $\epsilon$.

We also validate experimentally that our algorithms DP-IMED and DP-KLUCB achieve the lowest regret among DP bandit algorithms in the literature. Finally, in Appendix B, we extend the adaptive continual release model of Jain et al. [2023] to bandits and show that this definition is equivalent to the classic $\epsilon$-global DP notion adopted in the DP bandit literature [Mishra and Thakurta, 2015, Azize and Basu, 2022, 2024]. This result can be of independent interest.

## 2 Background: Regret Minimisation and Differential Privacy in Bandits

In this section, we formalise the essential components of our work, *i.e.*, the stochastic bandit problem, regret minimisation as a utility measure, and Differential Privacy (DP) as the privacy constraint.

**Stochastic Bandits.** A stochastic bandit problem is a sequential game between a policy $\pi$ and a stochastic environment $\nu$ [Thompson, 1933, Lai and Robbins, 1985]. The game is played over $T$ rounds, where $T \in \{1, 2, \dots\}$ is a natural number called the *horizon*. At each step $t \in \{1, \dots, T\}$, the policy $\pi$ chooses an action $a_t \in [K]$. The stochastic environment, which is a collection of distributions $\nu \triangleq (P_a : a \in [K])$, samples a reward $r_t \sim P_{a_t}$ and reveals it to the policy $\pi$. The interaction between the policy $\pi$ and environment $\nu \triangleq (P_a : a \in [K])$ over $T$ steps induces a probability measure on the sequence of outcomes $H_T \triangleq (a_1, r_1, a_2, r_2, \dots, a_T, r_T)$. Let each $P_a$ be a probability measure on $(\mathbb{R}, \mathcal{B}(\mathbb{R}))$ with $\mathfrak{B}$ being the Borel set. For each $t \in [T]$, let $\Omega_t = ([K] \times \mathbb{R})^t \subset \mathbb{R}^{2t}$ and $\mathcal{F}_t = \mathfrak{B}(\Omega_t)$. First, we formalise the definition of a policy.

**Definition 1** (Policy). *A policy $\pi$ is a sequence $(\pi_t)_{t=1}^T$ , where $\pi_t$ is a probability kernel from $(\Omega_t, \mathcal{F}_t)$ to $([K], 2^{[K]})$. Since $[K]$ is discrete, we adopt the convention that for $a \in [K]$, $\pi_t(a \mid a_1, r_1, \dots, a_{t-1}, r_{t-1}) = \pi_t(\{a\} \mid a_1, r_1, \dots, a_{t-1}, r_{t-1})$ .*

The interaction probability measure on $(\Omega_T, \mathcal{F}_T)$ depends on the environment and the policy: (a) the conditional distribution of action $a_t$ given $a_1, r_1, \dots, a_{t-1}, r_{t-1}$ is $\pi(a_t \mid H_{t-1})$, and (b) the conditional distribution of reward $r_t$ given $a_1, r_1, \dots, a_{t-1}, r_{t-1}, a_t$ is $P_{a_t}$. To construct the probability measure, let $\lambda$ be a $\sigma$-finite measure on $(\mathbb{R}, \mathcal{B}(\mathbb{R}))$ for which $P_a$ is absolutely continuous with respect to $\lambda$ for all $a \in [K]$. Let $p_a = dP_a/d\lambda$ be the Radon–Nikodym derivative of $P_a$ with respect to $\lambda$. Letting $\rho$ be the counting measure with $\rho(B) = |B|$, the density $p_{\nu\pi} : \Omega_T \to \mathbb{R}$ can now be defined with respect to the product measure $(\rho \times \lambda)^T$ by

$$p_{\nu\pi}(a_1, r_1, \dots, a_T, r_T) \triangleq \prod_{t=1}^T \pi_t(a_t \mid a_1, r_1, \dots, a_{t-1}, r_{t-1}) p_{a_t}(r_t) \tag{1}$$

and $\mathbb{P}_{\nu\pi}(B) \triangleq \int_B p_{\nu\pi}(\omega)(\rho \times \lambda)^T(\,d\omega)$ for all $B \in \mathcal{F}_T$. So $(\Omega_T, \mathcal{F}_T, \mathbb{P}_{\nu\pi})$ is a probability space over histories induced by the interaction between $\pi$ and $\nu$.

**Regret minimisation.** We study regret minimisation as the utility measure [Lai and Robbins, 1985]. Informally, the regret of a policy is the deficit suffered by the learner relative to the optimal policy which knows the environment and always plays the optimal arm. Let $\nu = (P_a : a \in [K])$ a bandit instance and define $\mu_a(\nu) = \int_{-\infty}^{\infty} x \, dP_a(x)$ the mean of arm $a$'s reward distribution. We assume throughout that $\mu_a(\nu)$ exists and is finite for all actions. Let $\mu^\star(\nu) = \max_{a \in [K]} \mu_a(\nu)$ the largest mean among all the arms. The regret of policy $\pi$ on bandit instance $\nu$ is

$$\text{Reg}_T(\pi, \nu) \triangleq T\mu^\star(\nu) - \mathbb{E}_{\nu\pi}\left[\sum_{t=1}^T r_t\right] = \sum_{a=1}^K \Delta_a(\nu)\mathbb{E}_{\nu\pi}\left[N_a(T)\right]. \tag{2}$$

where $N_a(T) \triangleq \sum_{t=1}^{T} \mathbb{1}\{a_t = a\}$ and $\Delta_a(\nu) \triangleq \mu^\star(\nu) - \mu_a(\nu)$. The expectation is taken with respect to the probability measure $\mathbb{P}_{\nu\pi}$ on action-reward sequences induced by the interaction of $\pi$ and $\nu$. Hereafter, we drop the dependence on $\nu$ when the context is clear.

For many classes of bandits, it is possible to define a notion of instance-dependent optimality that characterises the hardness of regret minimisation. Specifically, for any consistent policy $\pi$ over a class of bandits $\mathcal{E} \triangleq \mathcal{M}_1 \times \cdots \times \mathcal{M}_K$, i.e., a policy $\pi \in \Pi_{\text{cons}}(\mathcal{E})$ verifies $\lim_{T \to \infty} \frac{\text{Reg}_T(\pi, \nu)}{T^p} = 0$ for all $\nu \in \mathcal{E}$ and all $p > 0$, then the regret of $\pi$ on any environment $\nu \in \mathcal{E}$ is lower bounded by

$$\liminf_{T \to \infty} \frac{\text{Reg}_T(\pi, \nu)}{\log(T)} \geq \sum_{a:\Delta_a(\nu)>0} \frac{\Delta_a(\nu)}{\text{KL}_{\inf}(P_a, \mu^\star, \mathcal{M}_a)}, \tag{3}$$

where $\text{KL}_{\inf}(P, \mu^\star, \mathcal{M}) \triangleq \inf_{P' \in \mathcal{M}} \{\text{KL}(P, P') : \mu(P') > \mu^\star\}$, and KL is the Kullback-Leibler divergence, i.e., for two probability distributions $P, Q$ on $(\Omega, \mathcal{F})$, the KL divergence is $\text{KL}(P, Q) \triangleq \int \log\left(\frac{dP}{dQ}(\omega)\right) dP(\omega)$ when $P \ll Q$, and $+\infty$ otherwise. The lower bound of Equation (3) is tight for many classes of bandits, and the "KL-inf" is a fundamental quantity that characterises the complexity of regret minimisation in bandits.

**Bernoulli bandits.** A Bernoulli bandit is a stochastic environment where the distribution of each arm follows a Bernoulli distribution. Let $\mu \in [0, 1]^K$, then $\nu_\mu^{\mathcal{B}} = (\text{Bernoulli}(\mu_a) : a \in [K])$ is a Bernoulli environment. For Bernoulli bandits, $\text{KL}_{\inf}(P_a, \mu^\star, \mathcal{M}_a) = \text{kl}(\mu_a, \mu^\star)$, where kl is the relative entropy between Bernoullis, i.e., $\text{kl}(p, q) \triangleq p \log(p/q) + (1 - p) \log((1 - p)/(1 - q))$ for $p, q \in [0, 1]$ and singularities are defined by taking limits. Using the "optimism in the face of uncertainty" principle, it is possible to design algorithms tailored for Bernoulli bandits, such as KL-UCB [Cappé et al., 2013] or IMED [Honda and Takemura, 2015], that achieve the lower bound of Equation (3) asymptotically, *up to the same constant*.

**Differential Privacy (DP).** DP [Dwork and Roth, 2014] guarantees that any sequence of algorithm outputs is "essentially" equally likely to occur, regardless of the presence or absence of any individual. The probabilities are taken over random choices made by the algorithm, and "essentially" is captured by closeness parameters that we call privacy budgets. Formally, DP is a constraint on the class of mechanisms, where a mechanism $\mathcal{M}$ is a randomised algorithm that takes as input a dataset $D \triangleq \{x_1, \ldots, x_T\} \in \mathcal{X}^T$ and outputs $o \sim \mathcal{M}_D$. The probability space is over the coin flips of the mechanism $\mathcal{M}$. Given some event $E$ in the output space $(\mathcal{O}, \mathcal{F})$, we note $\mathcal{M}_D(E) \triangleq \mathcal{M}(E|D)$ the probability of observing the event $E$ given that the input of the mechanism is $D$.

**Definition 2** ($\epsilon$-DP [Dwork et al., 2006]). *A mechanism $\mathcal{M}$ satisfies $\epsilon$-DP for a given $\epsilon \geq 0$, if*

$$\forall D \sim D', \ \forall E \in \mathcal{O}, \ \mathcal{M}_D(E) \leq e^\epsilon \mathcal{M}_{D'}(E), \tag{4}$$

*where $D \sim D'$ if and only if $d_{Ham}(D, D') \triangleq \sum_{t=1}^{T} \mathbb{1}\{D_t \neq D'_t\} \leq 1$, i.e., $D$ and $D'$ differ by at most one record, and are said to be neighbouring datasets.*

DP is widely adopted as a privacy framework since the definition enjoys different interesting properties, and can be achieved by combining simple basic mechanisms. Hereafter, we mainly use two important DP properties: post-processing (Proposition 4) and group privacy (Proposition 5), and we use the Laplace mechanism (Theorem 5) to achieve DP.

**Bandits under DP.** We extend DP to bandits by reducing a policy $\pi = (\pi_1, \ldots, \pi_T)$ to a "batch" mechanism $\mathcal{M}^\pi$ [Azize and Basu, 2024]. Different ways of reducing a policy to a batch mechanism differ on the input representation and the nature of the mechanism.

(a) In Table DP, we represent each user $u_t$ by the vector $x_t \triangleq (x_{t,1}, \ldots, x_{t,K}) \in \mathbb{R}^K$ of all its $K$ "potential rewards." This is the vector of potential rewards since the policy only observes $r_t \triangleq x_{t,a_t}$ when it recommends action $a_t$. In Table DP, the induced "batch" mechanism $\mathcal{M}^\pi$ from the policy $\pi$ takes as input a table of rewards $\mathbf{x} \triangleq \{(x_{t,i})_{i \in [K]}\}_{t \in [T]} \in (\mathbb{R}^K)^T$, and outputs a sequence of actions $\mathbf{a} \triangleq (a_1, \ldots, a_T) \in [K]^T$ with probability $\mathcal{M}_\mathbf{x}^\pi(\mathbf{a}) \triangleq \prod_{t=1}^{T} \pi_t\left(a_t | a_1, x_{1,a_1}, \ldots a_{t-1}, x_{t-1,a_{t-1}}\right)$. This is the probability of observing $(a_1, \ldots, a_T)$ when $\pi$ interacts with the table of rewards $\mathbf{x}$. $\mathcal{M}_\mathbf{x}^\pi$ is a distribution over sequences of actions since $\sum_{\mathbf{a} \in [K]^T} \mathcal{M}_\mathbf{x}^\pi(\mathbf{a}) = 1$.

(b) In View DP, the induced "batch" mechanism from the policy $\pi$ takes as input a list of rewards and outputs a sequence of actions. The difference is in the representation of the input dataset. Since in bandits, the policy only observes the reward corresponding to the action chosen, another natural choice for the input is a list of rewards, *i.e.*, $\mathbf{r} \triangleq \{r_1, \ldots, r_T\} \in \mathbb{R}^T$. Now, the induced "batch" mechanism $\mathcal{V}^\pi$ from the policy $\pi$ takes as input a list of rewards $\mathbf{r} \triangleq \{r_1, \ldots, r_T\} \in \mathbb{R}^T$, and outputs a sequence of actions $\mathbf{a} \triangleq (a_1, \ldots, a_T) \in [K]^T$, with probability $\mathcal{V}_{\mathbf{r}}^\pi(\mathbf{a}) \triangleq \prod_{t=1}^{T} \pi_t(a_t | a_1, r_1, \ldots a_{t-1}, r_{t-1})$. This is the probability of observing $\mathbf{a}$ when $\pi$ interacts with $\mathbf{r}$. $\mathcal{V}_{\mathbf{r}}^\pi$ is a distribution over sequences of actions, since $\sum_{\mathbf{a} \in [K]^T} \mathcal{V}_{\mathbf{r}}^\pi(\mathbf{a}) = 1$.

**Definition 3** (Table DP and View DP [Azize and Basu, 2024]). *(a) A policy $\pi$ satisfies $\epsilon$-Table DP if and only if $\mathcal{M}^\pi$ is $\epsilon$-DP. (b) A policy $\pi$ satisfies $\epsilon$-View DP if and only if $\mathcal{V}^\pi$ is $\epsilon$-DP.*

Table DP and View DP have been formalised in Azize and Basu [2024], and have been used interchangeably in the private bandit literature. For $\epsilon$-pure, Proposition 1 in Azize and Basu [2024] shows that these two definitions are equivalent.

Thus, we refer to any policy that verifies $\epsilon$-Table DP or $\epsilon$-View DP as an $\epsilon$-**global DP** policy. In Appendix B, we also extend the interactive DP definition of Jain et al. [2023] to bandits and show that $\epsilon$-global DP is equivalent to it. In the following, our main goal is to design an $\epsilon$-global DP policy that minimises the regret $\mathrm{Reg}_T(\pi, \nu)$ on any Bernoulli environment $\nu$.

## 3 Regret Lower Bound under $\epsilon$-global DP

In this section, we present a new regret lower bound for bandits under $\epsilon$-global DP. We compare this result to the lower bound of Azize and Basu [2022], and provide a proof.

**Theorem 1** (Regret lower bound under $\epsilon$-global DP). *For every $\epsilon$-global DP consistent policy over the class of Bernoulli bandits, we have*

$$\liminf_{T \to \infty} \frac{\mathrm{Reg}_T(\pi, \nu)}{\log(T)} \geq \sum_{a:\Delta_a > 0} \frac{\Delta_a}{d_\epsilon(\mu_a, \mu^\star)}, \tag{5}$$

*where*

$$\mathrm{d}_\epsilon(x, y) \triangleq \inf_{z \in [x \wedge y, x \vee y]} \{\epsilon |z - x| + \mathrm{kl}(z, y)\}, \quad x \in \mathbb{R}, y \in [0, 1]. \tag{6}$$

For any suboptimal arm $a$, $\mu^\star > \mu_a$ and $\mathrm{d}_\epsilon(\mu_a, \mu^\star) = \inf_{\mu \in [\mu_a, \mu^\star]} \{\epsilon(\mu - \mu_a) + \mathrm{kl}(\mu, \mu^\star)\}$.

**Implications of Theorem 1.** (a) Theorem 1 improves the lower bound of Azize and Basu [2022]. Specifically, Theorem 3 in Azize and Basu [2022] gives a lower bound

$$\liminf_{T \to \infty} \frac{\mathrm{Reg}_T(\pi, \nu)}{\log(T)} \geq \sum_{a:\Delta_a > 0} \frac{\Delta_a}{\min\{\mathrm{kl}(\mu_a, \mu^\star), 6\epsilon\Delta_a\}}. \tag{7}$$

Theorem 1 is a strict improvement on the lower bound of Azize and Basu [2022] since $\mathrm{d}_\epsilon(\mu_a, \mu^\star) \leq \min\{\mathrm{kl}(\mu_a, \mu^\star), \epsilon\Delta_a\} \leq \min\{\mathrm{kl}(\mu_a, \mu^\star), 6\epsilon\Delta_a\}$, for any $\epsilon, \mu_a$ and $\mu^\star$.

(b) Solving the constrained optimisation problem defining $\mathrm{d}_\epsilon$ for Bernoulli variables gives

$$\mathrm{d}_\epsilon(\mu_a, \mu^\star) = \begin{cases} \mathrm{kl}(\mu_a, \mu^\star) & \text{if } \epsilon \geq \log \frac{\mu^\star}{\mu_a} + \log \frac{1 - \mu_a}{1 - \mu^\star} \\ \mathrm{kl}\left(\frac{\mu^\star}{\mu^\star + (1 - \mu^\star)e^\epsilon}, \mu^\star\right) + \epsilon\left(\frac{\mu^\star}{\mu^\star + (1 - \mu^\star)e^\epsilon} - \mu_a\right) & \text{if not} \end{cases} \tag{8}$$

This suggests the existence of two privacy regimes: a low privacy regime when $\epsilon \geq \log \frac{\mu^\star}{\mu_a} + \log \frac{1 - \mu_a}{1 - \mu^\star}$, and a high privacy regime when $\epsilon \leq \log \frac{\mu^\star}{\mu_a} + \log \frac{1 - \mu_a}{1 - \mu^\star}$. In the low privacy regime, $\mathrm{d}_\epsilon(\mu_a, \mu^\star)$ just reduces to the non-private $\mathrm{kl}(\mu_a, \mu^\star)$, and privacy can be achieved for *free*. In the high privacy regime, $\mathrm{d}_\epsilon(\mu_a, \mu^\star)$ can be written as the sum of two terms, *i.e.*, a KL term between Bernoullis with means $\frac{\mu^\star}{\mu^\star + (1 - \mu^\star)e^\epsilon}$ and $\mu^\star$, and TV distance between Bernoullis with means $\frac{\mu^\star}{\mu^\star + (1 - \mu^\star)e^\epsilon}$ and $\mu_a$. At the limit, we have that $\mathrm{d}_\epsilon(\mu_a, \mu^\star) \sim_{\epsilon \to 0} \epsilon \times \Delta_a$.

(c) Theorem 1 can be generalised beyond Bernoulli bandits: for a class $\mathcal{E}$ of unstructured stochastic bandits, *i.e.*, $\mathcal{E} \triangleq \mathcal{M}_1 \times \cdots \times \mathcal{M}_K$, the lower bound becomes

$$\liminf_{T \to \infty} \frac{\text{Reg}_T(\pi, \nu)}{\log(T)} \geq \sum_{a:\Delta_a > 0} \frac{\Delta_a}{\text{d}_{\text{inf}}(P_a, \mu^\star, \mathcal{M}_a, \epsilon)}, \tag{9}$$

where $\text{d}_{\text{inf}}(P_a, \mu^\star, \mathcal{M}_a, \epsilon) \triangleq \inf_{P' \in \mathcal{M}_a} \left\{ \text{d}_\epsilon^{\mathcal{M}_a}(P_a, P') : \mu(P') > \mu^\star \right\}$, and

$$\text{d}_\epsilon^{\mathcal{M}_a}(P_a, P') \triangleq \inf_{Q \in \mathcal{M}_a} \left\{ \epsilon \text{TV}(P_a, Q) + \text{KL}(Q, P') : \mu(P_a) \leq \mu(Q) \leq \mu(P') \right\},$$

for $P_a, P' \in \mathcal{M}_a$ such that $\mu(P_a) \leq \mu(P')$.

**Key Changes in Proof Techniques.** The proof improves the lower bound of Azize and Basu [2022] by introducing a "double" change of environment. (a) The first change of environment uses the group privacy property of the policy, and thus the TV transport. (b) The second change uses the classic "Lai-Robbins" change of measure and thus the KL transport. By optimising for the "in-between" environment, the double change always has smaller transport than any route led by purely KL or TV transport. The detailed proof is in Appendix C.

## 4 Algorithm Design and Regret Analysis

In this section, we propose two algorithms, DP-KLUCB and DP-IMED, presented in Algorithm 1. At the core of our algorithm design lies a new concentration bound for $\epsilon$-DP means of Bernoulli variables (Proposition 1). We analyse both the privacy and regret of our proposed algorithms, and show that their regret upper bound matches the lower bound up to a constant arbitrary close to 1.

First, we start with the concentration inequality for the private mean of IID Bernoullis.

**Proposition 1** (Concentration Bound of Private Mean). *For $\mu \in (0, 1)$ and $\epsilon > 0$, let $\tilde{S}_{n,m} = \sum_{i=1}^{n} X_i + \sum_{j=1}^{m} Y_j$, where $X_i \sim \text{Ber}(\mu)$ and $Y_j \sim \text{Lap}(1/\epsilon)$, be the sum of $n$ independent Bernoulli random variables with mean $\mu$ and $m$ independent Laplace variables with scale $1/\epsilon$. Let $x \in [0, 1]$ and $\{n_m\}_{m \in \mathbb{N}}$ be a sequence such that $m/n_m = o(1)$. Then, for any $a > 0$ there exists a constant $A_a > 0$ such that for all $m \in \mathbb{N}$,*

$$\Pr\left[ \frac{\tilde{S}_{n_m,m}}{n_m} \leq x \right] \leq A_a \text{e}^{-n_m(\text{d}_\epsilon(x,\mu)-a)}, \text{ for } x \leq \mu; \quad \Pr\left[ \frac{\tilde{S}_{n_m,m}}{n_m} \geq x \right] \leq A_a \text{e}^{-n_m(\text{d}_\epsilon(x,\mu)-a)}, \text{ for } x \geq \mu.$$

*We recall that $\text{d}_\epsilon(x, y) \triangleq \inf_{z \in [x \wedge y, x \vee y]} \text{kl}(z, y) + \epsilon|z - x|$.*

**Discussions.** (a) This concentration bound can be seen as a private version of the Chernoff bound (Lemma 11), where $\text{d}_\epsilon$ replaces the $\text{kl}$ in the exponent. (b) As soon as the number of summed Laplace noises $m$ is negligible with respect to the number of summed Bernoulli variables $n$, then the effect of $m$ on the dominant term is similar to when $m = 1$. (c) This concentration bound is a tighter version of Lemma 4 in Azize and Basu [2022] with $m = 1$. Lemma 4 of Azize and Basu [2022] and other works in bandits under DP [Mishra and Thakurta, 2015, Sajed and Sheffet, 2019, Hu et al., 2021, Hu and Hegde, 2022] deal with the concentration of the noise and random variables separately– they use an inequality $\Pr(X + Y \geq a) \leq \Pr(X \geq a) + \Pr(Y \geq 0)$, followed by a classic non-private concentration bound for the first term and concentration bound of Laplace noise for the second term. We improve this loose analysis by a coupled treatment of noise and variables.

**Proof Sketch.** Proposition 1 is a corollary of the general Lemma 5 that holds for any $n$ and $m$. To prove Lemma 5, we express $\Pr\left[ \tilde{S}_{n,m} \geq x \right]$ in the form of a convolution of the sums of Bernoulli rewards and Laplace noises. Even though we still resort to the Chernoff bound for each of the sums, considering the convolution of sums significantly improves the bound compared with the naïve use of the Chernoff bounds for noise and variables in $\tilde{S}_{n,m}$. The complete proof is in Appendix D.

**Algorithm Design.** Based on Proposition 1, we propose DP-KLUCB and DP-IMED in Algorithm 1. Both algorithms run in arm-dependent phases (Line 9 in Algorithm 1), and add Laplace noise to achieve $\epsilon$-global DP (Line 10 in Algorithm 1). This is similar to the algorithm design in Sajed and Sheffet [2019], Azize and Basu [2022], Hu and Hegde [2022], with two modifications.

**Algorithm 1:** DP-KLUCB and DP-IMED

**Input:** $\epsilon$: privacy parameter, $K$: number of arms, $T$: horizon, $\{B_m\}_{m=0}^{\infty}$: batch sizes

1 Pull each arm $B_0$ times and receive rewards $\{\{X_{i,n}\}_{n=1}^{B_0}\}_{i=1}^K$;

2 Compute private reward sum $\tilde{S}_{i,0} = \sum_{n=1}^{B_0} X_{i,n} + Y_{i,0}$ for $Y_{i,0} \sim \mathrm{Lap}(1/\epsilon)$;

3 Compute private mean $\tilde{\mu}_{i,0} = \tilde{S}_{i,0}/B_0$;

4 Set arm-dependent epoch $m_i := 0$ for each arm $i \in [K]$;

5 Set cumulative pull number $n_{m_i} := B_0$ for each arm $i \in [K]$;

6 Set $t \leftarrow KB_0 + 1$;

7 **while** $t \leq T$ **do**

8    (DP-KLUCB): compute $i(t) \in \arg\max_i \ \bar{\mu}_i(t)$ maximising the DP-KLUCB index given by

$$\bar{\mu}_i(t) = \max\left\{ \mu : \mathrm{d}_\epsilon\left([\tilde{\mu}_{i,m_i}]_0^1, \mu\right) \leq \frac{\log t}{n_{m_i}} \right\} \tag{10}$$

   (DP-IMED): compute $i(t) \in \arg\min_i \ I_i(t)$ minimising the DP-IMED index given by

$$I_i(t) = n_{m_i} \mathrm{d}_\epsilon\left([\tilde{\mu}_{i,m_i}]_0^1, [\tilde{\mu}^*(t)]_0^1\right) + \log n_{m_i}, \tag{11}$$

   where $\tilde{\mu}^*(t) = \max_j \tilde{\mu}_{j,m_j}$ and $[x]_0^1 = \max\{0, \min\{x, 1\}\}$ is the clipping of $x$ onto $[0, 1]$;

9    Pull arm $i(t)$ for $B_{m_{i(t)}+1}$ times and receive rewards $\{X_{i(t),n}\}_{n=n_{m_{i(t)}}+1}^{n_{m_{i(t)}}+B_{m_{i(t)}+1}}$;

10    Update the noisy sum $\tilde{S}_{i(t),m_{i(t)}+1} \leftarrow \tilde{S}_{i(t),m_{i(t)}} + \sum_{n=n_{m_{i(t)}}+1}^{n_{m_{i(t)}}+B_{m_{i(t)}+1}} X_{i(t),n} + Y_{i(t),m_{i(t)}+1}$

   where $Y_{i(t),m_{i(t)}+1} \sim \mathrm{Lap}(1/\epsilon)$;

11    Compute private mean $\tilde{\mu}_{i(t),m_{i(t)}+1} = \tilde{S}_{i(t),m_{i(t)}+1}/n_{m_{i(t)}+1}$;

12    Update $m_{i(t)} \leftarrow m_{i(t)} + 1$, $n_{m_{i(t)}} \leftarrow n_{m_{i(t)}} + B_{m_{i(t)}}$, $t \leftarrow t + B_{m_{i(t)}}$;

13 **end**

(a) *Our algorithms do not forget rewards from previous phases.* In contrast, the algorithms of Sajed and Sheffet [2019], Azize and Basu [2022], Hu and Hegde [2022] run in adaptive and "non-overlapping" phases. The sums of rewards are computed over non-overlapping sequences. Thus, the rewards collected in the past phases are "thrown away" in the future phases. By running non-overlapping phases, these algorithms avoid the use of sequential composition (Proposition 6), and use instead the "parallel composition" property (Lemma 10) of DP to add less noise. Specifically, if the rewards are in $[0, 1]$, forgetting ensures that adding one $\mathrm{Lap}(1/\epsilon)$ to each sum of rewards is enough to make the simultaneous release of all the partial sums achieving DP. In our algorithms, we do *not* forget previous private sums (Line 10 in Algorithm 1). The price of not forgetting is adding multiple Laplace noises with scale $1/\epsilon$ to the non-private sum. To overcome this price, we use the insights from the concentration inequality of Proposition 1, *i.e.*, as long as the number of added Laplace noises is negligible with respect to the number of Bernoulli variables, the effect of the added noise on the dominant term is similar to having one Laplace noise. This refined analysis removes forgetting.

(b) *Our algorithms use new indexes*, *i.e.* Eq. (10) and Eq. (11), inspired by Proposition 1, and are based on the $\mathrm{d}_\epsilon$ quantity appearing in the lower bound. In addition, the index of DP-KLUCB is instantiated with an exploration bonus of $\log(t)/n_{m_i}$. This contrasts AdaP-KLUCB and Lazy-DP-TS, which need an exploration bonus of roughly $3\log(t)/n_{m_i}$ needed for their regret analysis.

Now, we present the privacy guarantee of our algorithms.

**Proposition 2** (Privacy analysis)**.** DP-KLUCB *and* DP-IMED *are $\epsilon$-global DP for rewards in* $[0, 1]$.

**Proof Sketch.** First, given a sequence of rewards $\{r_1, \ldots, r_T\} \in [0, 1]^T$ and some time steps $1 = t_1 < t_2 < \cdots < t_\ell = T + 1$, releasing the partial sums $\left\{ \left(\sum_{s=t_k}^{t_{k+1}-1} r_s\right) + Y_k \right\}_{k=1}^{\ell-1}$ is $\epsilon$-DP, where $Y_k \sim \mathrm{Lap}(1/\epsilon)$. This is the main privacy lemma used to design DP bandit algorithms in prior work [Sajed and Sheffet, 2019, Azize and Basu, 2022, Hu and Hegde, 2022]. Now, by the post-processing property of DP, we also have that releasing the sums $\left\{ \left(\sum_{s=1}^{t_{k+1}-1} r_s\right) + \sum_{p=1}^k Y_p \right\}_{k=1}^{\ell-1}$ is $\epsilon$-DP, by summing the outputs of the previous DP mechanism. Finally, DP-IMED and DP-KLUCB

are $\epsilon$-global DP by adaptive post-processing of the sum of rewards. The detailed proof is presented in Appendix E.

To have a "good" regret bound, Proposition 1 suggests using a batching strategy where the number of batches is sublinear in $T$. For simplicity, we chose the batch sizes $B_m$ in Algorithm 1 such that $B_m \approx n_0 \alpha^m$, *i.e.*, a geometric sequence with initialisation $n_0 \in \mathbb{N}$ and ratio $\alpha > 1$. Thus, we take

$$B_m = \left\lceil n_0 \frac{\alpha^{m+1} - 1}{\alpha - 1} \right\rceil - \left\lceil n_0 \frac{\alpha^m - 1}{\alpha - 1} \right\rceil , \tag{12}$$

where $\lceil x \rceil$ is the smallest integer no less than $x$. When $\alpha$ is an integer, $B_m = n_0 \alpha^m$.

**Theorem 2** (Regret upper bound of DP-IMED and DP-KLUCB)**.** *Assume $\mu^\star < 1$. Under the batch sizes given in Equation* (12) *with $\alpha > 1$, and for any Bernoulli bandit $\nu$, we have*

$$\mathrm{Reg}_T(\text{DP-IMED}, \nu) \leq \sum_{i \neq i^*} \frac{\alpha \Delta_i \log T}{\mathrm{d}_\epsilon(\mu_i, \mu^\star)} + o(\log T),$$

$$\mathrm{Reg}_T(\text{DP-KLUCB}, \nu) \leq \sum_{i \neq i^*} \frac{\alpha \Delta_i \log T}{\mathrm{d}_\epsilon(\mu_i, \mu^\star)} + o(\log T) .$$

**Comments.**  (a) The regret upper bounds of DP-IMED and DP-KLUCB match asymptotically the lower bound of Theorem 1 up to the constant $\alpha > 1$, where $\alpha$ is the ratio of the georemetrically increasing batch sizes $B_m$. This parameter $\alpha > 1$ can be set arbitrarily close to 1 to match the dominant term in the asymptotic regret lower bound. In addition, our analysis only requires that the number of batches is sublinear in $T$, as seen from Proposition 1. As a result, we can also use a polynomially increasing batch size instead of $B_m \approx \alpha^m$, which fully makes the regret *asymptotically optimal*. We used a geometrically increasing batch size here just for simplicity. (b) Our algorithms strictly improve over the regret upper bounds of Azize and Basu [2022], Hu and Hegde [2022]. Also, our upper bounds are the first to show a dependence in the tighter quantity $\mathrm{d}_\epsilon$, compared to having $\min\{\Delta_a^2, \epsilon \Delta_a\}$ in the regrets for Azize and Basu [2022], Hu and Hegde [2022]. We provide additional comments that compare our regret upper bound to that of AdaP-KLUCB in Appendix F.

**Proof Sketch.**  The proof uses similar steps as those of Honda and Takemura [2015] for the IMED algorithm and the reduction technique for the KL-UCB algorithm by Honda [2019] with the new concentration inequality involving $\mathrm{d}_\epsilon$ (Proposition 1). The main technical challenge is dealing with the adaptive batching strategy in the analysis. We control this by a regret decomposition tailored for batched pulls of arms where the property of IMED/KL-UCB index can still be naturally incorporated. The full proof is presented in Appendix F.

**Beyond Bernoulli Bandits.**  First, we highlight that some of our results are already valid beyond Bernoulli bandit instances: (a) As explained in the Implications of Theorem 1, our regret lower bound is already true for any class of distributions. (b) As expressed in Proposition 2, our algorithms are already $\epsilon$-DP for any distribution with bounded support on $[0, 1]$. This could easily be generalised to any bounded rewards on $[a, b]$ by multiplying the noise terms with the range $(b-a)$. On the other hand, the parts only valid for Bernoullis are: the concentration inequality (Proposition 1) and the regret upper bounds (Theorem 2). It is also worth noting that the same regret upper bound of Theorem 2 is also valid for distributions over $[0, 1]$, since we only used the Chernoff bound for Bernoulli distributions, which is also valid for distributions over $[0, 1]$. Both the concentration inequality and regret upper bound can be extended beyond Bernoullis, to say sub-Gaussian distributions or exponential families. However, what is less clear is whether it is possible to get matching upper and lower bounds up to constants, like we achieve in the Bernoulli case. This represents an interesting open direction to explore. The following takeaways from our analysis can be helpful to achieve that goal: (a) $d_\epsilon$ is the information-theoretic quantity that tightly characterises the hardness of bandits with DP, (b) forgetting is not a fundamental design choice, and (c) it is important to have a coupled treatment of the signal and the noise to achieve tight concentration bounds, which are the building block for algorithm design.

## 5  Experimental Analysis

In this section, we numerically compare the performance of our algorithms, *i.e.*, DP-KLUCB and DP-IMED, to $\epsilon$-global DP algorithms from the literature: DP-SE [Sajed and Sheffet, 2019], AdaP-

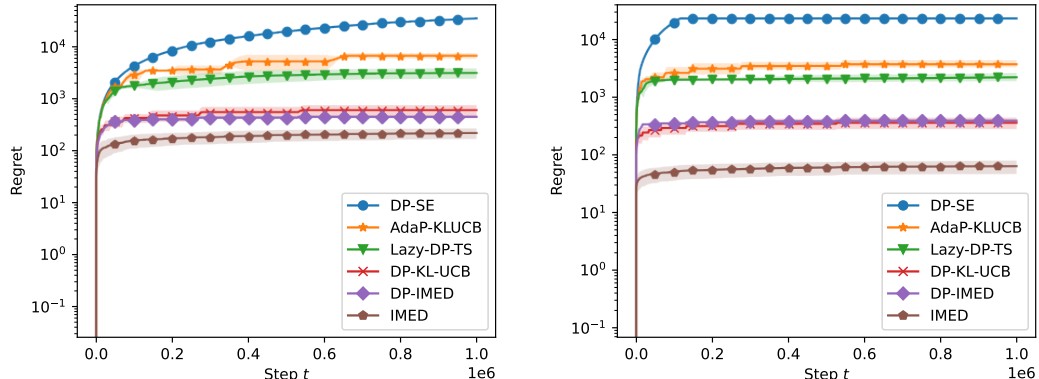

Figure 1: Evolution of the regret (mean $\pm 2$ std) over time for DP-SE, AdaP-KLUCB, Lazy-DP-TS, DP-KLUCB, and DP-IMED for $\epsilon = 0.25$, and Bernoulli bandits $\mu_1$ (left) and $\mu_2$ (right).

KLUCB [Azize and Basu, 2022] and Lazy-DP-TS [Hu and Hegde, 2022]. As a non-private benchmark, we include the IMED algorithm [Honda and Takemura, 2015]. Since both AdaP-KLUCB and Lazy-DP-TS explore each arm once, and use arm-dependent *doubling*, we chose $n_0 = 1$ and $\alpha = 2$ for DP-KLUCB and DP-IMED. Also, to comply with the regret analysis in [Azize and Basu, 2022, Sajed and Sheffet, 2019], we chose $\alpha = 3.1$ in AdaP-KLUCB, and $\beta = 1/T$ in DP-SE.

**Setup.** As in Sajed and Sheffet [2019], Azize and Basu [2022], Hu and Hegde [2022], we consider 4 different 5-arm Bernoulli environments, with specific arm-means choices. We run each algorithm 100 times for $T = 10^6$. For $\epsilon = 0.25$, we plot the mean regret in Figure 1 for $\mu_1 \triangleq [0.75, 0.7, 0.7, 0.7, 0.7]$ in the left and $\mu_2 \triangleq [0.75, 0.625, 0.5, 0.375, 0.25]$ in the right. In Appendix G, we present additional results for some other environments under different budgets.

**Results.** DP-KLUCB *and* DP-IMED *achieve lower regret for all Bernoulli environments and privacy budgets* under study (*up to 10 times less on an average*). This is explained by the fact that DP-KLUCB and DP-IMED do not forget half of the samples, and also thanks to their tighter $d_\epsilon$-based indexes.

## 6 Discussions and Future Works

We improve both regret lower bound (Theorem 1) and upper bounds (Theorem 2) for Bernoulli bandits under $\epsilon$-global DP. We introduce a new information-theoretic quantity $d_\epsilon$ (Equation (6)) that tightly characterises the hardness of minimising regret under DP, and smoothly interpolates between the KL and the TV. Our proposed algorithms share ingredients with algorithms from the literature while alleviating the need to forget rewards as a design technique. This is thanks to a new tighter concentration inequality for private means of Bernoullis (Proposition 1). Our results solve the open problem of having matching upper and lower bound up to the same constant posed by Azize and Basu [2022] and refute that forgetting is necessary for designing optimal DP bandit algorithms.

An interesting future work would be to generalise our concentration inequality, and in turn, the regret upper bounds to general distribution families (e.g. sub-Gaussians, exponential families).

## Acknowledgments and Disclosure of Funding

A. Azize thanks the support of the FairPlay Joint Team and THIA ANR program "AI_PhD@Lille". J. Honda was supported by JSPS KAKENHI Grant Number JP25K03184. A. Azize, J. Honda, and D. Basu acknowledge the Inria-Kyoto University Associate Team "RELIANT" for supporting the project. D. Basu acknowledges the supports of ANR JCJC project REPUBLIC (ANR-22-CE23-0003-01) and PEPR project FOUNDRY (ANR23-PEIA-0003).

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

## A   Outline

The appendices are organised as follows:

- In Appendix B, we extend the adaptive continual release model of Jain et al. [2023] to bandits, and link it to $\epsilon$-global DP.

- In Appendix C, we provide the proof of the three lemmas used to prove the regret lower bound of Theorem 1 and the proof of Theorem 1.

- In Appendix D, we provide the complete proof of the concentration inequality of Proposition 1.

- In Appendix E, we provide the complete proof of the privacy guarantee of Proposition 2.

- In Appendix F, we provide the complete proof of the regret upper bounds of Theorem 2.

- In Appendix G, we provide additional experimental results.

- In Appendix H, we discuss some limitations of our work.

- In Appendix I, we recall useful lemmas used throughout the paper.

## B   Adaptive Continual Release Model for Bandits

In this section, we extend the adaptive continual release model of Jain et al. [2023] to bandits. In this model, the policy interacts with an adversary that chooses adaptively rewards based on previous outputs of the policy.

In the following, we formalise the notion of an adaptive adversary from Jain et al. [2023] and call it a "reward-feeding" adversary.

**Definition 4** (Reward-Feeding Adversary). *A reward-feeding adversary $\mathcal{A}$ is a sequence of functions $(\mathcal{A}_t)_{t=1}^{T}$ such that, for $t \in \{1, \ldots, T\}$,*

$$\mathcal{A}_t : a_1, \ldots, a_t \to (r_t^L, r_t^R) \,.$$

A "reward-feeding" adversary $\mathcal{A}$ is a sequence of "reward" functions that take as input the action-history and outputs a pair of rewards $(r_t^L, r_t^R)$. The reward-feeding adversary $\mathcal{A}$ has two channels: a left "standard" channel $L$ and a right channel $R$. These channels are used to simulate "neighbouring" rewards.

Precisely, to simulate "neighbouring" rewards, the interactive protocol between the policy $\pi$ and the reward-feeding adversary $\mathcal{A}$ has two hyper-parameters: (a) a specific "challenge" time $t^\star \in \{1, T\}$, and (b) a binary $b \in \{L, R\}$. For steps $t \neq t^\star$, the policy observes a reward coming from the adversary's left "standard" channel, i.e. $r_t = r_t^L$. Otherwise, when $t = t^\star$, the policy observes a reward from the channel corresponding to the secret binary $b$.

In other words, if $b = L$, the policy $\pi$ always observes a reward from the left channel. When $b = R$, the policy observes the left channel reward for all steps, except at $t^\star$ where the policy observes a right channel reward. Thus, for any sequence of actions $(a_1, \ldots, a_T)$ chosen by the policy $\pi$, and for any $t^\star$, the sequence of rewards observed by $\pi$ when $b = L$ is neighbouring to the sequence of rewards observed when $b = R$. In addition, these two sequences only differ at the reward observed at the challenge time $t^\star$, and the rewards have been adaptively chosen by the adversary.

Thus, we formalise the adaptive continual release interaction as follows:

Let $b \in \{L, R\}$ and $t^\star \in \{1, \ldots, T\}$

For $t = 1, \ldots, T$

    1. The policy $\pi$ selects an action

$$a_t \sim \pi_t(\cdot \mid a_1, r_1, \ldots, a_{t-1}, r_{t-1}), \, a_t \in [K]$$

    2. The adversary $\mathcal{A}$ selects an adaptively chosen pair of rewards:

$$(r_t^L, r_t^R) = \mathcal{A}_t(a_1, \ldots, a_t)$$

        • If $t \neq t^\star$:

$$r_t = r_t^L$$

        • If $t = t^\star$:

$$r_{t^\star} = r_{t^\star}^b$$

    3. The policy $\pi$ observes the reward $r_t$

When this interaction is run with parameters $t^\star$ and $b$, we represent the interaction by $\pi \overset{b,t^\star}{\Leftrightarrow} \mathcal{A}$, and illustrate it in Figure 2. The view of the adversary $\mathcal{A}$ in the interaction $\pi \overset{b,t^\star}{\Leftrightarrow} \mathcal{A}$ is the sequence of actions chosen by the policy $\pi$, *i.e.*,

$$\text{View}_{\mathcal{A},\pi}^{b,t^\star} \triangleq \text{View}_{\mathcal{A}}(\pi \overset{b,t^\star}{\Leftrightarrow} \mathcal{A}) \triangleq (a_1, \ldots, a_T) \, .$$

A policy is DP in the adaptive continual release model if the view of the adversary is indistinguishable when the interaction is run on $b = L$ and $b = R$ for any challenge step $t^\star$.

**Definition 5** (DP in the Adaptive Continual Release Model).

- *A policy $\pi$ is $(\epsilon, \delta)$-DP in the adaptive continual release model for a given $\epsilon \geq 0$ and $\delta \in [0, 1)$, if for all reward-feeding adversaries $\mathcal{A}$, all subset of views $\mathcal{S} \subseteq [K]^T$,*

$$\sup_{t^\star \in \{1,\ldots,T\}} \Pr[\text{View}_{\mathcal{A},\pi}^{L,t^\star} \in \mathcal{S}] - e^\epsilon \Pr[\text{View}_{\mathcal{A},\pi}^{R,t^\star} \in \mathcal{S}] \leq \delta \, .$$

- *A policy $\pi$ is $\rho$-zCDP in the adaptive continual release model for a given $\rho \geq 0$, if for every $\alpha > 1$, and every reward-feeding adversary $\mathcal{A}$,*

$$\sup_{t^\star \in \{1,\ldots,T\}} D_\alpha(\text{View}_{\mathcal{A},\pi}^{L,t^\star} \| \text{View}_{\mathcal{A},\pi}^{R,t^\star}) \leq \rho\alpha \, .$$

**Remark 1.** *[Expanding the View of the Reward-feeding Adversary $\mathcal{A}$] For any reward-feeding adversary $\mathcal{A}$, any policy $\pi$ and any $t^\star \in \{1, \ldots, T\}$, and any $(a_1, \ldots, a_T) \in [K]^T$, we have for the left view:*

$$\Pr[\text{View}_{\mathcal{A},\pi}^{L,t^\star} = (a_1, \ldots, a_T)] = \pi_1(a_1)\pi_2(a_2 \mid a_1, \mathcal{A}_1^L(a_1)) \cdots \times$$

$$\pi_T(a_T \mid a_1, \mathcal{A}_1^L(a_1), \ldots, a_{T-1}, \mathcal{A}_{T-1}^L(a_1, \ldots, a_{T-1})) \, .$$

*On the other hand, for the right view:*

$$\Pr[\text{View}_{\mathcal{A},\pi}^{R,t^\star} = (a_1, \ldots, a_T)] = \pi_1(a_1)\pi_2(a_2 \mid a_1, \mathcal{A}_1^L(a_1)) \cdots \times$$

$$\pi_{t^\star+1}(a_{t^\star+1} \mid a_1, \mathcal{A}_1^L(a_1), \ldots, a_{t^\star}, \mathcal{A}_{t^\star}^R(a_1, \ldots, a_{t^\star})) \cdots \times$$

$$\pi_T(a_T \mid a_1, \mathcal{A}_t^L(a_1), \ldots, a_{T-1}, \mathcal{A}_{T-1}^L(a_1, \ldots, a_{t-1})) \, .$$

*Let us define*

$$\mathcal{A}^{L,t^\star}(a_1, \ldots, a_T) \triangleq (\mathcal{A}_1^L(a_1), \mathcal{A}_2^L(a_1, a_2), \ldots, \mathcal{A}_T^L(a_1, \ldots, a_T))$$

*to be the list of rewards that the policy observes when the protocol is run on the left channel. Also,*

$$\mathcal{A}^{R,t^\star}(a_1, \ldots, a_T) \triangleq (\mathcal{A}_1^L(a_1), \ldots, \mathcal{A}_{t^\star}^R(a_1, \ldots, a_{t^\star}) \ldots \mathcal{A}_T^L(a_1, \ldots, a_T))$$

*is the list of rewards that the policy observes when the protocol is run on the right channel and $t^\star$.*

*We observe that, for any $(a_1, \ldots, a_T) \in [K]^T$,*

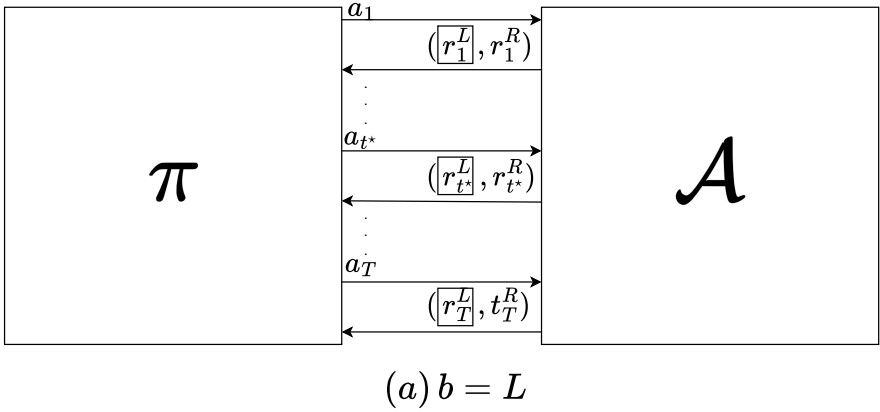

$$(a) \; b = L$$

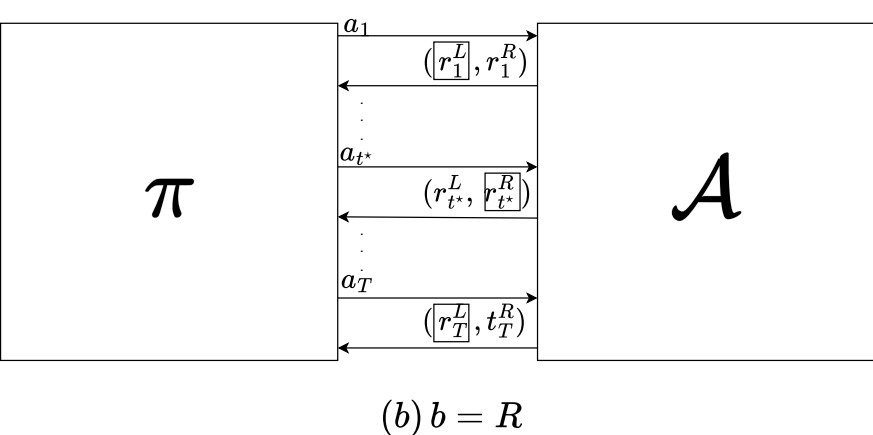

$$(b) \; b = R$$

Figure 2: Interactive protocol in the adaptive continual release model between a policy $\pi$ and a reward-feeding adversary $\mathcal{A}$. The protocol in Figure (a) is run with $b = L$, while the protocol in Figure (b) is run with $b = L$. The framed part corresponds to the reward observed by the policy.

(a) $\Pr[\text{View}_{\mathcal{A},\pi}^{L,t^\star} = (a_1, \ldots, a_T)] = \mathcal{V}^\pi((a_1, \ldots, a_T) \mid \mathcal{A}^{L,t^\star}(a_1, \ldots, a_T))$.

(b) $\Pr[\text{View}_{\mathcal{A},\pi}^{R,t^\star} = (a_1, \ldots, a_T)] = \mathcal{V}^\pi((a_1, \ldots, a_T) \mid \mathcal{A}^{R,t^\star}(a_1, \ldots, a_T))$.

(c) $\mathcal{A}^{L,t^\star}(a_1, \ldots, a_T)$ and $\mathcal{A}^{R,t^\star}(a_1, \ldots, a_T)$ *are neighbouring lists of rewards, and only differ at the $t^\star$-th element.*

*This remark will help connect the adaptive continual release model with View DP later.*

**Remark 2.** *[Reward-feeding Adversary as a Tree Reward Input] A reward-feeding adversary can be represented by a tree of rewards. Each node in the tree corresponds to a reward input. The tree has a depth of size $T$. At depth $t \in [T]$ of the tree reside all possible rewards the policy can observe at step $t$. Going from depth $t$ to depth $t + 1$ depends on the action $a_{t+1}$. Finally, the policy only observes the reward corresponding to its trajectory in the tree. An example of the tree is presented in Figure 3.c for $T = 3$ and $K = 2$.*

*A policy $\pi$ is DP in the adaptive continual release model if and only if $\pi$ is DP when interacting with two neighbouring trees of rewards. Two trees of rewards are neighbouring if they only differ in rewards at one depth $t^\star \in [T]$.*

Now, we relate DP in the adaptive continual release model with View DP and Table DP.

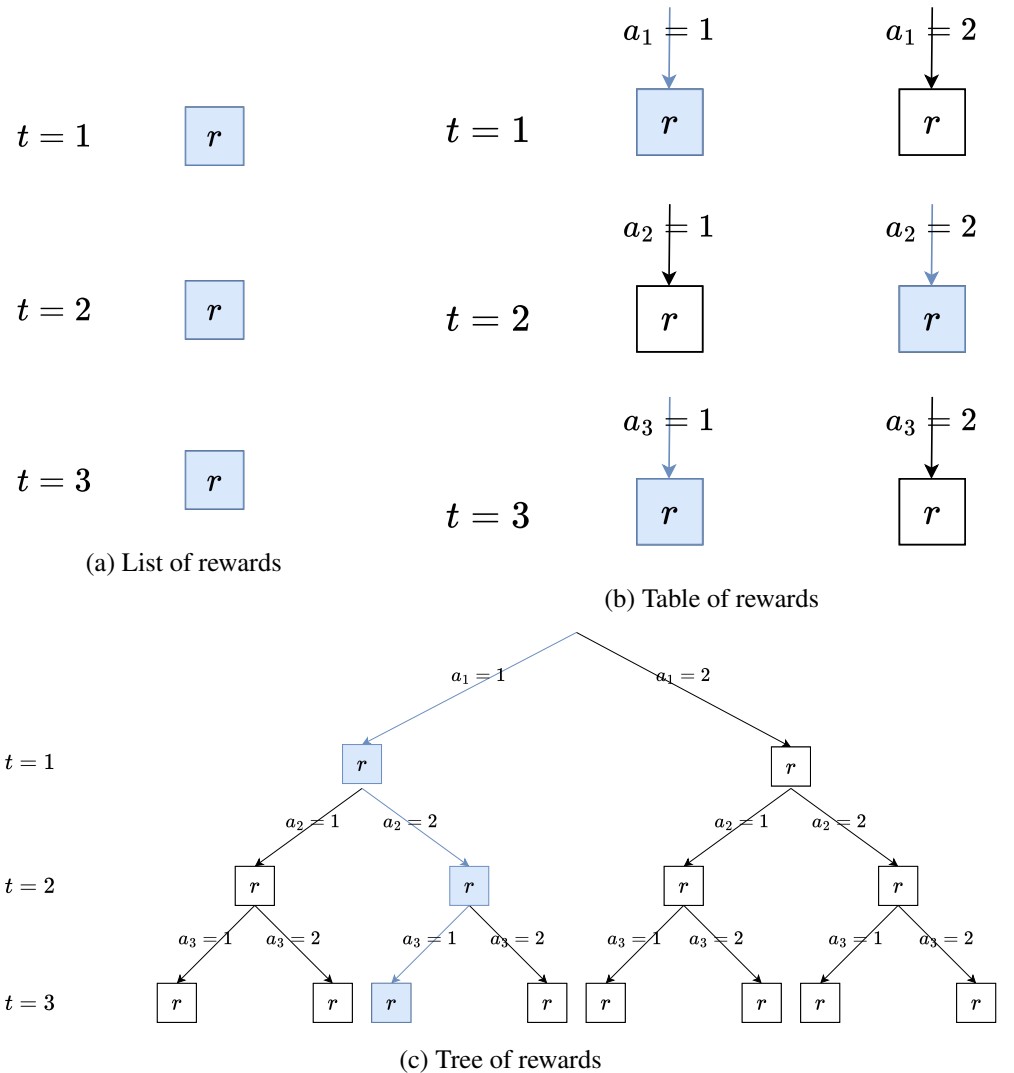

(a) List of rewards

(b) Table of rewards

(c) Tree of rewards

Figure 3: Different reward representations for $T = 3$ and $K = 2$. The highlighted rewards are the rewards observed by the policy for the trajectory $(a_1, a_2, a_3) = (1, 2, 1)$

**Proposition 3** (Link between the Adaptive Continual Release Model, View DP, and Table DP). *For any policy $\pi$, we have that*

> *(a) $\pi$ is DP in the adaptive continual release model $\Rightarrow \pi$ is Table DP.*

> *(b) $\pi$ is $\epsilon$-DP in the adaptive continual release model $\Leftrightarrow \pi$ is $\epsilon$-Table DP $\Leftrightarrow \pi$ is $\epsilon$-View DP.*

Proposition 3 shows that the adaptive continual release model is stronger than Table DP. For pure $\epsilon$-DP, the adaptive continual release model, Table DP and View DP are all equivalent.

To prove this proposition, we use the following reduction.

**Reduction 1** (From table of rewards to "reward-feeding" adversaries). *For a pair of reward tables $\boldsymbol{x}, \boldsymbol{x'} \in (\mathbb{R}^K)^T$, we define $\mathcal{A}(\boldsymbol{x}, \boldsymbol{x'})$ to be the "reward-feeding" adversary defined by*

$$\mathcal{A}(\boldsymbol{x}, \boldsymbol{x'})_t : a_1, \ldots, a_t \rightarrow (x_{t,a_t}, x'_{t,a_t}) .$$

*In other words, at step $t$, the adversary $\mathcal{A}(\boldsymbol{x}, \boldsymbol{x'})$ only uses the last action $a_t$ and returns the $a_t$-th column from $x_t$ on the left channel, and the $a_t$-th column from $x'_t$ on the right channel.*

*For neighbouring tables **x** and **x'** which only differ at some step $t^\star$, it is possible to show that, for every $S \in \mathbb{R}^T$, we have*

- $\Pr[\text{View}^{L,t^\star}_{\mathcal{A}(\mathbf{x},\mathbf{x'}),\pi} \in \mathcal{S}] = \mathcal{M}^\pi_{\mathbf{x}}(S).$

- $\Pr[\text{View}^{R,t^\star}_{\mathcal{A}(\mathbf{x},\mathbf{x'}),\pi} \in \mathcal{S}] = \mathcal{M}^\pi_{\mathbf{x'}}(S).$

*In other words, the batch mechanism $\mathcal{M}^\pi$ combined with neighbouring tables can be "simulated" using a specific type of "reward-feeding" adversaries that only care about the last action from the history.*

*Proof.* (a) Suppose that $\pi$ is DP in the adaptive continual release model.

Let $t^\star \in [T]$, and $x \sim x'$ be two tables of rewards in $(\mathbb{R}^K)^T$ that only differ at step $t^\star$. Using Reduction 1, we build $\mathcal{A}(x,x')$.

For this construction, we have that $\mathcal{M}^\pi_x = \text{View}^{L,t^\star}_{\mathcal{A}(\mathbf{x},\mathbf{x'}),\pi}$ and $\mathcal{M}^\pi_{x'} = \text{View}^{R,t^\star}_{\mathcal{A}(\mathbf{x},\mathbf{x'}),\pi}$.

Since $\pi$ is DP in the adaptive continual release model, $\text{View}^{L,t^\star}_{\mathcal{A}(\mathbf{x},\mathbf{x'}),\pi}$ and $\text{View}^{L,t^\star}_{\mathcal{A}(\mathbf{x},\mathbf{x'}),\pi}$ are indistinguishable. Thus, $\mathcal{M}^\pi_x$ and $\mathcal{M}^\pi_{x'}$ are indistinguishable, *i.e.*, $\mathcal{M}^\pi$ is DP and $\pi$ is Table DP.

(b) To prove this part, it is enough to show that $\epsilon$-View DP implies $\epsilon$-DP in the adaptive continual release model.

Suppose that $\pi$ is $\epsilon$-View DP, *i.e.* $\mathcal{V}^\pi$ is $\epsilon$-DP. Let $\mathcal{A}$ be a "reward-feeding" adversary, and $(a_1, \ldots, a_T) \in [K]^T$ a sequence of arms.

Using Remark 1 and the notation defined there, we have

$$\Pr[\text{View}^{L,t^\star}_{\mathcal{A},\pi} = (a_1, \ldots, a_T)] = \mathcal{V}^\pi((a_1, \ldots, a_T) \mid \mathcal{A}^{L,t^\star}(a_1, \ldots, a_T))$$
$$\leq e^\epsilon \mathcal{V}^\pi((a_1, \ldots, a_T) \mid \mathcal{A}^{R,t^\star}(a_1, \ldots, a_T))$$
$$= e^\epsilon \Pr[\text{View}^{L,t^\star}_{\mathcal{A},\pi} = (a_1, \ldots, a_T)],$$

where the inequality holds because $\mathcal{V}^\pi$ is DP, and $\mathcal{A}^{L,t^\star}(a_1, \ldots, a_T)$ and $\mathcal{A}^{R,t^\star}(a_1, \ldots, a_T)$ are neighbouring lists of rewards.

Finally, this means that $\pi$ is $\epsilon$-DP in the adaptive continual release model, since for pure DP, it is enough to check the atomic events $(a_1, \ldots, a_T)$.

Note that the proof breaks if we consider composite events, which are necessary for approximate DP proofs. □

**Summary of the relationship between definitions.** We introduced three increasingly stronger input representations and their corresponding DP definitions: list of rewards with View DP, table of rewards with Table DP, and tree of rewards with DP in the adaptive continual release. These representations are summarised in Figure 3 for $T = 3$ and $K = 2$.

In general, DP in the adaptive continual release is stronger than Table DP, which is stronger than View DP. For $\epsilon$-pure DP, these three definitions are equivalent, with the same privacy budget $\epsilon$. More care is needed for other variants of DP, where going from one definition to another happens with a loss in the privacy budgets (Proposition 1 in Azize and Basu [2022]).

## C  Lower Bound Proof

In this section, we present the proof of the three main lemma used to prove Theorem 1. We adopt the same notation introduced in the proof of Theorem 1.

**Lemma 1** (Controlling $\mathbb{P}_{\gamma\pi}(\Omega \cap L \cap A)$, aka Double change of environment). *We show that*

$$\mathbb{P}_{\gamma\pi}(\Omega \cap L \cap A) \leq e^{(1+\alpha)n_2\left(\epsilon\text{TV}(P_2,P_2') + \text{kl}(\mu_2', \mu_2'')\right)} \frac{O(T^a)}{T - n_2}, \tag{13}$$

*for any $a > 0$.*

*Proof.* We have

$$\mathbb{P}_{\gamma\pi}\left(\Omega \cap L \cap A\right)$$

$$= \sum_{\mathbf{a}} \int_{\mathbf{r}} \int_{\mathbf{r'}} \mathbb{1}(\Omega \cap L \cap A) \prod_{t=1}^{T} \pi_t(a_t \mid a_1, r_1, \ldots, a_{t-1}, r_{t-1}) c_{a_t}(r_t, r'_t) \, \mathrm{d}r_t \, \mathrm{d}r'_t$$

$$\overset{(a)}{\leq} \sum_{\mathbf{a}} \int_{\mathbf{r}} \int_{\mathbf{r'}} \mathbb{1}(\Omega \cap L \cap A) e^{\epsilon \mathrm{dham}(r, r')} \prod_{t=1}^{T} \pi_t(a_t \mid a_1, r'_1, \ldots, a_{t-1}, r'_{t-1}) c_{a_t}(r_t, r'_t) \, \mathrm{d}r_t \, \mathrm{d}r'_t$$

$$\overset{(b)}{\leq} e^{\epsilon(1+\alpha)n_2 \mathrm{TV}(P_2, P'_2)} \sum_{\mathbf{a}} \int_{\mathbf{r}} \int_{\mathbf{r'}} \mathbb{1}(\Omega \cap L \cap A) \prod_{t=1}^{T} \pi_t(a_t \mid a_1, r'_1, \ldots, a_{t-1}, r'_{t-1}) c_{a_t}(r_t, r'_t) \, \mathrm{d}r_t \, \mathrm{d}r'_t$$

$$\overset{(c)}{\leq} e^{\epsilon(1+\alpha)n_2 \mathrm{TV}(P_2, P'_2)} \sum_{\mathbf{a}} \int_{\mathbf{r}} \int_{\mathbf{r'}} \mathbb{1}(\Omega \cap A) \prod_{t=1}^{T} \pi_t(a_t \mid a_1, r'_1, \ldots, a_{t-1}, r'_{t-1}) c_{a_t}(r_t, r'_t) \, \mathrm{d}r_t \, \mathrm{d}r'_t$$

$$\overset{(d)}{=} e^{\epsilon(1+\alpha)n_2 \mathrm{TV}(P_2, P'_2)} \sum_{\mathbf{a}} \int_{\mathbf{r'}} \mathbb{1}(\Omega \cap A) \prod_{t=1}^{T} \pi_t(a_t \mid a_1, r'_1, \ldots, a_{t-1}, r'_{t-1}) p'_{a_t}(r'_t) \, \mathrm{d}r'_t$$

$$= e^{\epsilon(1+\alpha)n_2 \mathrm{TV}(P_2, P'_2)} \sum_{\mathbf{a}} \int_{\mathbf{r'}} \mathbb{1}(\Omega \cap A) e^{\sum_{t=1}^{T} \log \frac{\mathrm{d}P'_{a_t}(r'_t)}{\mathrm{d}P''_{a_t}(r'_t)}} \prod_{t=1}^{T} \pi_t(a_t \mid a_1, r'_1, \ldots, a_{t-1}, r'_{t-1}) p''_{a_t}(r'_t) \, \mathrm{d}r'_t$$

$$\overset{(e)}{\leq} e^{\epsilon(1+\alpha)n_2 \mathrm{TV}(P_2, P'_2)} e^{(1+\alpha)\mathrm{kl}(\mu'_2, \mu''_2)n_2} \sum_{\mathbf{a}} \int_{\mathbf{r'}} \mathbb{1}(\Omega) \prod_{t=1}^{T} \pi_t(a_t \mid a_1, r'_1, \ldots, a_{t-1}, r'_{t-1}) p''_{a_t}(r'_t) \, \mathrm{d}r'_t$$

$$= e^{(1+\alpha)n_2\left(\epsilon \mathrm{TV}(P_2, P'_2) + \mathrm{kl}(\mu'_2, \mu''_2)\right)} \mathbb{P}_{\nu''\pi}\left(N_2(T) \leq n_2\right),$$

where:

(a) is because $\pi$ is $\epsilon$-DP;

(b) is by definition of $L$;

(c) is because $\mathbb{1}(\Omega \cap L \cap A) \leq \mathbb{1}(\Omega \cap A)$;

(d) by definition of the coupling, and because $\Omega \cap A$ doesn't depend on $(r_t)_{t=1}^{T}$;

(e) by definition of $A$.

Then, using Markov inequality and the consistency of $\pi$, we get

$$\begin{aligned}
\mathbb{P}_{\nu''\pi}\left(N_2(T) \leq n_2\right) &= \mathbb{P}_{\nu''\pi}\left(T - N_2(T) \geq T - n_2\right) \\
&= \mathbb{P}_{\nu''\pi}\left(N_1(T) \geq T - n_2\right) \\
&\leq \frac{\mathbb{E}_{\nu''\pi}(N_1(T))}{T - n_2} = \frac{O(T^\alpha)}{T - n_2},
\end{aligned}$$

for any $a > 0$, since arm 1 is sub-optimal in environment $\nu''$ and $\pi$ is consistent.

All in all, we have that, for any $a > 0$,

$$\mathbb{P}_{\gamma\pi}\left(\Omega \cap L \cap A\right) \leq e^{(1+\alpha)n_2\left(\epsilon \mathrm{TV}(P_2, P'_2) + \mathrm{kl}(\mu'_2, \mu''_2)\right)} \frac{O(T^a)}{T - n_2}.$$

$\square$

**Lemma 2** (Controlling $\mathbb{P}_{\gamma\pi}(\Omega \cap L \cap A^c)$). *Choosing $n_2 = n_2(T)$ a function such that $n_2(T) \to \infty$ when $T \to \infty$, then*

$$\mathbb{P}_{\gamma\pi}(\Omega \cap L \cap A^c) = o_T(1),$$

*asymptotically in $T$.*

*Proof.* First, we have

$$\mathbb{P}_{\gamma\pi}\left(\Omega \cap L \cap A^c\right) \leq \mathbb{P}_{\gamma\pi}\left(\Omega \cap A^c\right).$$

Let us introduce the notation $r'_{a,s} \triangleq r'_{\tau_{a,s}}$ where $\tau_{a,s} \triangleq \min\{t \in \mathbb{N} : N_a(t) = s\}$. Then,

$$\sum_{t=1}^{T} \log \frac{\mathrm{d}P'_{a_t}(r'_t)}{\mathrm{d}P''_{a_t}(r'_t)} = \sum_{s=1}^{N_2(T)} \log \frac{\mathrm{d}P'_2(r'_{2,s})}{\mathrm{d}P''_2(r'_{2,s})} = \sum_{s=1}^{N_2(T)} W_s,$$

where $W_s \triangleq \log \frac{\mathrm{d}P'_2(r'_{2,s})}{\mathrm{d}P''_2(r'_{2,s})}$ are i.i.d bounded random variables, with positive mean $\mathbb{E}_{\gamma\pi}[W_s] = \mathrm{kl}(\mu'_2, \mu''_2)$. This is true since under the coupling $\gamma$, the marginal of $r'_{2,s}$ is $P'_2$.

Then, we get

$$\mathbb{P}_{\gamma\pi}\left(\Omega \cap A^c\right) \le \mathbb{P}_{\gamma\pi}\left(\exists m \le n_2 : \sum_{s=1}^{m} W_s > (1+\alpha)\mathrm{kl}(\mu'_2, \mu''_2)n_2\right)$$

$$\le \mathbb{P}_{\gamma\pi}\left(\frac{\max_{m \le n_2} \sum_{s=1}^{m} W_s}{n_2} > (1+\alpha)\mathrm{kl}(\mu'_2, \mu''_2)\right).$$

Using Asymptotic maximal Hoeffding inequality (Lemma 12), we have that

$$\lim_{n\to\infty} \mathbb{P}_{\gamma\pi}\left(\frac{\max_{m \le n} \sum_{s=1}^{m} W_s}{n} > (1+\alpha)\mathrm{kl}(\mu'_2, \mu''_2)\right) = 0.$$

Thus, by choosing $n_2 = n_2(T)$ a function such that $n_2(T) \to \infty$ when $T \to \infty$, then

$$\mathbb{P}_{\gamma\pi}(\Omega \cap L \cap A^c) = o_T(1),$$

asymptotically in $T$. $\qquad\square$

**Lemma 3** (Controlling $\mathbb{P}_{\gamma\pi}\left(\Omega \cap L^c\right)$). *choosing* $n_2 = n_2(T)$ *a function such that* $n_2(T) \to \infty$ *when* $T \to \infty$, *then*

$$\mathbb{P}_{\gamma\pi}\left(\Omega \cap L^c\right) = o_T(1),$$

*asymptotically in* $T$.

*Proof.* First, by the construction of the couplings, only rewards coming from arm 2 are different, *i.e.*,

$$\mathrm{dham}(r, r') \triangleq \sum_{t=1}^{T} \mathbb{1}(r_t \ne r'_t) = \sum_{t=1}^{T} \mathbb{1}(A_t = 2)\mathbb{1}(r_t \ne r'_t).$$

Let us introduce the notation $r_{a,s} \triangleq r_{\tau_{a,s}}$ where $\tau_{a,s} \triangleq \min\{t \in \mathbb{N} : N_a(t) = s\}$. Then,

$$\mathrm{dham}(r, r') = \sum_{s=1}^{N_2(T)} \mathbb{1}(r_{2,s} \ne r'_{2,s}) = \sum_{s=1}^{N_2(T)} Z_s,$$

where $Z_s \triangleq \mathbb{1}(r_{2,s} \ne r'_{2,s})$ are i.i.d Bernoulli random variables with positive mean $\mathbb{E}_{\gamma\pi}[Z_s] = \mathbb{P}_{\gamma\pi}(r_{2,s} \ne r'_{2,s}) = \mathrm{TV}(P_2, P'_2)$.

$$\mathbb{P}_{\gamma\pi}\left(\Omega \cap L^c\right) \le \mathbb{P}_{\gamma\pi}\left(\exists m \le n_2 : \sum_{s=1}^{m} Z_s > (1+\alpha)n_2\mathrm{TV}(P_2, P'_2)\right)$$

$$\le \mathbb{P}_{\gamma\pi}\left(\frac{\max_{m \le n_2} \sum_{s=1}^{m} Z_s}{n_2} > (1+\alpha)\mathrm{TV}(P_2, P'_2)\right).$$

Using Asymptotic maximal Hoeffding inequality (Lemma 12), we have that

$$\lim_{n\to\infty} \mathbb{P}_{\gamma\pi}\left(\frac{\max_{m \le n} \sum_{s=1}^{m} Z_s}{n} > (1+\alpha)\mathrm{TV}(P_2, P'_2)\right) = 0.$$

Thus, by choosing $n_2 = n_2(T)$ a function such that $n_2(T) \to \infty$ when $T \to \infty$, then

$$\mathbb{P}_{\gamma\pi}\left(\Omega \cap L^c\right) = o_T(1), \tag{14}$$

asymptotically in $T$. $\qquad\square$

### C.1 Complete Proof of Theorem 1

Before providing the proof, we introduce maximal couplings.

**Definition 6** (Maximal Couplings). *Let $\mathbb{P}$ and $\mathbb{Q}$ be two probability distributions that share the same $\sigma$-algebra and $\Pi(\mathbb{P}, \mathbb{Q})$ be the set of all couplings between $\mathbb{P}$ and $\mathbb{Q}$. We denote by $c_\infty(\mathbb{P}, \mathbb{Q})$ the maximal coupling between $\mathbb{P}$ and $\mathbb{Q}$, i.e., the coupling that verifies for any measurable A,*

$$\mathbb{P}_{(X,Y)\sim c_\infty(\mathbb{P},\mathbb{Q})}[X \in A] = \mathbb{P}_{X\sim\mathbb{P}}[X \in A], \mathbb{P}_{(X,Y)\sim c_\infty(\mathbb{P},\mathbb{Q})}[Y \in A] = \mathbb{P}_{Y\sim\mathbb{Q}}[Y \in A],$$

$$\mathbb{P}_{(X,Y)\sim c_\infty(\mathbb{P},\mathbb{Q})}[X \neq Y] = \inf_{c\in\Pi(\mathbb{P},\mathbb{Q})} \mathbb{P}_{(X,Y)\sim c}[X \neq Y] = \text{TV}(\mathbb{P}, \mathbb{Q}) .$$

Finally, we are ready to present now the detailed proof of Theorem 1.

*Proof of Theorem 1.* Without loss of generality, suppose that we have a 2-armed Bernoulli bandit instance $\nu = (P_1, P_2)$ with means $(\mu_1, \mu_2)$ where $\mu_1 \geq \mu_2$. Let $\pi$ be an $\epsilon$-global DP consistent policy. We also introduce *two* other environments $\nu' = (P_1, P_2')$ and $\nu'' = (P_1, P_2'')$ that only differ at the distribution of the second arm, where $\mu_2 \leq \mu_2' \leq \mu_1 \leq \mu_2''$, *i.e.*, arm 1 is still optimal in environment $\nu'$ but is not optimal in environment $\nu''$.

The main idea is to control the probability of the event $\Omega \triangleq \{N_2(T) \leq n_2\}$ in an augmented coupled history space, for some $n_2$ to be fine-tuned later (that may depend on the horizon $T$).

Step 1: Building the coupled bandit environment $\gamma$. We build a coupled bandit environment $\gamma$ of $\nu$ and $\nu'$. The policy $\pi$ interacts with the coupled environment $\gamma$ up to a given time horizon $T$ to produce an augmented history $\{(a_t, r_t, r_t')\}_{t=1}^T$. The steps of this interaction process are: (a) The probability of choosing an action $a_t = a$ at time $t$ is dictated only by the policy $\pi_t$ and $a_1, r_1, a_2, r_2, \ldots, a_{t-1}, r_{t-1}$, *i.e.*, the policy ignores $\{r_s'\}_{s=1}^{t-1}$. (b) The distribution of pair of rewards $(r_t, r_t')$ is $c_{a_t} \triangleq c_\infty(P_{a_t}, P_{a_t}')$ the maximal coupling of $(P_{a_t}, P_{a_t}')$ and is conditionally independent of the previous observed history $\{(a_s, r_s, r_s')\}_{t=1}^{t-1}$. The distribution of the augmented history induced by the interaction of $\pi$ and the coupled environment can be defined as $p_{\gamma\pi}(a_1, r_1, r_1' \ldots, a_T, r_T, r_T') \triangleq \prod_{t=1}^T \pi_t(a_t \mid a_1, r_1, \ldots, a_{t-1}, r_{t-1}) c_{a_t}(r_t, r_t')$.

Again, we introduce the notation $\mathbf{a} \triangleq (a_1, \ldots, a_T)$, $\mathbf{r} \triangleq (r_1, \ldots, r_T)$, and $\mathbf{r'} \triangleq (r_1', \ldots, r_T')$.

Step 2: Probability decomposition. We introduce $L \triangleq \{\text{dham}(\mathbf{r}, \mathbf{r'}) \leq (1+\alpha)n_2\text{TV}(P_2, P_2')\}$, and $A \triangleq \left\{ \sum_{t=1}^T \log \frac{\text{d}P_{a_t}'(r_t')}{\text{d}P_{a_t}''(r_t')} \leq (1+\alpha)\text{kl}(\mu_2', \mu_2'')n_2 \right\}$ for some $\alpha > 0$, where $\text{dham}(\mathbf{r}, \mathbf{r'}) \triangleq \sum_{t=1}^T \mathbb{1}_{r_t \neq r_t'}$. Also, here for Bernoullis, we have $\text{TV}(P_2, P_2') = \mu_2' - \mu_2$.

Event $L$ will be used to do a change of measure from environment $\nu$ to $\nu'$ using the group privacy property of $\pi$, then event $A$ will be used to do a classic "Lai-Robbins" change of measure using the KL from environment $\nu'$ to $\nu''$.

First, we start with the decomposition

$$\mathbb{P}_{\nu\pi}(N_2(T) \leq n_2) = \mathbb{P}_{\gamma\pi}(\Omega \cap L \cap A) + \mathbb{P}_{\gamma\pi}(\Omega \cap L \cap A^c) + \mathbb{P}_{\gamma\pi}(\Omega \cap L^c) . \tag{15}$$

Step 3: Controlling each probability. Using Lemma 1, which formalises the "double" change of environment idea, we get

$$\mathbb{P}_{\gamma\pi}(\Omega \cap L \cap A) \leq e^{(1+\alpha)n_2(\epsilon\text{TV}(P_2, P_2') + \text{kl}(\mu_2', \mu_2''))} \frac{O(T^a)}{T - n_2}, \tag{16}$$

for any $a > 0$. Using Lemma 2 and Lemma 3, we control the probabilities $\mathbb{P}_{\gamma\pi}(\Omega \cap L \cap A^c) = o_T(1)$ and $\mathbb{P}_{\gamma\pi}(\Omega \cap L^c) = o_T(1)$, for any choice of $n_2 = n_2(T)$ as a function of $T$ such that $n_2(T) \to \infty$ when $T \to \infty$.

Step 4: Putting everything together and choosing $n_2$. First, we chose $n_2 = \frac{(1-\alpha)\log(T)}{\epsilon\text{TV}(P_2, P_2') + \text{kl}(\mu_2', \mu_2'')}$, and $a = \frac{\alpha^2}{2}$, to get $\exp\left((1+\alpha)n_2\left(\epsilon\text{TV}(P_2, P_2') + \text{kl}(\mu_2', \mu_2'')\right)\right) \frac{O(T^a)}{T - n_2} = o_T(1)$.

With this choice of $n_2$, we have now that $\mathbb{P}_{\nu\pi}(N_2(T) \leq n_2) = o_T(1)$, and thus, using Markov inequality, we get, for any $\alpha > 0$, and all $\mu_2 \leq \mu_2' \leq \mu_1 \leq \mu_2''$.

$$\mathbb{E}_{\nu\pi}[N_2(T)] \geq n_2\mathbb{P}_{\nu\pi}(N_2(T) > n_2) = \frac{(1-\alpha)\log(T)}{\epsilon\text{TV}(P_2, P_2') + \text{kl}(\mu_2', \mu_2'')}(1 - o(1)) .$$

Finally, taking $\alpha \to 0$, and the supremum over all $\mu_2' \in [\mu_2, \mu_1]$ and $\mu_2'' \to \mu_1$, we get the result. $\qquad \square$

## D   Concentration Inequality Proof

**Lemma 4** (Tail Bound of Cumulative Laplacian Noise). *Let $Z_m = \sum_{l=1}^{m} Y_l$ where $Y_l \sim \mathrm{Lap}(1/\epsilon)$ are i.i.d. Laplace random variables with parameter $1/\epsilon$. Then, for $z > 0$, we have*

$$\mathbb{P}[Z_m \geq z] \leq \exp\left(-f(z)\right),$$

*where $f(z) = \epsilon z - 1 - m \log(1 + m\epsilon z)$.*

*Proof.* For a random variable $Y \sim \mathrm{Lap}(1/\epsilon)$, the probability density function is

$$f_Y(y) = \frac{\epsilon}{2} \exp(-\epsilon|y|).$$

The moment-generating function (MGF) is given by

$$M_Y(t) = \mathbb{E}[\exp(tY)] = \frac{\epsilon^2}{\epsilon^2 - t^2}, \quad |t| < \epsilon.$$

The random variable $Z_m = \sum_{l=1}^{m} Y_l$ is the sum of $m$ i.i.d. Laplace random variables. The MGF of $Z_m$ is the product of the MGFs of the individual $Y_l$:

$$M_{Z_m}(t) = (M_Y(t))^m.$$

Thus, we have

$$M_{Z_m}(t) = \left(\frac{\epsilon^2}{\epsilon^2 - t^2}\right)^m, \quad |t| < \epsilon.$$

To bound $\mathbb{P}[Z_m \geq z]$, we use the Chernoff bound:

$$\begin{aligned}
\mathbb{P}[Z_m \geq z] &\leq \inf_{0<t<\epsilon} \mathbb{E}[\exp(tZ_m - tz)] \\
&= \inf_{0<t<\epsilon} \exp\left(-tz\right) M_{Z_m}(t) \\
&= \inf_{0<t<\epsilon} \exp\left(-tz + m \log\left(\frac{\epsilon^2}{\epsilon^2 - t^2}\right)\right) \\
&= \inf_{0<t<\epsilon} \exp\left(-tz - m \log\left(1 - \frac{t^2}{\epsilon^2}\right)\right).
\end{aligned}$$

Consider

$$f_t(z) = tz + m \log\left(1 - \frac{t^2}{\epsilon^2}\right).$$

Letting $t = \epsilon\sqrt{1-c} \in (0, \epsilon)$ for $c = 1 \wedge 1/(m\epsilon z)$ we have

$$\begin{aligned}
f_t(z) &= \epsilon z \sqrt{1-c} + m \log c \\
&\geq \epsilon z - \epsilon z c + m \log(1 \wedge 1/(m\epsilon z)) \quad \left(\text{by } \sqrt{1-c} \geq 1 - c \text{ for } c \leq 1\right) \\
&= \epsilon z - (\epsilon z \wedge 1/m) + m \log(1 \wedge 1/(m\epsilon z)) \\
&\geq \epsilon z - 1 - m \log(1 \vee m\epsilon z) \\
&\geq \epsilon z - 1 - m \log(1 + m\epsilon z).
\end{aligned}$$

Then, we have

$$f_t(z) \geq \epsilon z - 1 - m \log(1 + m\epsilon z) = f(z),$$

for $z \geq 0$. Thus, we obtain

$$\mathbb{P}[Z_m \geq z] \leq \exp\left(-f(z)\right).$$

$\qquad \square$

**Lemma 5** (Concentration bound of private summation). *For $\mu \in (0,1)$ and $\epsilon > 0$, let*

$$\tilde{S}_{n,m} = \sum_{i=1}^{n} X_i + \sum_{j=1}^{m} Y_j, \qquad X_i \sim \text{Ber}(\mu), Y_j \sim \text{Lap}(1/\epsilon)$$

*be the sum of independent $n$ Bernoulli random variables (RVs) with mean $\mu$ and $m$ Laplace RVs with scale $1/\epsilon$. Then, for $x \geq n\mu$*

$$\Pr\left[\tilde{S}_{n,m} \geq x\right] \leq A_\epsilon(n,m,x,\mu)\text{e}^{-n\text{d}_\epsilon(x/n,\mu)},$$

*where*

$$A_\epsilon(m,n,x,\mu)$$
$$= (x - n\mu) \max_{y \in [\mu, x/n]} \left\{ \text{e}(1 + m\epsilon(x - yn))^m \log\frac{1}{\mu} \right\} + \text{e}(1 + m\epsilon(x - n\mu))^m + 1 .$$

*Similarly, for $x \leq n\mu$,*

$$\Pr\left[\tilde{S}_{n,m} \leq x\right] \leq A_\epsilon(m,n,x,\mu)\text{e}^{-n\text{d}_\epsilon(x/n,\mu)},$$

*where*

$$A_\epsilon(m,n,x,\mu)$$
$$= (n\mu - x) \max_{y \in [x/n, \mu]} \left\{ \text{e}(1 + m\epsilon(yn - x))^m \log\frac{1}{1-\mu} \right\} + \text{e}(1 + m\epsilon(n\mu - x))^m + 1 .$$

*Proof of Lemma 5.* For $\mu \in (0,1)$ and $\epsilon > 0$, the private summation can be written as

$$\tilde{S}_{n,m} = \sum_{i=1}^{n} X_i + \sum_{j=1}^{m} Y_j, \qquad X_i \sim \text{Ber}(\mu), Y_j \sim \text{Lap}(1/\epsilon) . \tag{17}$$

Re-define the non-private summation and the sum of the noise by

$$S_n = \sum_{i=1}^{n} X_i, \quad Z_m = \sum_{j=1}^{m} Y_j \tag{18}$$

and denote density of $Z_m$ by $f_m(z)$. Then, we can upper bound the probability by

$$\Pr\left[\tilde{S}_{n,m} \geq x\right] = \Pr\left[S_n + Z_m \geq x\right]$$
$$= \int_{-\infty}^{\infty} f_m(z) \Pr[S_n \geq x - z]\text{d}z$$
$$= \int_{-\infty}^{0} f_m(z) \Pr[S_n \geq x - z]\text{d}z + \int_{0}^{\infty} f_m(z) \Pr[S_n \geq x - z]\text{d}z$$
$$\leq \int_{-\infty}^{0} f_m(z) \Pr[S_n \geq x]\text{d}z + \int_{0}^{\infty} f_m(z) \Pr[S_n \geq x - z]\text{d}z$$
$$= \underbrace{\frac{1}{2} \Pr[S_n \geq x]}_{\text{(I)}} + \underbrace{\int_{0}^{\infty} f_m(z) \Pr[S_n \geq x - z]\text{d}z}_{\text{(II)}} . \tag{19}$$

Here, $\Pr[S_n \geq x - z]$ can be upper bounded by Chernoff bound. Let $\bar{P}(x - z)$ be such an upper bound. Then, from Lemma 11, we have

$$\bar{P}(x - z) = \text{e}^{-n \cdot \text{kl}((x-z)/n, \mu)}, \quad \text{for} \quad x - z \geq n\mu . \tag{20}$$

Based on this upper bound, we can bound the second term in (19):

$$\text{(II)} = \int_{0}^{\infty} f_m(z) \Pr[S_n \geq x - z]\text{d}z$$

$$\leq \int_0^\infty f_m(z)\bar{P}(x-z)\mathrm{d}z$$

$$= [-F_m(z)\bar{P}(x-z)]_0^\infty + \int_0^\infty F_m(z)(-\bar{P}'(x-z))\mathrm{d}z \quad \text{(integration by parts)}$$

$$= F_m(0)\bar{P}(x) + \int_0^\infty F_m(z)(-\bar{P}'(x-z))\mathrm{d}z$$

$$= \frac{1}{2}\bar{P}(x) + \int_0^\infty F_m(z)(-\bar{P}'(x-z))\mathrm{d}z, \tag{21}$$

where $F_m(z) = \int_z^\infty f_m(z)\mathrm{d}z = \Pr[Z_m \geq z]$ is the (complement) cumulative distribution. From Lemma 4, we have

$$F_m(z) = \Pr[Z_m \geq z] \leq \exp\left(-f(z)\right),$$

where $f(z) = \epsilon z - 1 - m\log(1 + m\epsilon z)$. Thus, we can bound the second term in (21):

$$\int_0^\infty F_m(z)(-\bar{P}'(x-z))\mathrm{d}z$$

$$= \int_0^{x-n\mu} F_m(z)(-\bar{P}'(x-z))\mathrm{d}z + \int_{x-n\mu}^\infty F_m(z)(-\bar{P}'(x-z))\mathrm{d}z \quad (F_m(z) \text{ is decreasing})$$

$$\leq \int_0^{x-n\mu} F_m(z)(-\bar{P}'(x-z))\mathrm{d}z + F_m(x-n\mu)\int_{x-n\mu}^\infty (-\bar{P}'(x-z))\mathrm{d}z$$

$$= \int_0^{x-n\mu} F_m(z)(-\bar{P}'(x-z))\mathrm{d}z + F_m(x-n\mu)\bar{P}(n\mu)$$

$$\leq \int_0^{x-n\mu} \mathrm{e}^{-f(z)}(-\bar{P}'(x-z))\mathrm{d}z + \mathrm{e}^{-f(x-n\mu)} \cdot 1 . \tag{22}$$

We now focus on bounding the first term in RHS of the last inequality. Observe that

$$-\bar{P}'(z) = \mathrm{kl}'(z/n, \mu)\mathrm{e}^{-n \cdot \mathrm{kl}(z/n,\mu)}, \tag{23}$$

where $\mathrm{kl}'(x,y) = \frac{\partial \mathrm{kl}(x,y)}{\partial x}$ is the derivative with respect to the first argument. Then, for $x - z \geq n\mu$, we have

$$\int_0^{x-n\mu} \mathrm{e}^{-f(z)}(-\bar{P}'(x-z))\mathrm{d}z$$

$$= \int_0^{x-n\mu} \mathrm{e}(1 + m\epsilon z)^m \mathrm{kl}'((x-z)/n, \mu)\mathrm{e}^{-\epsilon z}\mathrm{e}^{-n \cdot \mathrm{kl}((x-z)/n,\mu)}\mathrm{d}z$$

$$= \int_\mu^{x/n} n\mathrm{e}(1 + m\epsilon(x-yn))^m \mathrm{kl}'(y, \mu)\mathrm{e}^{-\epsilon(x-yn)}\mathrm{e}^{-n \cdot \mathrm{kl}(y,\mu)}\mathrm{d}y \quad (\text{ let } y := (x-z)/n)$$

$$\leq \mathrm{e}^{-\inf_{y\in[\mu,x/n]}\{\epsilon(x-yn)+n \cdot \mathrm{kl}(y,\mu)\}} \int_\mu^{x/n} n\mathrm{e}(1 + m\epsilon(x-yn))^m \mathrm{kl}'(y, \mu)\mathrm{d}y$$

$$\leq \mathrm{e}^{-n \cdot \mathrm{d}_\epsilon(x/n,\mu)} \int_\mu^{x/n} n\mathrm{e}(1 + m\epsilon(x-\mu n))^m \mathrm{kl}'(y, \mu)\mathrm{d}y$$

$$= \mathrm{e}^{-n \cdot \mathrm{d}_\epsilon(x/n,\mu)} n\mathrm{e}(1 + m\epsilon(x-\mu n))^m \mathrm{kl}(x/n, \mu)$$

$$\leq \mathrm{e}^{-n \cdot \mathrm{d}_\epsilon(x/n,\mu)} n\mathrm{e}(1 + m\epsilon(x-\mu n))^m \mathrm{kl}(1, \mu)$$

$$= \mathrm{e}^{-n \cdot \mathrm{d}_\epsilon(x/n,\mu)} n\mathrm{e}(1 + m\epsilon(x-\mu n))^m \log \frac{1}{\mu} \tag{24}$$

where $\mathrm{d}_\epsilon(x/n, \mu)$ is defined in (6). Now, we bound the second term in (22):

$$\mathrm{e}^{-f(x-n\mu)} = \mathrm{e}(1 + m\epsilon(x-n\mu))^m \mathrm{e}^{-n\epsilon(x/n-\mu)}$$

$$\leq \mathrm{e}(1 + m\epsilon(x-n\mu))^m \mathrm{e}^{-n\mathrm{d}_\epsilon(x/n,\mu)} . \tag{25}$$

Note that we have for $x \geq n\mu$

$$\Pr[S_n \geq x] \leq \bar{P}(x) \leq e^{-n\mathrm{kl}(x/n,\mu)} \quad \text{(by Lemma 11)}$$
$$\leq e^{-n\mathrm{d}_\epsilon(x/n,\mu)} . \tag{26}$$

Putting (24), (25), and (26) together we have for $x/n \geq \mu$

$$\Pr\left[\tilde{S}_{n,m} \geq x\right] \leq A_\epsilon(m,n,x,\mu)e^{-n\cdot\mathrm{d}_\epsilon(x/n,\mu)}$$

where

$$A_\epsilon(m,n,x,\mu) = (x-n\mu)\max_{y\in[\mu,x/n]}\left\{e(1+m\epsilon(x-yn))^m\log\frac{1}{\mu}\right\} + e(1+m\epsilon(x-n\mu))^m + 1 .$$

Similarly, we can get for $x/n \leq \mu$

$$\Pr\left[\tilde{S}_{n,m} \leq x\right] \leq A_\epsilon(m,n,x,\mu)e^{-n\mathrm{d}_\epsilon(x/n,\mu)},$$

where

$$A_\epsilon(m,n,x,\mu)$$
$$= (x-n\mu)\max_{y\in[\mu,x/n]}\left\{e(1+m\epsilon(x-yn))^m\log\frac{1}{1-\mu}\right\} + e(1+m\epsilon(x-n\mu))^m + 1 .$$

$\square$

**Corollary 1** (Concentration bound of private mean). *Consider $\tilde{S}_{n,m}$ given in Lemma 5. Let $x \in [0,1]$. Let $\{n_m\}_{m\in\mathbb{N}}$ be a sequence such that $m/n_m = o(1)$. Then, for any $a > 0$ there exists a constant $A_a > 0$ such that for all $m \in \mathbb{N}$*

$$\Pr\left[\frac{\tilde{S}_{n_m,m}}{n_m} \geq x\right] \leq A_a e^{-n_m(\mathrm{d}_\epsilon(x,\mu)-a)}, \qquad x \geq \mu .$$
$$\Pr\left[\frac{\tilde{S}_{n_m,m}}{n_m} \leq x\right] \leq A_a e^{-n_m(\mathrm{d}_\epsilon(x,\mu)-a)}, \qquad x \leq \mu.$$

*Proof of Corollary 1.* From Lemma 5, we have for $x \geq \mu$

$$A_\epsilon(m,n_m,x,\mu)$$
$$= n_m(x-\mu)\max_{y\in[\mu,x]}\left\{e(1+(m+1)\epsilon n_m(x-y))^{m+1}\log\frac{1}{\mu}\right\} + e(1+(m+1)\epsilon n_m(x-\mu))^{m+1} + 1 . \tag{27}$$

For $y \in [\mu, x]$,

$$A_\epsilon(m,n_m,x,\mu) \leq A(n_m) = n_m e(1+(m+1)\epsilon n_m)^{m+1}\log\frac{1}{\mu} + e(1+(m+1)\epsilon n_m)^{m+1} + 1 .$$

Since existing $b$ to make $1 + x \leq be^x$ hold, we have the result. The proof for the case of $x \leq \mu$ is completely analogous.

$\square$

# E   Privacy Analysis

First, we provide a simple lemma to motivate the intuition behind the algorithm design. Then, we provide a complete proof of Proposition 2.

**Lemma 6** (Continual release of noisy rewards). *Let rewards $\{r_1, \ldots, r_T\} \in [0,1]^T$. Let $1 = t_1 < t_2 \cdots < t_\ell = T + 1$ be $\ell$ time-step, with $\ell \leq T$. Then, the mechanism*

$$
\begin{pmatrix} r_1 \\ r_2 \\ \vdots \\ r_T \end{pmatrix} \xrightarrow{\mathcal{C}} \begin{pmatrix} r_1 + \cdots + r_{t_2-1} + Y_1 \\ r_1 + \cdots + r_{t_3-1} + Y_1 + Y_2 \\ \vdots \\ r_1 + \cdots + r_T + Y_1 + Y_2 + \cdots + Y_{\ell-1} \end{pmatrix}
$$

*is $\epsilon$-DP, where $(Y_1, \ldots, Y_\ell) \sim^{iid} \mathrm{Lap}(1/\epsilon)$.*

*Proof of Lemma 6.* First, consider trying to release the following partial sums

$$
\begin{pmatrix} r_1 \\ r_2 \\ \vdots \\ r_T \end{pmatrix} \rightarrow \begin{pmatrix} r_1 + \cdots + r_{t_2-1} \\ r_{t_2} + \cdots + r_{t_3-1} \\ \vdots \\ r_{t_{\ell-1}} + \cdots + r_T \end{pmatrix} .
$$

Because the rewards are in $[0,1]$, the sensitivity of each partial sum is 1. Since each partial sum is computed on non-overlapping sequences, combining the Laplace mechanism (Theorem 5) with the parallel composition property of DP (Lemma 10) gives that

$$
\begin{pmatrix} r_1 \\ r_2 \\ \vdots \\ r_T \end{pmatrix} \xrightarrow{\mathcal{P}} \begin{pmatrix} r_1 + \cdots + r_{t_2-1} + Y_1 \\ r_{t_2} + \cdots + r_{t_3-1} + Y_2 \\ \vdots \\ r_{t_{\ell-1}} + \cdots + r_T + Y_{\ell-1} \end{pmatrix}
$$

is $\epsilon$-DP, where $(Y_1, \ldots, Y_{\ell-1}) \sim^{iid} \mathrm{Lap}(1/\epsilon)$.

Consider the post-processing function $f : (x_1, \ldots x_{\ell-1}) \rightarrow (x_1, x_1 + x_2, \ldots, x_1 + x_2 + \cdots + x_{\ell-1})$. Then, we have that that $\mathcal{C} = f \circ \mathcal{P}$. So, by the post-processing property of DP, $\mathcal{C}$ is $\epsilon$-DP. $\square$

*Proof of Proposition 2.* Let $\pi$ be either DP-IMED or DP-KLUCB. Let $\mathbf{r} \triangleq \{r_1, \ldots, r_T\}$ and $\mathbf{r'} \triangleq \{r'_1, \ldots, r'_T\}$ be two neighbouring reward lists, that only differ at $t^\star \in \{1, \ldots, T\}$. Fix $\mathbf{a} \triangleq (a_1, \ldots, a_T) \in [K]^T$. We want to show that

$$
\mathcal{V}_{\mathbf{r}}^\pi(\mathbf{a}) \leq e^\epsilon \mathcal{V}_{\mathbf{r'}}^\pi(\mathbf{a}) .
$$

Step 1: Probability decomposition and time-steps before $t^\star$:

$$
\frac{\mathcal{V}_{\mathbf{r}}^\pi(\mathbf{a})}{\mathcal{V}_{\mathbf{r'}}^\pi(\mathbf{a})} = \prod_{t=1}^T \frac{\pi_t(a_t|a_1, r_1, \ldots a_{t-1}, r_{t-1})}{\pi_t(a_t|a_1, r'_1, \ldots a_{t-1}, r'_{t-1})}
$$

$$
= \prod_{t=t^\star+1}^T \frac{\pi_t(a_t|a_1, r_1, \ldots a_{t-1}, r_{t-1})}{\pi_t(a_t|a_1, r'_1, \ldots a_{t-1}, r'_{t-1})},
$$

since for $t < t^\star$, $r_t = r'_t$. Let us denote by $\Pr(\mathbf{a}^{>t^\star} \mid \mathbf{a}^{\leq t^\star}, \mathbf{r}) \triangleq \prod_{t=t^\star+1}^T \pi_t(a_t|a_1, r_1, \ldots a_{t-1}, r_{t-1})$ the probability of the policy recommending the sequence $(a_{t^\star+1}, \ldots, a_T)$, when interacting with $\mathbf{r} = \{r_1, \ldots, r_T\}$ and already recommending $a_1, \ldots, a_{t^\star}$ in the first steps.

Let us denote by $t_1, \ldots, t_\ell$ the time-steps of the beginning of the phases when $\pi$ interacts with $\mathbf{r}$, and $t'_1, \ldots, t'_{\ell'}$ the time-steps of the beginning of the phases when $\pi$ interacts with $\mathbf{r'}$. Also, let $t_{k_\star}$ be the beginning of the phase for which $t^\star$ belongs in list $\mathbf{r}$ phases. Similarly, let $t'_{k'_\star}$ be the beginning of the phase for which $t^\star$ belongs in list $\mathbf{r'}$ phases.

Since $(a_1, \ldots, a_T)$ is fixed, and $r_t = r'_t$ for $t < t^\star$, then $t_{k_\star} = t'_{k'_\star}$ and $k^\star = k'_\star$, i.e., $t^\star$ falls at the same phase in $\mathbf{r}$ and $\mathbf{r'}$.

Step 2: Considering the noisy sum of rewards at phase $k^\star$:

Let $\tilde{S}^p_{k^\star} = \sum_{s=t_{k^\star}}^{t_{k^\star+1}-1} r_s + Y_{k_\star}$ be the noisy partial sum of rewards collected at phase $k^\star$ for $\mathbf{r}$, where $Y_{k^\star} \sim \mathrm{Lap}(1/\epsilon)$. Similarly, let $\tilde{S}'^p_{k^\star} = \sum_{s=t_{k^\star}}^{t_{k^\star+1}-1} r'_s + Y'_{k_\star}$ be the noisy partial sum of rewards collected at phase $k^\star$ for $\mathbf{r'}$, where $Y'_{k^\star} \sim \mathrm{Lap}(1/\epsilon)$. We make two main observations:

(a) If the value of the noisy partial sum at phase $k^\star$ is exactly the same between the neighbouring $\mathbf{r}$ and $\mathbf{r'}$, then the policy $\pi$ will recommend the sequence of actions $\mathbf{a}^{>t^\star}$ with the same probability under $\mathbf{r}$ and $\mathbf{r'}$:

$$\Pr(\mathbf{a}^{>t^\star} \mid \mathbf{a}^{\leq t^\star}, \mathbf{r}, \tilde{S}^p_{k^\star} = s) = \Pr(\mathbf{a}^{>t^\star} \mid \mathbf{a}^{\leq t^\star}, \mathbf{r'}, \tilde{S}'^p_{k^\star} = s) . \tag{28}$$

This is due to the structure of the algorithm $\pi$, where the reward at step $t^\star$ only affects the statistic $\tilde{S}^p_{k^\star}$, and nothing else.

(b) Since rewards are $[0,1]$, using the Laplace mechanism, we have that

$$\Pr(\tilde{S}^p_{k^\star} = s \mid \mathbf{a}^{\leq t^\star}, \mathbf{r}) \leq e^\epsilon \Pr(\tilde{S}'^p_{k^\star} = s \mid \mathbf{a}^{\leq t^\star}, \mathbf{r'}) . \tag{29}$$

Step 3: Combining Eq. 28 and Eq. 29, aka post-processing:

We have

$$\Pr(\mathbf{a}^{>t^\star} \mid \mathbf{a}^{\leq t^\star}, \mathbf{r}) = \int_{s \in \mathbb{R}} \Pr(\tilde{S}^p_{k^\star} = s \mid \mathbf{a}^{\leq t^\star}, \mathbf{r}) \Pr(\mathbf{a}^{>t^\star} \mid \mathbf{a}^{\leq t^\star}, \mathbf{r}, \tilde{S}^p_{k^\star} = s)$$

$$\leq \int_{s \in \mathbb{R}} e^\epsilon \Pr(\tilde{S}'^p_{k^\star} = s \mid \mathbf{a}^{\leq t^\star}, \mathbf{r'}) \Pr(\mathbf{a}^{>t^\star} \mid \mathbf{a}^{\leq t^\star}, \mathbf{r'}, \tilde{S}'^p_{k^\star} = s)$$

$$= e^\epsilon \Pr(\mathbf{a}^{>t^\star} \mid \mathbf{a}^{\leq t^\star}, \mathbf{r'}) .$$

This concludes the proof:

$$\frac{\mathcal{V}^\pi_{\mathbf{r}}(\mathbf{a})}{\mathcal{V}^\pi_{\mathbf{r'}}(\mathbf{a})} = \frac{\Pr(\mathbf{a}^{>t^\star} \mid \mathbf{a}^{\leq t^\star}, \mathbf{r})}{\Pr(\mathbf{a}^{>t^\star} \mid \mathbf{a}^{\leq t^\star}, \mathbf{r'})} \leq e^\epsilon .$$

$\square$

# F   Regret Analysis Proof

**Lemma 7** (Explicit solution of $\mathrm{d}_\epsilon$). *If $\mu, \mu' \in (0,1)$ and $\mu \leq \mu'$, we have*

$$\mathrm{d}_\epsilon(\mu, \mu') \triangleq \inf_{z \in [\mu, \mu']} \{\mathrm{kl}(z, \mu') + \epsilon(z - \mu)\}, \tag{30}$$

*under Bernoulli cases, then*

$$z^* = \max\left(\mu, \frac{\mu'}{\mu' + (1-\mu')e^\epsilon}\right).$$

*solves the optimisation problem. Thus, we have*

$$\mathrm{d}_\epsilon(\mu, \mu') = \begin{cases} \mathrm{kl}\left(\mu, \mu'\right), & \text{if } \mu \geq \frac{\mu'}{\mu' + (1-\mu')e^\epsilon}, \\ \mathrm{kl}\left(\frac{\mu'}{\mu' + (1-\mu')e^\epsilon}, \mu'\right) + \epsilon\left(\frac{\mu'}{\mu' + (1-\mu')e^\epsilon} - \mu\right), & \text{if } \mu \leq \frac{\mu'}{\mu' + (1-\mu')e^\epsilon}. \end{cases} \tag{31}$$

*For $\mu \geq \mu'$,*

$$z^* = \min\left(\frac{\mu'e^\epsilon}{\mu'e^\epsilon + (1-\mu')}, \mu\right).$$

*and*

$$\mathrm{d}_\epsilon(\mu, \mu') = \begin{cases} \mathrm{kl}\left(\mu, \mu'\right), & \text{if } \mu \leq \frac{\mu'e^\epsilon}{\mu'e^\epsilon + (1-\mu')}, \\ \mathrm{kl}\left(\frac{\mu'e^\epsilon}{\mu'e^\epsilon + (1-\mu')}, \mu'\right) + \epsilon\left(\mu - \frac{\mu'e^\epsilon}{\mu'e^\epsilon + (1-\mu')}\right), & \text{if } \mu \geq \frac{\mu'e^\epsilon}{\mu'e^\epsilon + (1-\mu')}. \end{cases} \tag{32}$$

*Proof.* The Kullback-Leibler divergence between two Bernoulli random variables with means $z$ and $\mu'$ is given by

$$\mathrm{kl}(z, \mu') = z \log \frac{z}{\mu'} + (1 - z) \log \frac{1 - z}{1 - \mu'} \, .$$

The optimisation problem is

$$\mathrm{d}_\epsilon(\mu, \mu') = \inf_{z \in [\mu, \mu']} \left\{ z \log \frac{z}{\mu'} + (1 - z) \log \frac{1 - z}{1 - \mu'} + \epsilon(z - \mu) \right\} \, .$$

To find the optimal $z^*$, take the derivative of the objective function with respect to $z$ and let it equal to 0:

$$\frac{\partial}{\partial z} \left[ z \log \frac{z}{\mu'} + (1 - z) \log \frac{1 - z}{1 - \mu'} + \epsilon(z - \mu) \right] = 0 \, .$$

By calculation, we have

$$\log \frac{z(1 - \mu')}{\mu'(1 - z)} + \epsilon = 0 \, .$$

Rearrange for $z$, to obtain

$$z = \frac{\mu'}{\mu' + (1 - \mu')e^\epsilon} \, .$$

The optimal $z^*$ must lie within the interval $[\mu, \mu']$, hence we have

$$z^* = \max \left( \mu, \min \left( \mu', \frac{\mu'}{\mu' + (1 - \mu')e^\epsilon} \right) \right) \, .$$

We always have $\frac{\mu'}{\mu' + (1 - \mu')e^\epsilon} \le \mu'$, so we can remove the min part:

$$z^* = \max \left( \mu, \frac{\mu'}{\mu' + (1 - \mu')e^\epsilon} \right) \, .$$

Thus, we obtain

$$\mathrm{d}_\epsilon(\mu, \mu') = \begin{cases} \mathrm{kl}(\mu, \mu') & \text{if} \quad \mu \ge \dfrac{\mu'}{\mu' + (1 - \mu')e^\epsilon} \\[3mm] \mathrm{kl}\left( \dfrac{\mu'}{\mu' + (1 - \mu')e^\epsilon}, \mu' \right) + \epsilon \left( \dfrac{\mu'}{\mu' + (1 - \mu')e^\epsilon} - \mu \right) & \text{if} \quad \mu \le \dfrac{\mu'}{\mu' + (1 - \mu')e^\epsilon} \end{cases}$$

Now, we consider $\mu \ge \mu'$,

$$\mathrm{d}_\epsilon(\mu, \mu') = \inf_{z \in [\mu', \mu]} \left\{ \mathrm{kl}(z, \mu') + \epsilon(\mu - z) \right\} \, .$$

So, we need to minimise

$$\mathrm{d}_\epsilon(\mu, \mu') = \inf_{z \in [\mu', \mu]} \left\{ z \log \frac{z}{\mu'} + (1 - z) \log \frac{1 - z}{1 - \mu'} + \epsilon(\mu - z) \right\}$$

over $z \in [\mu', \mu]$. Differentiating the objective function with respect to $z$ and setting it equal to 0, we have

$$\log \frac{z}{\mu'} - \log \frac{1 - z}{1 - \mu'} - \epsilon = 0.$$

Solving for $z$, we get

$$z^* = \frac{\mu' e^\epsilon}{\mu' e^\epsilon + (1 - \mu')} \ge \mu' \, .$$

Projecting the solution to $[\mu', \mu]$, then we have that the optimal solution is

$$z^* = \min \left( \frac{\mu' e^\epsilon}{\mu' e^\epsilon + (1 - \mu')}, \mu \right) \, .$$

Thus, the explicit solution is

$$\mathrm{d}_\epsilon(\mu, \mu') = \begin{cases} \mathrm{kl}(\mu, \mu'), & \text{if} \quad \mu \le \dfrac{\mu' e^\epsilon}{\mu' e^\epsilon + (1 - \mu')}, \\[3mm] \mathrm{kl}\left( \dfrac{\mu' e^\epsilon}{\mu' e^\epsilon + (1 - \mu')}, \mu' \right) + \epsilon \left( \mu - \dfrac{\mu' e^\epsilon}{\mu' e^\epsilon + (1 - \mu')} \right), & \text{if} \quad \mu \ge \dfrac{\mu' e^\epsilon}{\mu' e^\epsilon + (1 - \mu')}. \end{cases}$$

$\square$

**Lemma 8.** *For any $\mu, \mu' \in [0, 1]$,*

$$\left| \frac{d\{d_\epsilon(\mu, \mu')\}}{d\mu} \right| \leq \epsilon.$$

*Proof.* For $\mu \leq \mu'$, from (31), we have the explicit solution. If $\mu \geq \frac{\mu'}{\mu'+(1-\mu')e^\epsilon}$,

$$d_\epsilon(\mu, \mu') = kl(\mu, \mu') = \mu \log \frac{\mu}{\mu'} + (1 - \mu) \log \frac{1 - \mu}{1 - \mu'}$$

Its derivative with respect to $\mu$ is

$$\frac{d\{d_\epsilon(\mu, \mu')\}}{d\mu} = \frac{d}{d\mu} kl(\mu, \mu') = \log \frac{\mu(1 - \mu')}{\mu'(1 - \mu)}.$$

We have the condition

$$\mu' \geq \mu \geq \frac{\mu'}{\mu' + (1 - \mu')e^\epsilon}.$$

Since $\mu' \geq \mu$, we note that

$$\frac{d\{d_\epsilon(\mu, \mu')\}}{d\mu} \leq 0.$$

Similarly, since $\mu \geq \frac{\mu'}{\mu'+(1-\mu')e^\epsilon}$, we substitute this into the derivative

$$\frac{\mu(1 - \mu')}{\mu'(1 - \mu)} \geq \frac{\left(\frac{\mu'}{\mu'+(1-\mu')e^\epsilon}\right)(1 - \mu')}{\mu'\left(1 - \frac{\mu'}{\mu'+(1-\mu')e^\epsilon}\right)} = \frac{1}{e^\epsilon}.$$

Thus,

$$-\epsilon \leq \log \frac{\mu(1 - \mu')}{\mu'(1 - \mu)} \leq 0.$$

If $\mu \leq \frac{\mu'}{\mu'+(1-\mu')e^\epsilon}$, then

$$\frac{d\{d_\epsilon(\mu, \mu')\}}{d\mu} = -\epsilon.$$

Therefore, for $\mu \leq \mu'$,

$$-\epsilon \leq \frac{d\{d_\epsilon(\mu, \mu')\}}{d\mu} \leq 0.$$

Now, we consider the case of $\mu \geq \mu'$. From the explicit solution (32), when $\mu \geq \frac{\mu'e^\epsilon}{\mu'e^\epsilon+(1-\mu')}$, $\frac{d\{d_\epsilon(\mu,\mu')\}}{d\mu} = \epsilon$ and the result holds. Let's consider $\mu \leq \frac{\mu'e^\epsilon}{\mu'e^\epsilon+(1-\mu')}$, similar to the above argument, we have

$$\frac{d\{d_\epsilon(\mu, \mu')\}}{d\mu} = \frac{d}{d\mu} kl(\mu, \mu') = \log \frac{\mu(1 - \mu')}{\mu'(1 - \mu)} \in [0, \epsilon].$$

Thus, we have the result in the lemma. $\qquad\square$

**Lemma 9.** *For any $0 \leq \mu \leq \mu' < 1$,*

$$\frac{d\{d_\epsilon(\mu, \mu')\}}{d\mu'} \leq \frac{1}{1 - \mu'}.$$

*Proof.* Considering the definition of $d_\epsilon$ in (6), we have for $0 \leq \mu \leq \mu' < 1$

$$d_\epsilon(\mu, \mu') = \inf_{z \in [\mu, \mu']} kl(z, \mu') + \epsilon(z - \mu).$$

We first show

$$\frac{d\{d_\epsilon(\mu, \mu')\}}{d\mu'} \leq \frac{d\{kl(\mu, \mu')\}}{d\mu'}. \tag{33}$$

From the explicit solution in (31), we have if $\mu \geq \frac{\mu'}{\mu'+(1-\mu')e^\epsilon}$, then $d_\epsilon(\mu, \mu') = kl(\mu, \mu')$. So the inequality holds. If $\mu \leq \frac{\mu'}{\mu'+(1-\mu')e^\epsilon}$, let $f(\mu') = \frac{\mu'}{\mu'+(1-\mu')e^\epsilon}$, then $f'(\mu') = \frac{e^\epsilon}{(\mu'+(1-\mu')e^\epsilon)^2}$. In this case, $d_\epsilon(\mu, \mu') = kl(f(\mu'), \mu') + \epsilon(f(\mu') - \mu)$ where $\mu \leq f(\mu') \leq \mu'$. By calculation, we have for this case,

$$\frac{d\{d_\epsilon(\mu, \mu')\}}{d\mu'} = f'(\mu')\left(\log\frac{f(\mu')}{\mu'} - \log\frac{1-f(\mu')}{1-\mu'} + \epsilon\right) + \frac{\mu' - f(\mu')}{\mu'(1-\mu')}.$$

Note that $\log\frac{f(\mu')}{\mu'} - \log\frac{1-f(\mu')}{1-\mu'} + \epsilon = 0$ and $\mu \leq f(\mu')$. And we bound

$$\begin{aligned}\frac{d\{kl(\mu, \mu')\}}{d\mu'} &= \frac{1-\mu}{1-\mu'} - \frac{\mu}{\mu'}\\ &= \frac{1}{1-\mu'}\frac{\mu'-\mu}{\mu'}\\ &\leq \frac{1}{1-\mu'}.\end{aligned}$$

Thus, we have the result. $\qquad\square$

**Theorem 3** (Regret upper bound of DP-IMED). *Assume $\mu^\star < 1$. Under the batch sizes given in* (12) *with $\alpha > 1$, the regret bound of DP-IMED for a Bernoulli bandit $\nu$ is*

$$\text{Reg}_T(\text{DP-IMED}, \nu) \leq \sum_{i \neq i^*} \frac{\alpha \Delta_i \log T}{d_\epsilon(\mu_i, \mu^\star)} + o(\log T).$$

*Proof of Theorem 3.* Let $\mathcal{T}$ be the set of rounds $t$ such that Lines 7–12 are run, that is, the rounds such that the arm selection occurred. For $t \in \mathcal{T}$, we define $\tilde{\mu}_i(t)$ as $\tilde{\mu}_{i,n_m}$ when $N_i(t-1) = n_m$. Let $j$ be any optimal arm, that is, $j$ such that $\Delta_j = 0$. By the batched structure of the algorithm, we have

$$\begin{aligned}\text{Regret}(T) &= \sum_{i \neq i^*} \sum_{t=1}^{T} (\mu^\star - \mu_i) \mathbb{1}[i(t) = i]\\ &\leq n_0 \sum_{i \neq i^*} (\mu^\star - \mu_i) + \sum_{i \neq i^*} (\mu^\star - \mu_i) \sum_{t=1}^{T}\sum_{m=0}^{\infty} B_{m+1}\mathbb{1}[i(t) = i, N_i(t-1) = n_m, t \in \mathcal{T}]\\ &\leq n_0 \sum_{i \neq i^*} (\mu^\star - \mu_i)\\ &\quad + \sum_{i \neq i^*} (\mu^\star - \mu_i) \underbrace{\sum_{t=1}^{T}\sum_{m=0}^{\infty} B_{m+1}\mathbb{1}[i(t) = i, N_i(t-1) = n_m, t \in \mathcal{T}, \tilde{\mu}_j(t) < \mu^\star - \delta]}_{(A)}\\ &\quad + \sum_{i \neq i^*} (\mu^\star - \mu_i) \underbrace{\sum_{t=1}^{T}\sum_{m=0}^{\infty} B_{m+1}\mathbb{1}[i(t) = i, N_i(t-1) = n_m, t \in \mathcal{T}, \tilde{\mu}_j(t) \geq \mu^\star - \delta]}_{(B)},\end{aligned}$$

$$(34)$$

where $\delta > 0$ is a small constant. (A) and (B) correspond to the regret before and after the convergence, respectively.

Note that we have

$$n_m = \left\lceil n_0 \frac{\alpha^{m+1}-1}{\alpha-1}\right\rceil \leq n_0 \frac{\alpha^{m+1}-1}{\alpha-1} + 1, \qquad (35)$$

and

$$B_m = n_m - n_{m-1} \leq n_0 \frac{\alpha^{m+1}-1}{\alpha-1} - n_0 \frac{\alpha^m - 1}{\alpha-1} + 1 \leq 2n_0\alpha^m. \qquad (36)$$

**Pre-convergence Term.** First consider (A). Define

$$\bar{I}_j = \max_{m:\tilde{\mu}_{j,m}<\mu^\star-\delta} \left\{ n_m d_\epsilon([\tilde{\mu}_{j,m}]_0^1, \mu^\star-\delta) + \log n_m \right\}, \tag{37}$$

where we define $\bar{I}_j = -\infty$ if $\tilde{\mu}_{j,m} \geq \mu^\star - \delta$ for all $m \in \mathbb{Z}_+$. Then, $\{i(t) = i, t \in \mathcal{T}, \tilde{\mu}_j(t) < \mu^\star - \delta\}$ implies that

$$I_i(t) = I^*(t) \leq I_j(t) \leq N_j(t-1) d_\epsilon([\tilde{\mu}_j(t)]_0^1, \mu^\star-\delta) + \log N_j(t) \leq \bar{I}_j,$$

where $I^*(t) = \max_{i'} I_i(t)$ is the optimal arm obtained by Line 8 in Algorithm 1. By this fact we have

$$(A) \leq \sum_{t=1}^T \sum_{m=0}^\infty B_{m+1} \mathbb{1}\left[ i(t) = i, N_i(t-1) = n_m, t \in \mathcal{T}, I_i(t) \leq \bar{I}_j \right]$$

$$\leq \sum_{t=1}^T \sum_{m=0}^\infty B_{m+1} \mathbb{1}\left[ i(t) = i, N_i(t-1) = n_m, t \in \mathcal{T}, \log N_i(t-1) \leq \bar{I}_j \right]$$

$$\leq \sum_{t=1}^T \sum_{m=0}^\infty B_{m+1} \mathbb{1}\left[ i(t) = i, N_i(t-1) = n_m, t \in \mathcal{T}, n_m \leq e^{\bar{I}_j} \right]$$

$$\leq \sum_{m=0}^\infty B_{m+1} \sum_{t=1}^T \mathbb{1}\left[ i(t) = i, N_i(t-1) = n_m, t \in \mathcal{T}, n_0 \frac{\alpha^m - 1}{\alpha - 1} \leq e^{\bar{I}_j} \right]$$

$$= \sum_{m=0}^{\left\lfloor \log_\alpha((\alpha-1)e^{\bar{I}_j}/n_0+1) \right\rfloor} B_{m+1} \sum_{t=1}^T \mathbb{1}\left[ i(t) = i, N_i(t-1) = n_m, t \in \mathcal{T} \right].$$

Since $\{i(t) = i, N_i(t-1) = n_m\}$ can occur at most once for each $m$, we have

$$(A) \leq \sum_{m=0}^{\left\lfloor \log_\alpha((\alpha-1)e^{\bar{I}_j}/n_0+1) \right\rfloor} B_{m+1}$$

$$= n_{\left\lfloor \log_\alpha((\alpha-1)e^{\bar{I}_j}/n_0+1) \right\rfloor+1} - n_0$$

$$= \left\lceil n_0 \frac{\alpha^{\left\lfloor \log_\alpha((\alpha-1)e^{\bar{I}_j}/n_0+1) \right\rfloor+2} - 1}{\alpha - 1} \right\rceil - n_0$$

$$\leq n_0 \frac{\alpha^2((\alpha-1)e^{\bar{I}_j}/n_0 + 1) - 1}{\alpha - 1} - n_0 + 1$$

$$= \alpha^2 e^{\bar{I}_j} + \alpha n_0 + 1$$

$$= \alpha^2 \max_{m:\tilde{\mu}_{j,m}<\mu^\star-\delta} \left\{ n_m e^{n_m d_\epsilon([\tilde{\mu}_{j,m}]_0^1, \mu^\star-\delta)} \right\} + \alpha n_0 + 1$$

$$\leq \alpha^2 \sum_{m=0}^\infty \mathbb{1}\left[ \tilde{\mu}_{j,m} < \mu^\star - \delta \right] n_m e^{n_m d_\epsilon([\tilde{\mu}_{j,m}]_0^1, \mu^\star-\delta)} + \alpha n_0 + 1. \tag{38}$$

Now, let us consider the expectation of (38). When $\tilde{\mu}_{j,m} < \mu^\star - \delta$ we have

$$d_\epsilon([\tilde{\mu}_{j,m_j}]_0^1, \mu^\star - \delta) = d_\epsilon([\tilde{\mu}_{j,m_j}]_0^1, \mu^\star) - \int_{\mu^\star-\delta}^{\mu^\star} \left. \frac{d\{d_\epsilon\left([\tilde{\mu}_{j,m_j}]_0^1, \mu\right)\}}{d\mu} \right|_{\mu=u} du$$

$$\geq d_\epsilon([\tilde{\mu}_{j,m_j}]_0^1, \mu^\star) - \frac{\delta}{1-\mu^\star} \quad \text{(by Lemma 9)}$$

$$= d_\epsilon([\tilde{\mu}_{j,m_j}]_0^1, \mu^\star) - \delta',$$

where we set $\delta' = \delta/(1-\mu^\star)$.

Let $P(x) = \Pr[\mathrm{d}_\epsilon([\tilde\mu_{j,m_j}]_0^1, \mu^\star) \geq x, \tilde\mu_{j,m} < \mu^\star - \delta]$. If $\tilde\mu_{j,m} < \mu^\star - \delta$, then $0 \leq \mathrm{d}_\epsilon([\tilde\mu_{j,m_j}]_0^1, \mu^\star) \leq d_1 := \mathrm{d}_\epsilon(0, \mu^\star)$. Hence, we have

$$\mathbb{E}\left[\mathbb{1}\left[\tilde\mu_{j,m} < \mu^\star - \delta\right] n_m \mathrm{e}^{n_m \mathrm{d}_\epsilon([\tilde\mu_{j,m}]_0^1, \mu^\star - \delta)}\right]$$

$$\leq \mathbb{E}\left[\mathbb{1}\left[\tilde\mu_{j,m} < \mu^\star - \delta\right] n_m \mathrm{e}^{n_m\left(\mathrm{d}_\epsilon([\tilde\mu_{j,m}]_0^1, \mu^\star) - \delta'\right)}\right]$$

$$= \int_0^{d_1} n_m \mathrm{e}^{n_m(x - \delta')} \,\mathrm{d}(-P(x))$$

$$= [n_m \mathrm{e}^{n_m(x-\delta')}(-P(x))]_{x=0}^{d_1} + \int_0^{d_1} n_m^2 \mathrm{e}^{n_m(x-\delta')} P(x) \,\mathrm{d}x$$

$$\leq n_m \mathrm{e}^{-n_m\delta'} + \int_0^{d_1} n_m^2 \mathrm{e}^{n_m(x-\delta')} P(x) \,\mathrm{d}x \;.$$

Let $c_x \in [0, \mu^\star]$ be such that $\mathrm{d}_\epsilon(c_x, \mu^\star) = x$. Then

$$\left\{\mathrm{d}_\epsilon([\tilde\mu_{j,m_j}]_0^1, \mu^\star) \geq x, \tilde\mu_{j,m} < \mu^\star - \delta\right\} \Leftrightarrow \left\{\tilde\mu_{j,m_j} < c_x, \tilde\mu_{j,m} < \mu^\star - \delta\right\} \;.$$

Therefore,

$$P(x) = \Pr[\tilde\mu_{j,m_j} < c_x, \tilde\mu_{j,m} < \mu^\star - \delta] \leq A_a \mathrm{e}^{n_m a} \mathrm{e}^{-n_m \mathrm{d}_\epsilon(c_x, \mu^\star)} = A_a \mathrm{e}^{n_m a} \mathrm{e}^{-n_m x}, \qquad (39)$$

for any $a > 0$ by Corollary 1. Thus, we have

$$\mathbb{E}\left[\mathbb{1}\left[\tilde\mu_{j,m} < \mu^\star - \delta\right] n_m \mathrm{e}^{n_m \mathrm{d}_\epsilon([\tilde\mu_{j,m}]_0^1, \mu^\star - \delta)}\right]$$

$$\leq n_m \mathrm{e}^{-n_m\delta'} + \int_0^{d_1} n_m^2 \mathrm{e}^{n_m(x-\delta')} A_a \mathrm{e}^{n_m a} \mathrm{e}^{-n_m x} \,\mathrm{d}x$$

$$= n_m \mathrm{e}^{-n_m\delta'} + d_1 n_m^2 A_a \mathrm{e}^{-n_m(\delta'-a)} \;. \qquad (40)$$

By letting $a < \delta'$ and combining (38) with (40), we obtain

$$\mathbb{E}[(A)] \leq \alpha^2 \sum_{m=0}^\infty \left(n_m \mathrm{e}^{-n_m\delta'} + d_1 n_m^2 A_a \mathrm{e}^{-n_m(\delta'-a)}\right) + \alpha n_0 + 1$$

$$\leq \alpha^2 \sum_{n=0}^\infty \left(n \mathrm{e}^{-n\delta'} + d_1 n^2 A_a \mathrm{e}^{-n(\delta'-a)}\right) + \alpha n_0 + 1$$

$$= \alpha^2 \left(\frac{\mathrm{e}^{-(\delta'-a)}}{(1 - \mathrm{e}^{-(\delta'-a)})^2} + d_1 A_a \frac{\mathrm{e}^{-(\delta'-a)}(\mathrm{e}^{-(\delta'-a)} + 1)}{(1 - \mathrm{e}^{-(\delta'-a)})^3}\right) + \alpha n_0 + 1$$

$$= \alpha^2 \left(\frac{\mathrm{e}^{-(\delta'-a)}}{(1 - \mathrm{e}^{-(\delta'-a)})^2} + d_1 A_a \frac{\mathrm{e}^{-(\delta'-a)}(\mathrm{e}^{-(\delta'-a)} + 1)}{(1 - \mathrm{e}^{-(\delta'-a)})^3}\right) + \alpha n_0 + 1$$

$$= O(1) \;. \qquad (41)$$

**Post-convergence Term** Next we consider (B). Since $\mathrm{d}_\epsilon(\mu, \mu) = 0$ for any $\mu \in [0, 1]$, we have

$$I^*(t) \leq \max_{i':\tilde\mu_{i'}(t) = \tilde\mu^*(t)} I_{i'}(t) = \max_{i':\tilde\mu_{i'}(t) = \tilde\mu^*(t)} \log N_{i'}(T) \leq \log T \;.$$

On the other hand, $i(t) = i$, $N_i(t-1) = n_m$, $t \in \mathcal{T}$, $\tilde\mu_j(t) \geq \mu^\star - \delta$ implies that

$$I^*(t) = I_i(t) \geq n_m \mathrm{d}_\epsilon\left([\tilde\mu_i(t)]_0^1, [\tilde\mu^*(t)]_0^1\right) = n_m \mathrm{d}_\epsilon\left([\tilde\mu_i(t)]_0^1, \mu^\star - \delta\right),$$

from which we have

$$\left\{i(t) = i, N_i(t-1) = n_m, t \in \mathcal{T}, \tilde\mu_j(t) \geq \mu^\star - \delta\right\} \subset \left\{n_m \mathrm{d}_\epsilon\left([\tilde\mu_i(t)]_0^1, \mu^\star - \delta\right) \leq \log T\right\} \;.$$

So, we have

$$(B) = \sum_{t=1}^T \sum_{m=0}^\infty B_{m+1} \mathbb{1}\left[i(t) = i, N_i(t-1) = n_m, t \in \mathcal{T}, \tilde\mu_j(t) \geq \mu^\star - \delta\right]$$

$$\leq \sum_{t=1}^{T} \sum_{m=0}^{\infty} B_{m+1} \mathbb{1}\left[i(t)=i,\ N_i(t-1)=n_m,\ n_m \mathrm{d}_\epsilon\left([\tilde{\mu}_i(t)]_0^1, \mu^\star - \delta\right) \leq \log T\right]$$

$$= \sum_{m=0}^{\infty} B_{m+1} \mathbb{1}\left[n_m \mathrm{d}_\epsilon\left([\tilde{\mu}_{i,n_m}]_0^1, \mu^\star - \delta\right) \leq \log T\right] \sum_{t=1}^{T} \mathbb{1}\left[i(t)=i,\ N_i(t-1)=n_m\right]$$

$$\leq \sum_{m=0}^{\infty} B_{m+1} \mathbb{1}\left[n_m \mathrm{d}_\epsilon\left([\tilde{\mu}_{i,n_m}]_0^1, \mu^\star - \delta\right) \leq \log T\right] \tag{42}$$

$$\leq \sum_{m=0}^{\infty} B_{m+1} \mathbb{1}\left[n_m(\mathrm{d}_\epsilon([\tilde{\mu}_{i,n_m}]_0^1, \mu^\star) - \delta') \leq \log T\right],$$

where recall that $\delta' = \delta/(1-\mu^\star)$ and the last inequality follows from Lemma 9.

Let

$$H = \frac{\log T}{\mathrm{d}_\epsilon(\mu_i, \mu^\star) - 2\delta'} \tag{43}$$

and

$$m^* = \inf\{m \in \mathbb{N} : n_m \geq H\}\,.$$

Then, by $n_{m^*-1} < H$ and (35) we have

$$H > n_{m^*-1} = \left\lceil n_0 \frac{\alpha^{m^*}-1}{\alpha-1} \right\rceil \geq n_0 \frac{\alpha^{m^*}-1}{\alpha-1},$$

which implies

$$m^* \leq \log_\alpha\left(\frac{(\alpha-1)H}{n_0}+1\right)\,. \tag{44}$$

Now, the post-convergence term can be bounded as follows:

$$\mathbb{E}[(\mathrm{B})] \leq \sum_{m=0}^{\infty} B_{m+1} \Pr\left[n_m(\mathrm{d}_\epsilon([\tilde{\mu}_{i,n_m}]_0^1, \mu^\star) - \delta') \leq \log T\right]$$

$$\leq \sum_{m=0}^{m^*-1} B_{m+1} + \sum_{m=m^*}^{\infty} B_{m+1} \Pr\left[n_m(\mathrm{d}_\epsilon([\tilde{\mu}_{i,n_m}]_0^1, \mu^\star) - \delta') \leq \log T\right]$$

$$\leq n_{m^*} - n_0 + \sum_{m=m^*}^{\infty} B_{m+1} \Pr\left[H\left(\mathrm{d}_\epsilon([\tilde{\mu}_{i,m}]_0^1, \mu^\star) - \delta'\right) \leq \log T\right]$$

$$< n_0 \frac{\alpha^{m^*+1}-1}{\alpha-1} + 1 - n_0 + \sum_{m=m^*}^{\infty} B_{m+1} \Pr\left[H\left(\mathrm{d}_\epsilon([\tilde{\mu}_{i,m}]_0^1, \mu^\star) - \delta'\right) \leq \log T\right]$$

$$\leq n_0 \frac{\alpha\left(\frac{(\alpha-1)H}{n_0}+1\right)-1}{\alpha-1} + 1 - n_0 + \sum_{m=m^*}^{\infty} B_{m+1} \Pr\left[\mathrm{d}_\epsilon([\tilde{\mu}_{i,m}]_0^1, \mu^\star) \leq \mathrm{d}_\epsilon(\mu_i, \mu^\star) - \delta'\right]$$

$$\text{(by (43) and (44))}$$

$$\leq \alpha H + 1 - \frac{n_0 \alpha}{\alpha-1} + \sum_{m=m^*}^{\infty} B_{m+1} \Pr\left[\tilde{\mu}_{i,m} \geq \mu_i + \delta'/\epsilon\right] \qquad \text{(by Lemma 8)}$$

$$\leq \alpha H + \sum_{m=m^*}^{\infty} B_{m+1} A_{a'} \mathrm{e}^{a' n_m} \mathrm{e}^{-n_m(\mathrm{d}_\epsilon(\mu_i+\delta'/\epsilon, \mu^\star))} \qquad \text{(by Corollary 1)}$$

$$\leq \alpha H + \sum_{m=m^*}^{\infty} 2n_0 \alpha^{m+1} A_{a'} \mathrm{e}^{-n_0 \frac{\alpha^{m+1}-1}{\alpha-1}(\mathrm{d}_\epsilon(\mu_i+\delta'/\epsilon, \mu^\star)-a')} \qquad \text{(by (36))} \tag{45}$$

$$= \alpha H + A_{a'} \mathrm{e}^\Lambda \sum_{m=m^*}^{\infty} 2n_0 \alpha^{m+1} \mathrm{e}^{-\alpha^{m+1}\Lambda} \tag{46}$$

$$\leq \alpha H + 2n_0 A_{a'} \mathrm{e}^\Lambda \int_{m^*}^\infty \alpha^{x+1} \mathrm{e}^{-\alpha^x \Lambda} \, \mathrm{d}x$$

$$= \alpha H + \frac{2\alpha n_0 A_{a'} \mathrm{e}^\Lambda}{\ln(\alpha)\Lambda} e^{-\alpha^{m^*}\Lambda}$$

$$= \frac{\alpha \log T}{\mathrm{d}_\epsilon(\mu_i, \mu^\star) - 2\delta'} + o(1) \,. \tag{47}$$

Here, in (45) we took $a' < \mathrm{d}_\epsilon(\mu_i + \delta'/\epsilon, \mu^\star)$ and in (46) we defined

$$\Lambda = \frac{n_0(\mathrm{d}_\epsilon(\mu_i + \delta'/\epsilon, \mu^\star) - a')}{\alpha - 1} \,.$$

We complete the proof by combining (34), (41), and (47), and letting $\delta' = \frac{\delta}{1-\mu^\star} \downarrow 0$. $\qquad \square$

**Theorem 4** (Regret upper bound of DP-KLUCB). *Assume $\mu^\star < 1$. Under the batch sizes given in (12) with $\alpha > 1$, the regret bound of DP-KLUCB for a Bernoulli bandit $\nu$ is*

$$\mathrm{Reg}_T(\mathsf{DP\text{-}KLUCB}, \nu) \leq \sum_{i \neq i^*} \frac{\alpha \Delta_i \log T}{\mathrm{d}_\epsilon(\mu_i, \mu^\star)} + o(\log T) \,.$$

*Proof of Theorem 4.* By the same argument as the analysis for DP-IMED we have

$$\mathrm{Regret}(T) \leq n_0 \sum_{i \neq i^*} (\mu^\star - \mu_i)$$

$$+ \sum_{i \neq i^*} (\mu^\star - \mu_i) \underbrace{\sum_{t=1}^T \sum_{m=0}^\infty B_{m+1} \mathbb{1}\left[i(t) = i,\, N_i(t-1) = n_m,\, t \in \mathcal{T},\, \bar{\mu}^\star(t) < \mu^\star - \delta\right]}_{(\mathrm{A})}$$

$$+ \sum_{i \neq i^*} (\mu^\star - \mu_i) \underbrace{\sum_{t=1}^T \sum_{m=0}^\infty B_{m+1} \mathbb{1}\left[i(t) = i,\, N_i(t-1) = n_m,\, t \in \mathcal{T},\, \bar{\mu}^\star(t) \geq \mu^\star - \delta\right]}_{(\mathrm{B})}, \tag{48}$$

where $\bar{\mu}^\star(t) = \max_i \bar{\mu}_i(t)$.

We use a transformation of these terms that is similar to Honda [2019] but more suitable for the batched algorithm. First, we have

$$(\mathrm{A}) = \sum_{t=1}^T \sum_{m=0}^\infty B_{m+1} \mathbb{1}\left[i(t) = i,\, N_i(t-1) = n_m,\, t \in \mathcal{T},\, \bar{\mu}^\star(t) < \mu^\star - \delta\right]$$

$$\leq \sum_{t=1}^T \sum_{m=0}^\infty B_{m+1} \mathbb{1}\left[i(t) = i,\, N_i(t-1) = n_m,\, t \in \mathcal{T},\, \bar{\mu}_j(t) < \mu^\star - \delta\right].$$

Let

$$\bar{I}'_j = \max_{m:\tilde{\mu}_{j,m} < \mu^\star - \delta} \left\{ n_m \mathrm{d}_\epsilon([\tilde{\mu}_{j,m}]_0^1, \mu^\star - \delta) \right\}.$$

Since

$$\{\bar{\mu}_j(t) < \mu^\star - \delta\} \Leftrightarrow \left\{ \sup\left\{ \mu : \mathrm{d}_\epsilon([\tilde{\mu}_j(t)]_0^1, \mu) \leq \frac{\log t}{N_j(t-1)} \right\} < \mu^\star - \delta \right\}$$

$$\Rightarrow \left\{ \mathrm{d}_\epsilon([\tilde{\mu}_j(t)]_0^1, \mu^\star - \delta) > \frac{\log t}{N_j(t-1)},\, \tilde{\mu}_j(t) < \mu^\star - \delta \right\}$$

$$\Leftrightarrow \left\{ t < \mathrm{e}^{N_j(t-1)\mathrm{d}_\epsilon([\tilde{\mu}_j(t)]_0^1, \mu^\star - \delta)},\, \tilde{\mu}_j(t) < \mu^\star - \delta \right\},$$

we see that

$$(A) \leq \sum_{t=1}^{T} \sum_{m=0}^{\infty} B_{m+1} \mathbb{1} \left[ i(t) = i, N_i(t-1) = n_m, t \in \mathcal{T}, t < e^{N_j(t-1)d_\epsilon([\tilde{\mu}_{j,m}]_0^1, \mu^\star - \delta)}, \tilde{\mu}_{j,m} < \mu^\star - \delta \right]$$

$$\leq \sum_{t=1}^{T} \sum_{m=0}^{\infty} B_{m+1} \mathbb{1} \left[ i(t) = i, N_i(t-1) = n_m, t < e^{\bar{I}'_j} \right]$$

$$\leq \sum_{t=1}^{T} \sum_{m=0}^{\infty} B_{m+1} \mathbb{1} \left[ i(t) = i, N_i(t-1) = n_m, n_m < e^{\bar{I}'_j} - 1 \right] \quad \text{(by } N_i(t-1) \leq t-1\text{)}$$

$$= \sum_{m=0}^{\infty} B_{m+1} \mathbb{1} \left[ n_m < e^{\bar{I}'_j} - 1 \right] \sum_{t=1}^{T} \mathbb{1} \left[ i(t) = i, N_i(t-1) = n_m \right]$$

$$\leq \sum_{m=0}^{\infty} B_{m+1} \mathbb{1} \left[ n_m < e^{\bar{I}'_j} - 1 \right]$$

$$\leq (\alpha+1)e^{\bar{I}'_j} \quad \text{(by } n_m = \sum_{i=0}^{m} B_m \text{ and } B_{m+1} \leq \alpha n_m\text{)}$$

$$\leq (\alpha+1)e^{\bar{I}_j},$$

where $\bar{I}_j$ is defined in (37). The evaluation of this expectation is the one same as (38), which results in $\mathbb{E}[(A)] = O(1)$.

Now, we consider the second term. Note that $i(t) = i$ implies $\bar{\mu}^\star(t) = \bar{\mu}_i(t)$ and we also have

$$\{\bar{\mu}_i(t) \geq \mu^\star - \delta\} \Leftrightarrow \left\{ \sup \left\{ \mu : d_\epsilon([\tilde{\mu}_j(t)]_0^1, \mu) \leq \frac{\log t}{N_j(t)} \right\} \geq \mu^\star - \delta \right\}$$

$$\Rightarrow \left\{ d_\epsilon([\tilde{\mu}_i(t)]_0^1, \mu^\star - \delta) \leq \frac{\log t}{N_i(t)} \right\}.$$

Then, we have

$$(B) = \sum_{t=1}^{T} \sum_{m=0}^{\infty} B_{m+1} \mathbb{1} \left[ i(t) = i, N_i(t-1) = n_m, t \in \mathcal{T}, \bar{\mu}^\star(t) \geq \mu^\star - \delta \right]$$

$$= \sum_{t=1}^{T} \sum_{m=0}^{\infty} B_{m+1} \mathbb{1} \left[ i(t) = i, N_i(t-1) = n_m, t \in \mathcal{T}, \bar{\mu}_i(t) \geq \mu^\star - \delta \right]$$

$$\leq \sum_{t=1}^{T} \sum_{m=0}^{\infty} B_{m+1} \mathbb{1} \left[ i(t) = i, N_i(t-1) = n_m, t \in \mathcal{T}, d_\epsilon([\tilde{\mu}_{i,n_m}]_0^1, \mu^\star - \delta) \leq \frac{\log t}{n_m} \right]$$

$$\leq \sum_{m=0}^{\infty} B_{m+1} \mathbb{1} \left[ d_\epsilon([\tilde{\mu}_{i,n_m}]_0^1, \mu^\star - \delta) \leq \frac{\log t}{n_m} \right] \sum_{t=1}^{T} \mathbb{1} \left[ i(t) = i, N_i(t-1) = n_m \right]$$

$$\leq \sum_{m=0}^{\infty} B_{m+1} \mathbb{1} \left[ d_\epsilon([\tilde{\mu}_{i,n_m}]_0^1, \mu^\star - \delta) \leq \frac{\log t}{n_m} \right],$$

whose expectation is analysed in (42). $\qquad\square$

**Comparison to the regret bound of AdaP-KLUCB in Azize and Basu [2022]** Theorem 8 in Azize and Basu [2022] shows that for $\tau > 3$, AdaP-KLUCB yields a regret

$$\text{Reg}_T(\text{AdaP-KLUCB}, \nu) \leq \sum_{a:\Delta_a > 0} \left( \frac{C_1(\tau)\Delta_a}{\min\{\text{kl}(\mu_a, \mu^*), C_2\epsilon\Delta_a\}} \log(T) + \frac{3\tau}{\tau - 3} \right), \quad (49)$$

where $C_1(\tau)$ and $C_2 > 0$ are defined as

$$\inf_{\beta \in \mathbf{B}} \max \left\{ \frac{(1+\beta)\alpha}{\text{kl}(\mu_a, \mu^*)}, \frac{(1+\tau)}{(c(\beta) - \gamma_{\ell,T})\epsilon\Delta_a} \right\} \log(T) \triangleq \frac{\frac{1}{4}C_1(\tau)}{\min\{\text{kl}(\mu_a, \mu^*), C_2\epsilon\Delta_a\}} \log(T),$$

such that $\tau$ is a constant that controls the optimism in AdaP-KLUCB, $\mathbf{B} \triangleq \{\beta > 0 : c(\beta) > \gamma_{\ell,T}\}$, for $\beta > 0$, $c(\beta) \in [0,1]$ is defined such that: $\mathrm{kl}(\mu_a + c(\beta)\Delta_a, \mu^\star) = \frac{d(\mu_a, \mu^\star)}{1+\beta}$, and $\gamma_{\ell,T}$ such that $\mathrm{kl}(\mu_a + \gamma_{\ell,T}\Delta_a, \mu_a) = \frac{\log(T)}{2^\ell}$ for $T$ the horizon and $\ell$ the phase.

In general, $C_1$ and $C_2$ may depend on $\mu_a$ and $\mu^\star$, and thus are not "constants". In contrast, our bound in Theorem 2 matches the asymptotic lower bound of Theorem 1 up to the exact constant $\alpha > 1$ that controls the geometrically increasing batches and which can be chosen arbitrarily close to 1. In addition, our analysis only requires that the number of batches is sublinear in $T$, as seen from Proposition 1. As a result, we can also use a polynomially increasing batch size instead of $B_m \approx \alpha^m$, which fully makes the regret *asymptotically optimal*. We used a geometrically increasing batch size here just for simplicity.

**Comment on KL-UCB and IMED algorithms.** We present both DP-KLUCB and DP-IMED to show that, for two different algorithmic design philosophies in bandits (KL-UCB and IMED), our privacy framework of geometric batching without forgetting, combined with our new concentration inequality, can design algorithms with optimal regret upper bounds.

(a) KL-UCB and IMED belong to fundamentally different algorithmic bandit families:

- KL-UCB is a UCB-style algorithm that relies on optimism in the face of uncertainty, and constructs upper confidence bounds based on Chernoff's inequality.

- IMED is an information-theoretic method that selects arms based on empirical divergence minimisation, comparing empirical rewards to the estimated best arm.

(b) Our proposed privacy framework works for both KL-UCB and IMED: our framework estimates the unknown means privately by running the algorithm in geometrically increasing phases, and accumulating Laplace noises from each phase, i.e. no forgetting. In addition, our tight DP-Chernoff concentration inequality (Proposition 1) directly provides new $d_\epsilon$-based indexes for both KL-UCB and IMED style algorithms, tightly balancing exploration and exploitation under noisy DP observations. Combining everything with a generic regret upper bound analysis provides two optimal DP bandit algorithms.

**Improved regret bounds of KL-UCB/IMED v.s. UCB** The improvement introduced by using asymptotically optimal algorithms (KL-UCB/IMED) compared to the vanilla UCB algorithm Lattimore and Szepesvári [2020] boils down to comparing $\mathrm{kl}(\mu_a, \mu^\star)$ with the squared gap $\Delta_a^2 = (\mu^\star - \mu_a)^2$.

(a) Using Pinsker's inequality, we always have that $\mathrm{kl}(\mu_a, \mu^\star) \geq 2\Delta_a^2$

(b) However, for close values of $\mu_a$ and $\mu^\star$, a Taylor expansion shows that

$$\mathrm{kl}(\mu_a, \mu^\star) = \frac{\Delta_a^2}{2\mu_a(1-\mu_a)} + o(\Delta_a^2)$$

which means that

$$\frac{\Delta_a^{-2}}{\mathrm{kl}(\mu_a, \mu^\star)^{-1}} = \frac{1}{2\mu_a(1-\mu_a)} + o(1) \to \infty$$

when $\mu_a$ tends to either 0 or 1. This means that our algorithms DP-KLUCB and DP-IMED improve over the state-of-the-art algorithms (AdaP-UCB and Lazy-DP-TS) in a problem-dependent constant (related to the variance of Bernoullis), which could blow up for some hard instances close to the borders 0 and 1.

## G   Extended Experiments

In this section, we present additional experiments comparing the algorithms in four bandit environments with Bernoulli distributions, as defined by Sajed and Sheffet [2019], namely

$$\mu_1 = \{0.75, 0.70, 0.70, 0.70, 0.70\}, \quad \mu_2 = \{0.75, 0.625, 0.5, 0.375, 0.25\},$$
$$\mu_3 = \{0.75, 0.53125, 0.375, 0.28125, 0.25\}, \quad \mu_4 = \{0.75, 0.71875, 0.625, 0.46875, 0.25\}.$$

and four budgets $\epsilon \in \{0.01, 0.1, 0.5, 1\}$. The results are presented in Figure 4 for $\mu_1$, Figure 5 for $\mu_2$, Figure 6 for $\mu_3$ and Figure 7 for $\mu_4$. We implement all the algorithms in Python (version 3.8) and on an 8 core 64-bits Intel i5@1.6 GHz CPU.

For all the environments and privacy budgets tested, DP-IMED and DP-KLUCB achieve the lowest regret.

**Comparison to the lower bound.** We also plot the regret as a function of the privacy budget $\epsilon$ in Figure 8. The algorithm chosen is DP-IMED with $\alpha = 1.1$, $T = 10^7$ and for bandit environment $\mu = [0.8, 0.1, 0.1, 0.1, 0.1]$. We discretise the $[0, 1]$ interval into 100 values of $\epsilon$. For each $\epsilon$, we run the algorithm 100 times and plot the mean and standard deviation of the regret in $[0, 1]$. We also plot the asymptotic regret lower bound in Figure 8 for $T = 10^7$ and $\mu$ as a function of $\epsilon$. The performance of our algorithm DP-IMED matches the regret lower bound. We also remark that the change between the high and the low privacy regimes happens smoothly.

**Effect of $\alpha$, the geometric batching hyper-paramter.** In all previous figures, we took $\alpha = 2$, which corresponds to arm-dependent doubling. The reason we chose $\alpha = 2$ is to have a "fair" comparison to the other algorithms in the literature, i.e. DP-SE, AdaP-KLUCB and Lazy-DP-TS, which all use an arm-dependent doubling in the original papers, and in theory, could also be implemented using geometrically increasing batches of any ratio $\alpha > 1$. By taking $\alpha = 2$, we mainly focus on the effect of our algorithm's two main algorithmic novelties: getting rid of forgetting and new $d_\epsilon$-based indexes. In Figure 9, we plot the regret of DP-IMED as a function of time steps for different values of $\alpha$. As $\alpha$ gets smaller, the performance of DP-IMED gets better. However, the performance worsens when $\alpha$ is very close to 1. At the limit when $\alpha \to 1$, each arm-dependent phase length tends to 1, and thus, one Laplace noise is added to each Bernoulli reward sample. This is equivalent to local DP, where the price of privacy is high.

**Real-world dataset.** We add an experiment inspired by the COV-BOOST Munro et al. [2021], Kone et al. [2023] dataset. COV-BOOST is a Phase 2 clinical trial, conducted on 2,883 participants, to measure the immunogenicity of different COVID-19 vaccines as a third dose. This resulted in a total of 20 vaccination strategies being assessed, each of them defined by the vaccines received as first, second and third doses. In Table 4 of Kone et al. [2023], the authors report the average immune responses induced by each candidate strategy in cohorts of participants. From this study, we extract and process the Anti-spike IgG average immune response for each strategy, then project them in $[0, 1]$ to simulate Bernoulli bandits with $K = 20$ arms, and run our algorithms with different values of $\epsilon$. We report the evolution of regret for this specific Bernoulli instance, under different values of $\epsilon$ in Figure 10. DP-IMED and DP-KLUCB still achieve the lowest regret for this instance.

# H  Limitations

In this section, we describe some of the limitations of our results.

- Our matching upper and lower bounds are asymptotic in the horizon $T$. This is also the case in classic multi-armed bandit results without privacy. An interesting direction is to investigate the effect of privacy on the $o(\log(T))$ terms, which are committed in the current analysis.

- Our algorithms and regret upper bounds are tailored for Bernoulli distributions. This is a fundamental setting in bandits and an important first step for understanding the interplay between privacy and sequential decision-making. Generalising the analysis to other distributions is an interesting future direction.

- An important ingredient in our algorithms is geometrically increasing batching. Our concentration results allow for any batching strategy where the batch size $n_T$ is negligible in $T$, i.e. $n_T = o(T)$. However, it is unclear if it is possible to eliminate this design choice altogether, like we did with forgetting. This is an important direction to explore, especially for adversarial bandits, where arm-dependent batching strategies like those used in the stochastic setting are bound to fail.

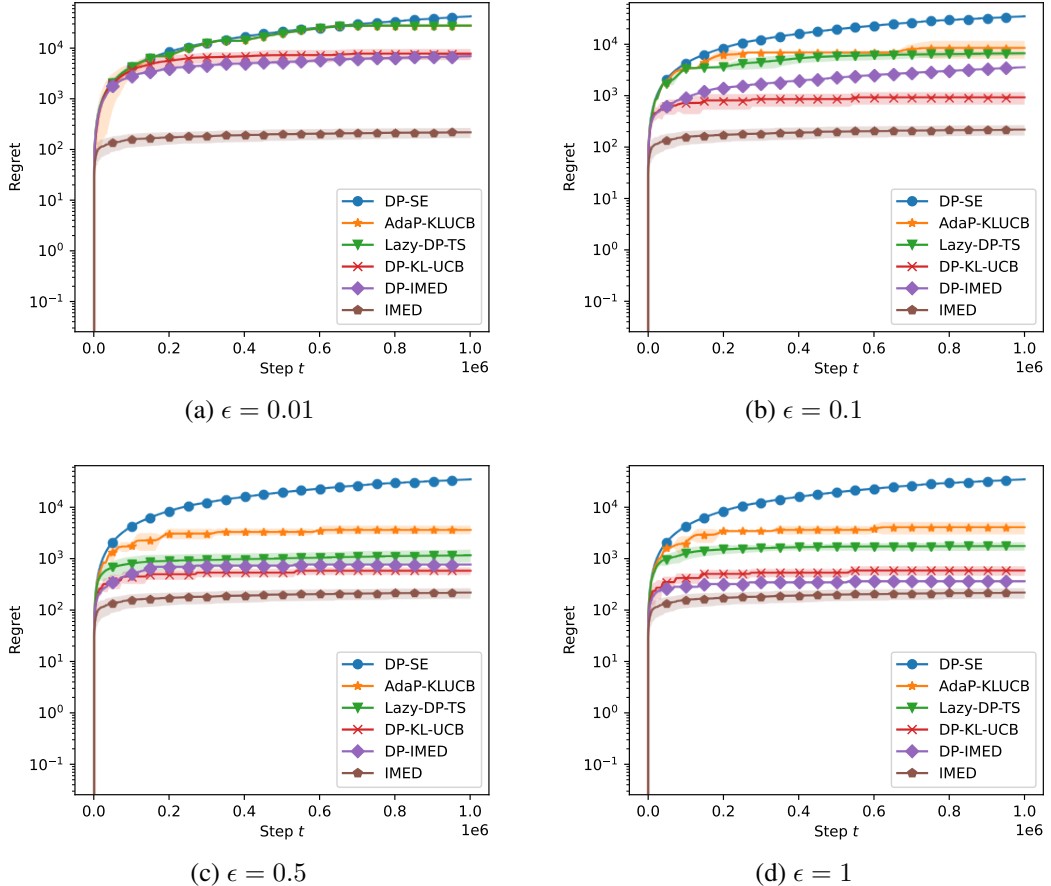

Figure 4: Evolution of regret (mean $\pm 2$ std) over time for $\mu_1$ for different budgets $\epsilon$.

# I  Existing technical results and Definitions

**Proposition 4** (Post-processing [Dwork and Roth, 2014]). *Let $\mathcal{M}$ be a mechanism and $f$ be an arbitrary randomised function defined on $\mathcal{M}$'s output. If $\mathcal{M}$ is $\epsilon$-DP, then $f \circ \mathcal{M}$ is $\epsilon$-DP.*

The post-processing property ensures that any quantity constructed only from a private output is still private, with the same privacy budget. This is a consequence of the data processing inequality.

**Proposition 5** (Group Privacy [Dwork and Roth, 2014]). *Let $D$ and $D'$ be two datasets in $\mathcal{X}^n$. If $\mathcal{M}$ is $(\epsilon, \delta)$-DP, then for any event $E \in \mathcal{F}$*

$$\mathcal{M}_D(A) \le e^{\epsilon d_{Ham}(D,D')} \mathcal{M}_{D'}(A) . \tag{50}$$

Group privacy translates the closeness of output distributions on neighbouring input datasets to a closeness of output distributions on any two datasets $D$ and $D'$ that depend on the Hamming distance $d_{\mathrm{Ham}}(D, D')$. This property will be the basis for proving lower bounds in Section 3.

**Proposition 6** (Simple Composition). *Let $\mathcal{M}^1, \ldots, \mathcal{M}^k$ be $k$ mechanisms. We define the mechanism*

$$\mathcal{G} : D \to \bigotimes_{i=1}^{k} \mathcal{M}_D^i$$

*as the $k$ composition of the mechanisms $\mathcal{M}^1, \ldots, \mathcal{M}^k$.*

- *If each $\mathcal{M}^i$ is $(\epsilon_i, \delta_i)$-DP, then $\mathcal{G}$ is $(\sum_{i=1}^{k} \epsilon_i, \sum_{i=1}^{k} \delta_i)$-DP.*

- *If each $\mathcal{M}^i$ is $\rho_i$-zCDP, then $\mathcal{G}$ is $\sum_{i=1}^{k} \rho_i$-zCDP.*

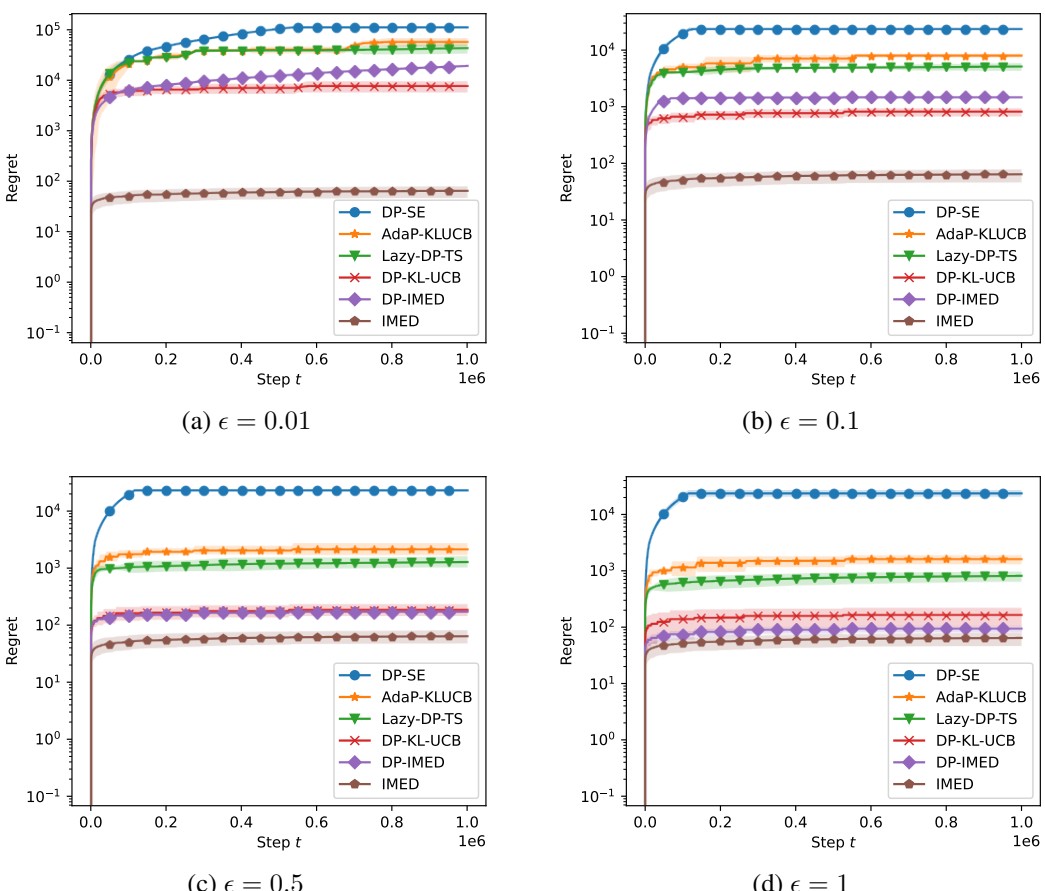

Figure 5: Evolution of regret (mean $\pm 2$ std) over time for $\mu_2$ for different budgets $\epsilon$.

Composition is a fundamental property of DP. Composition helps to analyse the privacy of sophisticated algorithms, by understanding the privacy of each building block, and summing directly the privacy budgets. Proposition 6 can be improved in two directions. (a) It is possible to show that the result is still true if the mechanisms are chosen adaptively, and that the mechanism at step $i$ takes as auxiliary input the outputs of the last $i-1$ mechanisms. (b) Advanced composition theorems Kairouz et al. [2015] for $(\epsilon, \delta)$-DP improve the dependence on $k$ the number of composed mechanisms. Specifically, if the same mechanism is composed $k$ times, Proposition 6 concludes that the composed mechanism is $(k\epsilon, k\delta)$-DP. Advanced composition Kairouz et al. [2015] shows that the k-fold adaptively composed mechanism is $(\epsilon', \delta' + k\delta)$-DP for any $\delta'$ where $\epsilon' \triangleq \sqrt{2k\log(1/\delta')}\epsilon + k\epsilon(e^\epsilon - 1)$. Roughly speaking, advanced composition provides a $(\sqrt{k}\epsilon, \delta)$-DP guarantee, improving by $\sqrt{k}$ the $(k\epsilon, k\delta)$-DP guarantee of simple composition.

In addition to the classic composition theorems, we provide here an additional property of interest: parallel composition.

**Lemma 10** (Parallel Composition). *Let $\mathcal{M}^1, \ldots, \mathcal{M}^k$ be $k$ mechanisms, such that $k < n$, where $n$ is the size of the input dataset. Let $t_1, \ldots t_k, t_{k+1}$ be indexes in $[1, n]$ such that $1 = t_1 < \cdots < t_k < t_{k+1} - 1 = n$.*
*Let's define the following mechanism*

$$\mathcal{G} : \{x_1, \ldots, x_n\} \to \bigotimes_{i=1}^{k} \mathcal{M}^i_{\{x_{t_i}, \ldots, x_{t_{i+1}-1}\}}$$

*$\mathcal{G}$ is the mechanism that we get by applying each $\mathcal{M}^i$ to the $i$-th partition of the input dataset $\{x_1, \ldots, x_n\}$ according to the indexes $t_1 < \cdots < t_k < t_{k+1}$.*

- *If each $\mathcal{M}^i$ is $(\epsilon, \delta)$-DP, then $\mathcal{G}$ is $(\epsilon, \delta)$-DP*

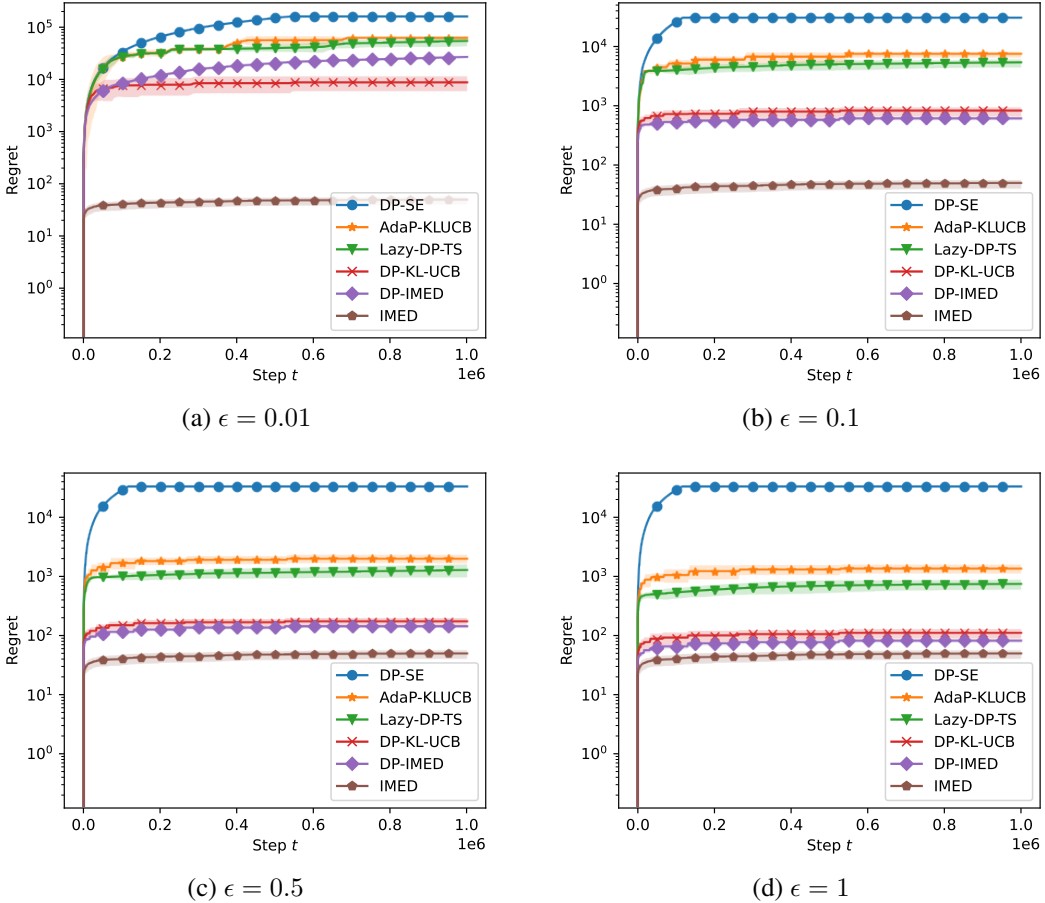

Figure 6: Evolution of regret (mean $\pm 2$ std) over time for $\mu_3$ for different budgets $\epsilon$.

- *If each $\mathcal{M}^i$ is $\rho_i$-zCDP, then $\mathcal{G}$ is $\rho$-zCDP*

In parallel composition, the $k$ mechanisms are applied to different "non-overlapping" parts of the input dataset. If each mechanism is DP, then the parallel composition of the $k$ mechanisms is DP, *with the same privacy budget*. This property will be the basis for designing private bandit algorithms in Section 4.

**Theorem 5** (The Laplace Mechanism [Dwork and Roth, 2014])**.** *Let $f : \mathcal{X} \to \mathbb{R}^k$ be a deterministic algorithm with $\ell_1$ sensitivity $s_1(f) \triangleq \max\limits_{D \sim D'} \|f(D) - f(D')\|_1$. Let*

$$\mathcal{M}_L(f, \epsilon) \triangleq f + (Y_1, \ldots, Y_k),$$

*where $Y_i$ are i.i.d from $\mathrm{Lap}\left(\frac{s_1(f)}{\epsilon}\right)$, where the Laplace distribution centred at 0 with scale b, denoted $\mathrm{Lap}(b)$, is the distribution with probability density function*

$$\mathrm{Lap}(x|b) \triangleq \frac{1}{2b} \exp\left(-\frac{|x|}{b}\right),$$

*for any $x \in \mathbb{R}$.*

*The mechanism $\mathcal{M}_L(f, \epsilon)$ is called the Laplace mechanism and satisfies $\epsilon$-DP.*

**Lemma 11** (Chernoff Tail Bound via KL Divergence [Boucheron et al., 2003])**.** *Let $X_1, X_2, \ldots, X_n$ be independent Bernoulli random variables with success probabilities $p_1, p_2, \ldots, p_n$. Define $S_n = \sum_{i=1}^{n} X_i$, and let $\mu = \mathbb{E}[S_n] = \sum_{i=1}^{n} p_i$. Then the following bounds hold:*

- *Upper Tail Bound: for any $a > \mu$*

$$P(S_n \geq a) \leq \exp\left(-n \cdot \mathrm{kl}\left(\frac{a}{n}, \frac{\mu}{n}\right)\right),$$

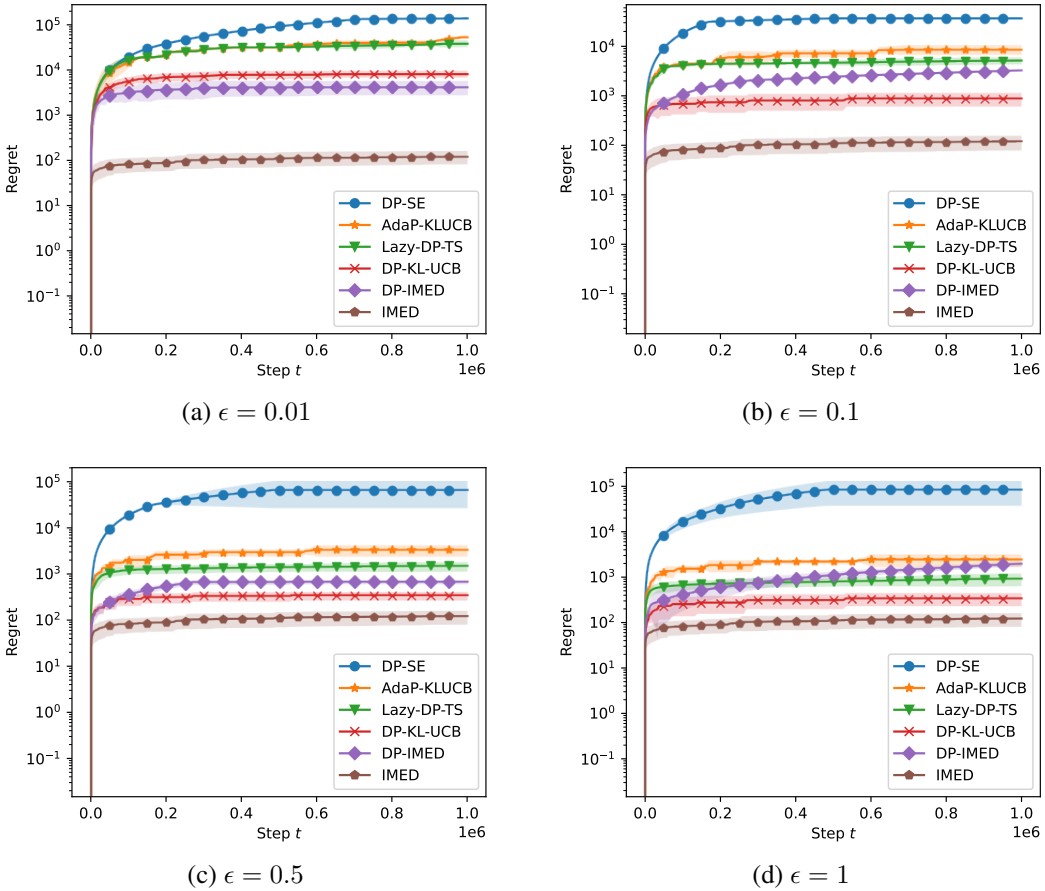

(a) $\epsilon = 0.01$          (b) $\epsilon = 0.1$

(c) $\epsilon = 0.5$          (d) $\epsilon = 1$

Figure 7: Evolution of regret (mean $\pm 2$ std) over time for $\mu_4$ for different budgets $\epsilon$.

where $\mathrm{kl}(p, q)$ is defined as

$$\mathrm{kl}(p, q) = p \log \frac{p}{q} + (1-p) \log \frac{1-p}{1-q} \, .$$

- *Lower Tail Bound: for any $a < \mu$*

$$P(S_n \leq a) \leq \exp\left(-n \cdot \mathrm{kl}\left(\frac{a}{n}, \frac{\mu}{n}\right)\right) \, .$$

**Lemma 12** (Asymptotic Maximal Hoeffding Inequality). *Assume that $X_i$ has positive mean $\mu$ and that $X_i - \mu$ is $\sigma$-sub-Gaussian. Then,*

$$\forall \epsilon > 0, \lim_{n \to \infty} \mathbb{P}\left(\frac{\max_{s \leq n} \sum_{i=1}^{s} X_i}{n} \leq (1+\epsilon)\mu\right) = 1 \, .$$

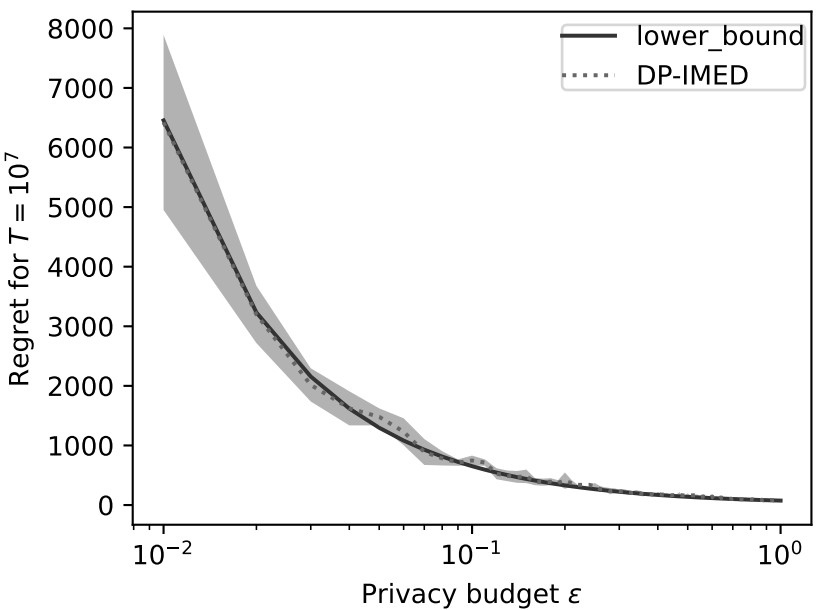

Figure 8: Evolution of the regret for $T = 10^7$ with respect to $\epsilon$ for DP-IMED on $\mu \triangleq [0.8, 0.1, 0.1, 0.1, 0.1]$, compared to the asymptotic regret lower bound of Theorem 1.

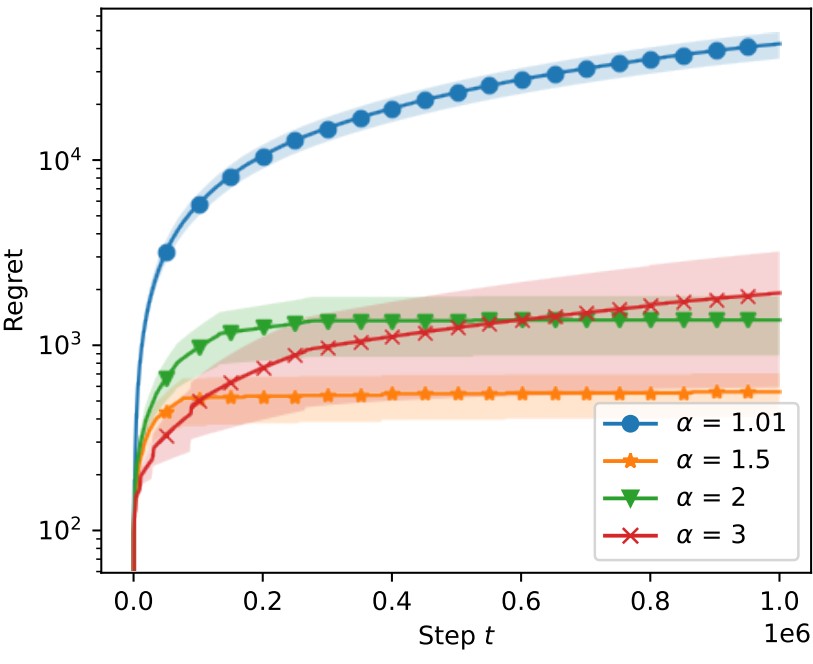

Figure 9: Effect of $\alpha$ on the regret of DP-IMED, on $\mu_2$ and $\epsilon = 0.25$.

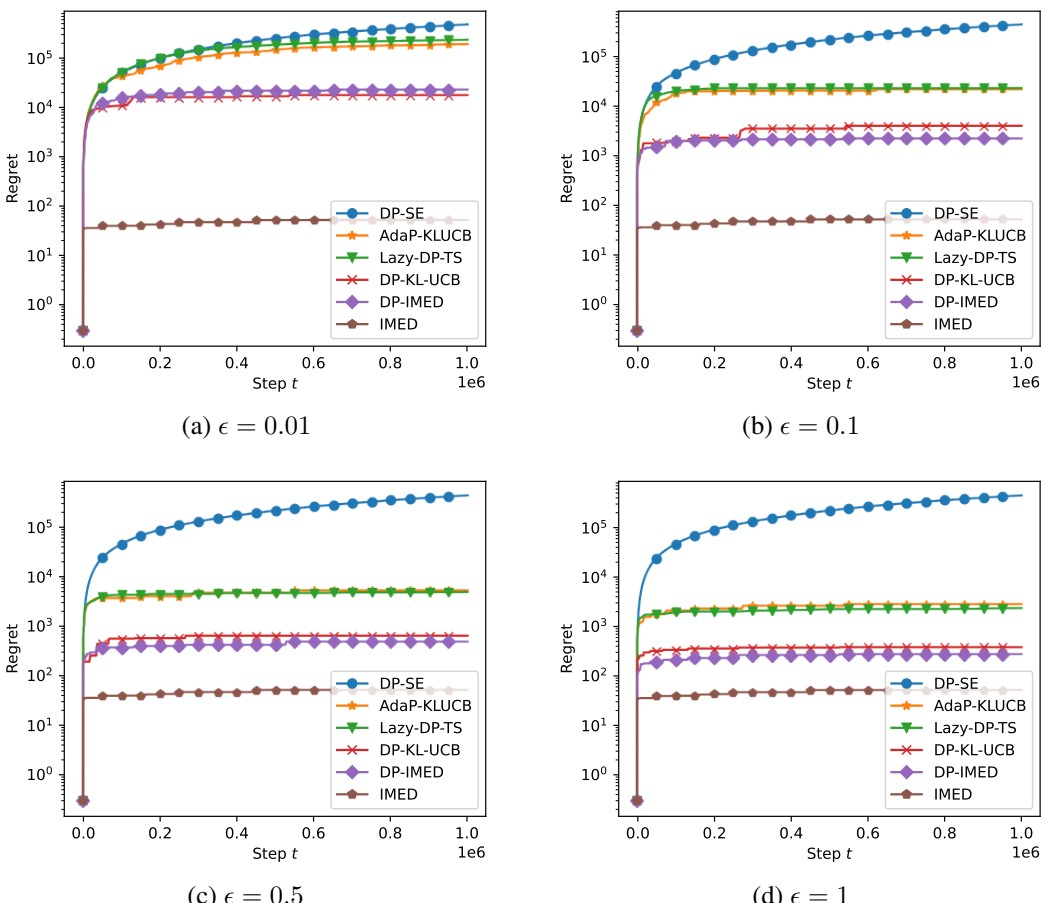

Figure 10: Evolution of regret (mean $\pm 2$ std) over time for the COV-BOOST dataset, for different budgets $\epsilon$.

