# OpenReview forum: "Optimal Regret of Bandits under Differential Privacy"
_NeurIPS.cc/2025/Conference — NeurIPS 2025 poster_

### Official Review · Reviewer_aTyM · 2025-06-30

**Clarity:** 3
**Significance:** 2
**Originality:** 2
**Rating:** 4
**Confidence:** 4

**Summary:**

This paper investigates regret minimization in Bernoulli bandits under $\epsilon$-global Differential Privacy (DP). The authors first define DP in the context of bandit settings, formalizing privacy constraints for sequential decision-making. They then derive a tighter regret lower bound for the general problem. Next, they develop a new concentration inequality for sums of Bernoulli random variables with added Laplace noise, serving as a DP analog of the Chernoff bound. Based on this, they propose two algorithms, DP-KLUCB and DP-IMED, which operate in arm-dependent phases with geometrically increasing batch sizes. Finally, the authors prove the DP properties of these algorithms and establish regret upper bounds that asymptotically match the lower bound up to a constant arbitrarily close to 1, with numerical experiments validating their superior performance.

**Questions:**

(a) The current results, particularly the regret lower bound and algorithms, are specific to Bernoulli bandits. Could the authors discuss the potential for extending these results to other reward distributions, such as sub-Gaussian or heavy-tailed distributions? What specific challenges do they anticipate in such extensions, and are there any preliminary ideas or approaches to address these challenges? Additionally, can the strategy of using rewards from previous phases be applied to bandits with other distributions?

(b) The design of the algorithms heavily relies on Proposition 1, a new concentration inequality. Could the authors provide intuition behind this proposition, explaining how it facilitates differentially private bandit algorithms? Additionally, how did the concentration inequality from Azize and Basu [2022] inspire the development of this new, tighter inequality?

(c) The introduction highlights adaptive clinical trials as a motivating example. Could the authors provide a detailed description of how the proposed algorithms, such as DP-KLUCB and DP-IMED, could be applied in this context? Specifically, how do these algorithms ensure patient privacy while optimizing treatment allocation?

**Ethical Concerns:**

["NO or VERY MINOR ethics concerns only"]

**Final Justification:**

I have added the score by 1, raising my rating from 3 to 4 in my previous reply. I think the paper can be accepted.

**Limitations:**

Yes.

**Paper Formatting Concerns:**

No.

**Quality:**

2

**Strengths And Weaknesses:**

(a) Quality:  The technical contributions of this paper include the proposed algorithms and the proofs for both the lower and upper regret bounds. The lower bound proof is particularly compelling, introducing a novel term (\mathrm{d}_\epsilon(\mu_a, \mu^*)) and employing a double change of environment technique to achieve a tighter bound. However, the algorithms appear as straightforward extensions of existing methods like , DP-KLUCB and DP-IMED,, lacking significant innovation in their formulation. Similarly, the upper bound proof adheres to standard methodologies in differentially private bandit literature, offering no novel technical insights.


(b) Clarity: The paper is clearly written and well-organized, with a logical progression from motivation to empirical validation. However, it assumes substantial prior knowledge of differential privacy and bandit algorithms, such as KL-UCB and IMED, which may hinder accessibility for readers less familiar with these areas. To improve clarity, the authors could provide additional background on these concepts, perhaps in the introduction or an appendix, to broaden the paper’s accessibility.

(c) Significance： This paper addresses regret minimization in differentially private bandits, building on prior work such as Azize and Basu [2022], and achieves a tighter lower bound, enhancing theoretical understanding. However, the theorem’s restriction to Bernoulli bandits limits its applicability to other reward distributions, potentially reducing its impact on the broader field. The algorithms’ primary novelty lies in using rewards from previous phases for parameter estimation, but the resulting improvement in regret performance is marginal, further constraining the paper’s practical significance.


(d) Originality: The paper contributes a tighter lower bound, deepening the understanding of the current differentially private bandit model, and clearly distinguishes its contributions from prior work with appropriate citations. However, the originality of the algorithms is limited, as their design heavily relies on existing frameworks like DP-KLUCB, with minimal novel components beyond the use of historical rewards.

---

> ### Author Rebuttal · Authors · 2025-07-30
>
> We thank Reviewer aTyM for the time spent reviewing, careful reading and detailed comments.
>
> **W1 - Straightforward extensions of existing methods**
>
> (a) *Novelty in algorithm design:* As explained in the Algorithm Design paragraph, there are two main novelties in our design:
>
> - Eliminating reward forgetting: This may appear as a minor change. However, it was previously believed (Hu et al. [2021]) that reward forgetting is fundamental to designing near-optimal DP bandit algorithms. Many works in similar settings (e.g., linear/contextual bandits, best arm identification) adopted forgetting. We show it is not necessary, and that this misconception arose from non-tight concentration bounds. Our result could inform improved bounds in related settings.
>
> - New $d_\epsilon$ indexes (Eq. 10 & 11): This is the heart of every bandit algorithm design. Our new indexes tightly balance exploration and exploitation in an $\epsilon$-aware way, targeting the lower bound. To derive them, we develop a new DP-Chernoff inequality (Proposition 1), the first in the DP bandit literature to consider jointly  the noise and the signal for tighter concentration. This result may be of independent interest.
>
> (b) *Technical novelty in the upper bound proof:* Please refer to Q3 of Reviewer NvHt. In short, the new proof handles the adaptive batching strategy and the new properties of $d_\epsilon$ indexes.
>
> **W2 - Assuming substantial prior knowledge in DP and bandits**
>
> Thank you for the suggestion. We will add an Extended Background section in the Appendix. It will include more details on KL-UCB, IMED, the Chernoff bound, and how these improve upon classic UCB in Bernoulli bandits.
>
> **W3 - Limited to Bernoulli with marginal improvement in regret**
>
> Since this is important, we stress the following points:
>
> (a) As explained in Point (c) of the Implications of Theorem 1 (Line 244), our regret lower bound holds for *any* class of distributions.
>
> (b) Proposition 2 shows that our algorithms are already $\epsilon$-DP for any distribution supported on $[0,1]$. Generalising to bounded rewards in $[a, b]$ just requires scaling the noise terms by $(b - a)$. Removing forgetting relies on adaptive composition, not on any stochastic assumptions on the rewards. The only important element is to be able to control the sensitivity of the empirical mean to guarantee DP using the Laplace mechanism.
>
> (c) The parts of the paper that are only valid for Bernoulli distributions are: the concentration inequality (Proposition 1) and the regret upper bounds (Theorem 2).  It is also worth noting that the same regret upper bounds of Theorem 2 are also valid for distributions over $[0,1]$, since to prove these results, we only used the Chernoff bound for Bernoulli distributions, which is also valid for distributions over $[0,1]$.
>
> (d) We believe both the concentration inequality and regret upper bound could be extended to sub-Gaussian or exponential families. What’s unclear and more technical is whether matching upper and lower bounds (up to the constants) can be achieved beyond the Bernoulli case. See Q1 below for more.
>
> (e) Bernoulli bandits are a fundamental setting found in many applications (clinical trials, ad recommendations through clicks, etc). Studying the exact cost of DP in this setting helps focus on the tradeoffs introduced by the DP constraint, without all the technical overhead introduced by other distributions of rewards. Finally, the takeaways from our analysis can be helpful beyond Bernoullis or stochastic MABs:
>
> 1. $d_\epsilon$ is the information-theoretic quantity that tightly characterises the hardness of DP bandits. We believe that $d_\epsilon$ will appear in other DP bandit settings (e.g., contextual/linear, RL), other accuracy measures (BAI with fixed confidence/budget) or even beyond bandits (sequential hypothesis testing, sequential mean estimation, etc).
>
> 2. Forgetting is *not* a necessary design choice. Prior work (Hu et al. [2021]) conjectured it was essential to design near-optimal DP bandit algorithms. Our results show otherwise, and can improve many later works that built on this design choice.
>
> 3. Our tight concentration inequality for the sum of Bernoulli and Laplace variables is of independent interest and can improve DP accuracy in other continual observation settings.
>
> (f) All prior work in stochastic DP bandits also focused on Bernoulli or bounded rewards. For bounded rewards, designing exactly optimal algorithms is already technical (even without DP), making DP analysis harder to follow.
>
> (g) *Marginal improvement:* We would like to stress that our result improves a problem-dependent quantity that can be arbitrarily large, especially for Bernoulli means near 0 or 1. See Reviewer Xdjy's W2 and Q for more.
>
> We will include a detailed "Generalisations Beyond Bernoullis" paragraph in the updated version.
>
> **Q2 - Intuition behind Proposition 1 and differences to prior work**
>
> (a) *Intuition:* Proposition 1 is a DP analogue of Chernoff’s bound (Lemma 11). It controls the concentration of the empirical mean of Bernoulli variables estimated privately by aggregating Laplace noise over phases. The estimator includes $n_m$ Bernoulli variables and $m$ Laplace noises. The key insight is: if $m = o(n_m)$ (which holds under our geometric batching), then the dominant term in concentration is $e^{- n_m d_\epsilon}$. This is the main ingredient needed to prove a regret bound in $\log(T)/d_\epsilon$. Prior work focused on reducing noise additions but sacrificed tightness in concentration.
>
> (b) *Comparison to prior work:* Lemma 4 in Azize and Basu [2022] (and others) gives concentration of noisy means using Laplace noise. However, as noted in Point (c) after Proposition 1 (Line 269), they apply a union bound that *separates* the Bernoulli and Laplace components, then apply Hoeffding to each. Using this naive concentration in our 'no-forgetting' setting would result in regret bounds with extra poly-log$(T)$ factors. To match exactly the lower bound, our proof uses a *coupled treatment* of Laplace noises and Bernoulli variables. Please refer to Appendix D for the details of this coupled treatment.
>
> **Q1 - Extension beyond Bernoulli bandits**
>
> As stated in W3, the two key components needing generalisation beyond Bernoulli are Proposition 1 and Theorem 2. We believe the “coupled treatment” approach from Proposition 1 could extend to exponential and sub-Gaussian families. What is unclear is whether the $d_\epsilon$ term would still appear in the exponent. As detailed in Q2 above, this is key to matching the lower bound.
>
> Finally, as the current privacy analysis is only valid for bounded rewards, to have a complete privacy/utility analysis, the class of rewards to be considered should be something like 'bounded sub-Gaussians' or 'bounded exponential families'.  Achieving matching bounds in these distributions is already technically challenging, even without DP.
>
> **Q3 - Application to clinical trials**
>
> In adaptive clinical trials, treatments are allocated sequentially. At step $t$, patient $p_t$ arrives, the doctor assigns treatment $a_t$, and observes reaction/reward $r_t$ ($=1$ if cured, $0$ otherwise). The goal is to maximise cured patients (minimise regret) while protecting the privacy of the patients.
>
> Here, the private data is $(r_1, \dots, r_T)$, since the reaction of a patient may reveal sensitive health information. The privacy risk is that publishing actions $(a_1, \dots, a_T)$ might leak information about the private rewards $(r_1, \dots, r_T)$.
>
> Running DP-KLUCB/DP-IMED to recommend adaptively the allocations $(a_1, \dots, a_T)$ ensures that even if an adversary knows the reactions of all but one patient, the adversary cannot infer that patient’s membership to the trial by looking at $(a_1, \dots, a_T)$. The $\epsilon$-DP guarantee controls this inference risk via a trade-off between Type I and II errors of the adversary, depending on $\epsilon$. Among all algorithms satisfying that same privacy guarantee, DP-KLUCB and DP-IMED achieve the lowest regret.
>
> The same applies to other DP bandit applications, like ad recommendation through users' clicks or private hyper-parameter tuning of models.
>
> We hope that our response addresses the reviewer's questions, and will convince them to raise their score.

---

> ### Comment · Reviewer_aTyM · 2025-08-05
> **Siginificance +1 Overall score +1**
>
> Thanks for the author's detailed response to my comment. Your explanation of the regret lower bound, demonstrating its applicability to any distribution, effectively addresses my concerns about the generalization of your results. However, the concentration inequality in Proposition 1, which is limited to the Bernoulli distribution, may still represent a limitation in the result's broader applicability.
>
> Regarding the novelty of your algorithm, I understand that the construction of $d_\epsilon$ is a key innovative aspect. However, after reviewing Azize and Basu [2022], I still believe the improvement is minor, since the algorithm heavily relies on the "Forgotten" method, DP-KLUCB, and DP-IMED.
>
> Your discussion on applying the algorithm to clinical trials convincingly illustrates its real-world potential. However, the assumption of a discrete reward $r_t$, restricted to values 0 and 1 raises questions about whether your results could be extended to continuous rewards.
>
> Based on these considerations, I have decided to increase the scores for Significance and Overall by one point each.

---

> > ### Author Response · Authors · 2025-08-06
> >
> > We are glad that our rebuttal addressed your concerns, and thank you for raising your score. We will add a paragraph on "Generalisations Beyond Bernoullis" to the main paper, an Extended Background section, and an additional motivation section in the appendix to incorporate all of your constructive feedback.

---

### Official Review · Reviewer_NvHt · 2025-07-02

**Clarity:** 3
**Significance:** 2
**Originality:** 3
**Rating:** 4
**Confidence:** 2

**Summary:**

The paper studies the $K$-armed bandit problem under $\epsilon$-global differential privacy (DP) and establishes matching lower and upper regret bounds up to a constant $\alpha$ that can be made arbitrarily close to one. The contributions include: a tighter lower bound (Theorem 1) based on a new divergence that interpolates between KL-divergence and total variation distance; an improved concentration inequality (Proposition 1); and differentially private variants of two asymptotically optimal bandit algorithms (KL-UCB and IMED), which are shown to match the regret lower bound. The paper also includes experimental results comparing the performance of the proposed algorithms to prior methods (DP-SE, AdaP-KLUCB, Lazy-DP-TS); see Figure 1.

**Questions:**

1. What is the limitation in extending the concentration inequality for Bernoulli plus Laplace noise to more general reward distributions beyond Bernoulli?

2. Shouldn’t the limitation to Bernoulli reward distributions be reflected in the title of the paper?

3. Is it correct to say that the analysis of DP-KLUCB and DP-IMED follows similar lines as the analyses of their non-private variations, with the key difference being the use of the concentration inequality in Proposition 1?

4. In the definition of $d_\epsilon(x, y)$ (Eq. (6)), the infimum is taken over $z \in [x \wedge y, x \vee y]$. Is this infimum always attained for Bernoulli distributions, and how computationally efficient is it to evaluate this in practice when applying the algorithm?

**Ethical Concerns:**

["NO or VERY MINOR ethics concerns only"]

**Final Justification:**

As mentioned in my review, the technical content of the paper is solid. However, the work addresses a rather niche problem: closing the constant gap between lower and upper bounds for differentially private multi-armed bandits, in a narrow setting: Bernoulli rewards. Overall, I maintain my evaluation of the submission that is weak accept.

**Quality:**

3

**Strengths And Weaknesses:**

The paper is well written.

It closes a gap in the literature regarding the performance of differentially private bandit algorithms. In particular, Table 1 reports the lower and upper bounds from Azize and Basu [2022], which exhibit a constant gap (assuming the KL divergence is within constant factors of the squared reward gap). This paper improves that gap to an arbitrarily close-to-1 constant by introducing a new information-theoretic quantity that interpolates between KL divergence and total variation distance. See Theorem 1, Corollary 1, and Theorem 2. The lower bound is proven using standard change-of-measure techniques from bandit theory (e.g., Lai & Robbins, 1985). While I did not go through all the proofs in full detail, the results appear sound and consistent with prior techniques. The regret analysis for the proposed algorithms builds on the corresponding non-private analyses, leveraging the new concentration inequality in Proposition 1.

A limitation is that the results apply only to Bernoulli reward distributions, which restricts the generality of the findings. While the technical content is solid and compelling, the paper addresses a rather niche problem (closing the constant gap between lower and upper bounds for differentially private multi-armed bandits) in a narrow setting (Bernoulli rewards). The limited scope of the contribution is my main reservation about the paper.

Minor comment:
Typo on line 85: “uppe bound” → “upper bound”

---

> ### Author Rebuttal · Authors · 2025-07-30
>
> We thank Reviewer NvHt for the time spent reviewing, careful reading and precise questions.
>
> **W/Q1/Q2 - Limitation to Bernoulli distributions.**
>
> Here are the main elements about this point : (a) the regret lower bound (Equation 9) and privacy analysis (Proposition 2) in the paper are already valid beyond Bernoullis, (b) only the concentration inequality and the regret upper bound are specific to Bernoullis,  (c) it is possible to extend these two results, beyond Bernoullis, to sub-Gaussian or exponential distributions. (d) what is less clear is whether it is possible to obtain matching upper and lower bounds up to the constant for these distributions.  (e) Bernoulli bandits are a fundamental setting found in many applications and the takeaways from our analysis can be helpful beyond Bernoullis or stochastic MABs, (f) all prior work in bandits with DP is for either bounded or Bernoulli rewards, since it is important to control the sensitivity of the empirical mean to use the Laplace mechanism, (g) the improvement we introduce in this setting is non-trivial, and not just a constant factor (see answers to W2 and Q for Reviewer Xdjy).
>
> Please refer to the answers to W3 and Q1 of Reviewer aTyM for a detailed discussion about each point.
>
> **Q3 - The analysis of DP-KLUCB/DP-IMED compared to their non-private versions.**
>
> Indeed, the analysis of DP-KLUCB and DP-IMED follows the general structure of the proof of their non-private versions. However, as explained in the Proof Sketch after Theorem 2, there are some technical challenges in the proof: (i) The main one is dealing with the adaptive batching strategy needed for privacy. To deal with this, the proof starts with a new regret decomposition different from the classic ones used in the non-private analysis, (ii) Dealing with the new $d_\epsilon$-based indexes (Equations 10 and 11), in contrast to the KL-based ones in the non-private analysis. For the analysis to still work, we had to prove some technical properties on the $d_\epsilon$ divergence (Lemmas 7, 8 and 9).
>
> **Q4 - The infimum in $d_\epsilon$ and computational efficiency.**
>
> (a) The expression of $d_\epsilon$: For Bernoulli distributions, we present a closed-form expression of $d_\epsilon(\mu_a, \mu^\star)$ in Equation 8. This expression shows that the infimum is always attained, and even provides a closed-form expression of the value at which this infimum is attained:
>
> - In the low privacy regime (when $\epsilon$ is big), $d_\epsilon$ reduces to the kl, which means that the infimum is attained at the left corner of the interval, i.e., $z^\star = \mu_a$.
>
> - In the high privacy regime, the expression of Equation 8 indicates that the values at which the infimum is attained is $z^\star = \frac{\mu^\star}{\mu^\star + (1 - \mu^\star) e^\epsilon}$, which is always inside the interval $[\mu_a, \mu^\star]$ for the values of $\epsilon$ in this regime. Finally, when $\epsilon \rightarrow 0$, the infimum is attained at the right corner of the interval, i.e., $z^\star = \mu^\star$. Please refer to Lemma 7 in the appendix for a proof of Equation 8.
>
> - Beyond Bernoullis, there may be no closed-form solution for $d_\epsilon$, which depends on whether there are closed-form expressions of the KL and the TV. For example, the TV between two Gaussians with different covariances has no simple formula. On the other hand, the infimum can be shown to always exist, as the expression to be minimised is an interpolation of the kl and tv, so it is continuous and convex on a compact interval.
>
> (b) Computational efficiency: Again, for Bernoullis, we have a closed-form expression in Equation 8, and it is a straightforward computation. This may indeed be a different question for distributions beyond Bernoullis. But again, since the objective is convex and the interval is compact, it is possible to run off-the-shelf convex optimisation solvers to compute $d_\epsilon$ for distributions beyond Bernoullis.
>
> We thank the Reviewer for this question, which we will include in an added 'Generalisations Beyond Bernoullis' paragraph.
>
> We hope that our response addresses the reviewer's questions, and will convince them to raise their score.

---

> > ### Comment · Reviewer_NvHt · 2025-08-04
> >
> > Thank you for your response addressing my questions regarding the algorithm and analysis. As mentioned in my review, the technical content of the paper is solid. However, the work addresses a rather niche problem: closing the constant gap between lower and upper bounds for differentially private multi-armed bandits, in a narrow setting: Bernoulli rewards. The limited scope of the contribution remains my main reservation about the paper. In response to reviewers aTyM and 64S8, the authors have elaborated on the real-world application to clinical trials, which I support including in the paper. Overall, I maintain my evaluation of the submission.

---

### Official Review · Reviewer_64S8 · 2025-07-03

**Clarity:** 3
**Significance:** 2
**Originality:** 3
**Rating:** 4
**Confidence:** 3

**Summary:**

This paper studies differentially private (DP) multi-armed bandits. It first provides an improved lower bound of the regret of bandits algorithms under the constraint of differential privacy. Then, it proposes two algorithms for DP bandits, and proves that their regret bounds match with the lower bound up to a constant factor. Finally, the synthetic experiments validates the theoretical results.

**Questions:**

1. Why the authors would like to propose two algorithms? Any specific reasons, for example, their design ideas are fundamentally different?

**Ethical Concerns:**

["NO or VERY MINOR ethics concerns only"]

**Final Justification:**

The authors have addressed my concerns. I decide to keep my score.

**Limitations:**

Yes.

**Paper Formatting Concerns:**

No further concerns.

**Quality:**

3

**Strengths And Weaknesses:**

Strengths:
1. The theoretical results seem solid and important. Both lower and upper bounds are provided and they are matching up to a constant factor.
2. The presentation is clear because it lists the key improvements in the proofs.

Weaknesses:
1. The setting is limited to multi-armed bandits and Bernoulli distribution, while linear or contextual bandits are more general models. In particular, Proposition 1 is limited to Bernoulli distribution. It is unclear that Gaussian or sub-Gaussian distribution can also have similar results so that the algorithm is still optimal.
2. The experiments could be benefit from some real-world dataset.

---

> ### Author Rebuttal · Authors · 2025-07-30
>
> We thank Reviewer 64S8 for the time spent reviewing, careful reading and interesting suggestions.
>
> **W1 - The setting is limited to multi-armed bandits and Bernoulli distribution.**
>
> Here are the main elements about this point : (a) the regret lower bound (Equation 9) and privacy analysis (Proposition 2) in the paper are already valid beyond Bernoullis, (b) only the concentration inequality and the regret upper bound are specific to Bernoullis,  (c) it is possible to extend these two results, beyond Bernoullis, to sub-Gaussian or exponential distributions. (d) what is less clear is whether it is possible to obtain matching upper and lower bounds up to the constant for these distributions.  (e) Bernoulli bandits are a fundamental setting found in many applications and the takeaways from our analysis can be helpful beyond Bernoullis or stochastic MABs, (f) all prior work in bandits with DP is for either bounded or Bernoulli rewards, since it is important to control the sensitivity of the empirical mean to use the Laplace mechanism, (g) the improvement we introduce in this setting is non-trivial, and not just a constant factor (see answers to W2 and Q for Reviewer Xdjy).
>
>
> Please refer to the answers to W3 and Q1 of Reviewer aTyM for a detailed discussion about each point.
>
>
> **W2 - Real-world dataset in experiments.**
>
> We thank the Reviewer for this suggestion. Indeed, our current experimental analysis is only done on simulated environments. There are two reasons for this:
>
> (a) Starting from the DP-SE paper, most experiments on DP bandit papers (Sajed and
> Sheffet [2019], Hu et al. [2021], Azize and Basu [2022], Hu and Hegde [2022]) have been run on the same sets of Bernoulli environments and privacy budgets. This helps in providing a fair comparison of the algorithms, run using the same hyperparameters as the original papers.
>
> (b) Working with simulated environments helps in providing a complete picture by varying multiple parameters that control the hardness of the problem.
>
> We agree that the paper could benefit from adding a real-world dataset. In the Extended Experiments section, we will add an experiment inspired by the COV-BOOST [1] dataset. COV-BOOST is a phase 2 clinical trial, conducted on 2883 participants to measure the immunogenicity of different Covid-19 vaccines as a third dose. This resulted in a total of 20 vaccination strategies being assessed, each of them defined by the vaccines received as first, second and third doses. In [1], the authors have reported the average immune responses induced by each candidate strategy on cohorts of participants. From this study, we extract and process these average immune responses for each strategy to simulate Bernoulli bandits with $K = 20$ arms, and run our algorithms with different values of $\epsilon$. Though this is still a 'semi-synthetic' experiment, it would illustrate the performance of the DP algorithms in more practical scenarios. We will report the evolution of regret for this specific Bernoulli instance, under different values of $\epsilon$ in Appendix G of the updated version. DP-KLUCB and DP-IMED still achieve the lowest regret for this instance.
>
> [1] A.P.S. Munro, L. Janani, V. Cornelius, and et al. Safety and immunogenicity of seven COVID-19 vaccines as a third dose (booster) following two doses of ChAdOx1 nCov-19 or BNT162b2 in the UK (COV-BOOST): a blinded, multicentre, randomised, controlled, phase 2 trial. The Lancet, 398(10318):2258–2276, 2021.
>
> **Q - On proposing two algorithms.** The reason for presenting both DP-KLUCB and DP-IMED is to show that, for two different algorithmic design philosophies in bandits (KL-UCB and IMED), our privacy framework of geometric batching without forgetting, combined with our new concentration inequality, can design algorithms with optimal regret upper bounds.
>
> (a) KL-UCB and IMED belong to fundamentally different algorithmic bandit families:
>
> - KL-UCB is a UCB-style algorithm that relies on *optimism in the face of uncertainty*, and constructs upper confidence bounds based on Chernoff's inequality
>
> - IMED is an information-theoretic method that selects arms based on *empirical divergence minimisation*, comparing empirical rewards to the estimated best arm.
>
> We will add an extended discussion about these two families of algorithms in the Extended Background section in the appendix, as suggested by Reviewer aTyM.
>
>
> (b) Our proposed privacy framework works for both KL-UCB and IMED: our framework estimates the unknown means privately by running the algorithm in geometrically increasing phases, and accumulating Laplace noises from each phase, i.e. no forgetting. In addition, our tight DP-Chernoff concentration inequality (Proposition 1) directly provides new $d_\epsilon$-based indexes for both KL-UCB and IMED style algorithms (Equations 10 and 11), tightly balancing exploration and exploitation under noisy DP observations. Combining everything with a generic regret upper bound analysis provides two optimal DP bandit algorithms.
>
> We hope that our response addresses the reviewer's questions, and will convince them to raise their score.

---

> > ### Comment · Reviewer_64S8 · 2025-08-05
> >
> > Thank you for your response and for addressing my concerns. I have no further questions.

---

### Official Review · Reviewer_Xdjy · 2025-07-03

**Clarity:** 3
**Significance:** 3
**Originality:** 3
**Rating:** 5
**Confidence:** 2

**Summary:**

This paper studies the classical DP MAB problem, with an objective of closing the instance-dependent upper and lower bound that existed in previous works, namely $\Delta_a^{-2}$ v.s. $\text{kl}(\mu_a,\mu_\ast)^{-1}$. The authors proposed a new measure $d_\epsilon$ such that 1) a new lower bound of $d_\epsilon^{-1}$ is given, and 2) a new algorithm achieving $d_\epsilon^{-1}$ is also yielded.

**Questions:**

Line 95--96: "However, this approximation can be arbitrarily bad, exposing a gap between the state-of-the-art upper and lower bounds in DP bandits." What does "arbitrarily bad" exactly mean?

**Ethical Concerns:**

["NO or VERY MINOR ethics concerns only"]

**Final Justification:**

I am satisfied with Authors' responses and decide to keep my original evaluation.

**Limitations:**

Yes

**Quality:**

3

**Strengths And Weaknesses:**

Strength:
1. The exact match of two $d_\epsilon^{-1}$ is exciting. Previously the upper and lower bounds only agree in an asymptotical setting (but now still this case due to $\alpha$; nevertheless the $\alpha$ here can be arbitrarily close to $1^+$ in contrast to the previous instance-dependent approximation).
2. The algorithm design illustrates that it is possible to get rid of "forgetting", thus offering more algorithmic tools in the future. However, I have to admit that I'm not super familiar with DP literature, and therefore I am not super certain whether this innovation is significant enough.
3. Numerical illustraions are provided, which supports the superiority of proposed algorithms.

Weakness:
1. The match between upper and lower bounds only holds in asymptotic settings where one can reduce $\alpha\to 1^+$ as much as they want.
2. It is unclear how much improvement is really yielded by the new $d_\epsilon$ measure, given that $\text{kl}$ and $\Delta_a^{-2}$ are already approx. equal in asymp. cases.

Overall, I feel the improvements may look kind of "marginal" given the previously already-good (but non-perfect) upper and lower bounds. However, given the perfect matching coefficient (if I did not misunderstand), I feel this "perfect" result deserves to be presented.

---

> ### Author Rebuttal · Authors · 2025-07-30
>
> We thank Reviewer Xdjy for the time spent reviewing, careful reading, and kind words about the novelty and the significance of the contributions. We agree that highlighting the perfect match between our regret upper and lower bounds may be of interest to the community, as it is rare in online learning with DP constraints.
>
> **W1 - The upper and lower bound match when $\alpha \rightarrow 1$.**
>
> Indeed, our regret upper bound and lower bound match up to a constant $\alpha >1$, where $\alpha$ is the ratio of the geometrically increasing batch sizes. This parameter $\alpha$ can be set arbitrarily close to 1 to have 'perfect' matching upper and lower bounds. Still, it is easy to remove the constant $\alpha>1$ since the only essential requirement in our analysis is that the number of phases is sublinear in $T$ (in other words, the batch size $B_m$ goes to $\infty$). We choose the batch size of $B_m \approx \alpha^m$ just for simplicity, which is practically sufficient. This point is discussed in Appendix F, and we will make it explicit in the revised version.
>
>
> **W2 - The improvement yielded by $d_\epsilon$.**
>
> (a) In the regret lower bound: the improvement reduces to a comparison between our $d_\epsilon$ and the $\min \lbrace \mathrm{kl}, 6 \epsilon \cdot \mathrm{tv} \rbrace$ of Azize and Basu [2022]. In the high privacy regime ($\epsilon \rightarrow 0$), $d_\epsilon$ reduces to $\epsilon \cdot \mathrm{tv}$ and thus improves over a $6$ factor from the lower bound of Azize and Basu [2022]. In the low privacy regime (big $\epsilon$), both expressions reduce to the $\mathrm{kl}$, which is the tightest quantity for non-private bandits. Finally, for intermediate values of $\epsilon$, we always have that $d_\epsilon < \min\lbrace\mathrm{kl}, 6 \epsilon \cdot \mathrm{tv} \rbrace$ as explained in point (a) after Theorem 1 (Line 235-236).
>
> (b) In the regret upper bound: the improvement reduces to a comparison between our $d_\epsilon$ and the $O(\min\lbrace \Delta_a^2, \epsilon \Delta_a \rbrace)$ of Azize and Basu [2022]. Again, in the high privacy regime, we improve over constants. However, in the low privacy regime, the improvement reduces to a comparison between $\mathrm{kl}$ and $\Delta_a^2$. This improvement is a problem-dependent quantity that can be arbitrarily good for some instances. Refer to the answer to Q below for a detailed discussion. This is also the case for intermediate values of $\epsilon$.
>
>
>
> (c) $d_\epsilon$ is a smooth interpolation between the TV and the KL, that has nice transport properties allowing it to be a good index for DP-KLUCB and DP-IMED. These smoothness properties are also important for the regret upper bound proof.
>
> **Q - Meaning of arbitrarily bad in Line 95-96.**
>
> In Line 95-96, we are talking about approximating the $\mathrm{kl}(\mu_a, \mu^\star)$ with the squared gap $\Delta_a^2 = (\mu^\star - \mu_a)^2$.
>
> (a) Using Pinsker's inequality, we always have that $\mathrm{kl}(\mu_a, \mu^\star) \geq 2 \Delta_a^2$
>
> (b) However, for close values of $\mu_a$ and $\mu^\star$, a Taylor expansion shows that
> $$ \mathrm{kl}(\mu_a, \mu^\star) = \frac{\Delta_a^2}{2 \mu_a (1 - \mu_a)} + o(\Delta_a^2) $$
> which means that $$\frac{\Delta_a^{-2}}{\mathrm{kl}(\mu_a, \mu^\star)^{-1}} = \frac{1}{2\mu_a(1 - \mu_a)} + o(1) \rightarrow \infty$$
> when $\mu_a$ tends to either $0$ or $1$. This means that our algorithms improve over the state-of-the-art algorithms in a problem-dependent constant (related to the variance of Bernoullis), which could blow up for some hard instances close to the borders $0$ and $1$.
>
> We will add a discussion about this in the Extended Background section in the Appendix, as suggested by Reviewer aTyM.

---

> > ### Comment · Reviewer_Xdjy · 2025-08-03
> >
> > Thank you for the responses. My concerns are resolved.

---

### Decision · Program_Chairs · 2025-09-17

**Decision:**

Accept (poster)

**Comment:**

The paper studies the multi-armed bandit problem with global differential privacy, and establishes matching lower and upper regret bounds. The contributions include: a tighter lower bound based on a new divergence that interpolates between KL-divergence and total variation distance, an improved concentration inequality, and differentially private variants of two asymptotically optimal bandit algorithms that are shown to match the regret lower bound. Although this paper addresses a rather niche problem (closing the constant gap between lower and upper bounds for differentially private multi-armed bandits) in a narrow setting (of Bernoulli rewards), which limit its contributions and broader impact, the reviewers all agree that this paper is technically solid. Based on their comments, I recommend Accept.

The authors are encouraged to incorporate the reviewers’ comments and polish the presentation during preparing the final version.